# New Remains of *Scandiavis mikkelseni* Inform Avian Phylogenetic Relationships and Brain Evolution

**Miriam Heingård** [1,*] **, Grace Musser** [2,*] **, Stephen A. Hall** [3] **and Julia A. Clarke** [2]

[1] Department of Geology, Lund University, SE-223 62 Lund, Sweden
[2] Department of Geological Sciences, The University of Texas at Austin, Austin, TX 78712, USA; Julia_Clarke@jsg.utexas.edu
[3] Department of Construction Sciences, Lund University, SE-221 00 Lund, Sweden; stephen.hall@solid.lth.se
[*] Correspondence: miriam.heingard@geol.lu.se (M.H.); gmusser@utexas.edu (G.M.)

**Abstract:** Although an increasing number of studies are combining skeletal and neural morphology data in a phylogenetic context, most studies do not include extinct taxa due to the rarity of preserved endocasts. The early Eocene avifauna of the Fur Formation of Denmark presents an excellent opportunity for further study of extinct osteological and endocranial morphology as fossils are often exceptionally preserved in three dimensions. Here, we use X-ray computed tomography to present additional material of the previously described taxon *Scandiavis mikkelseni* and reassess its phylogenetic placement using a previously published dataset. The new specimen provides novel insights into the osteological morphology and brain anatomy of *Scandiavis*. The virtual endocast exhibits a morphology comparable to that of modern avian species. Endocranial evaluation shows that it was remarkably similar to that of certain extant Charadriiformes, yet also possessed a novel combination of traits. This may mean that traits previously proposed to be the result of shifts in ecology later in the evolutionary history of Charadriiformes may instead show a more complex distribution in stem Charadriiformes and/or Gruiformes depending on the interrelationships of these important clades. Evaluation of skeletal and endocranial character state changes within a previously published phylogeny confirms both *S. mikkelseni* and a putative extinct charadriiform, *Nahmavis grandei*, as charadriiform. Results bolster the likelihood that both taxa are critical fossils for divergence dating and highlight a biogeographic pattern similar to that of Gruiformes.

**Keywords:** Paleogene; Charadriiformes; Neoaves; endocast; brain evolution; Gruiformes; avian phylogeny; fossil; divergence time estimation





## 1. Introduction

The early Eocene Fur Formation of north-western Jutland, Denmark, is well-known for its rich fossil bird fauna that includes three-dimensionally preserved, articulated specimens, as well as occasional soft tissues [1–3]. These fossils constitute one of the oldest diverse Cenozoic bird faunas [1,3], existing only 11 Ma after the Cretaceous–Paleogene mass extinction during a time of rapid neornithine radiation that saw the appearance of most major extant avian clades [4–6]. The diatomaceous sediments of the Fur Formation are also unusual in the Paleogene fossil record of birds in deriving from a marine offshore environment [3]. Articulated, three-dimensional bird skeletons in this formation are typically preserved within hard calcareous carbonate concretions, believed to have protected the remains from compaction [3,7].

The age and exceptional level of anatomical detail preserved within recovered fossils from the Fur Formation makes them critical to better understanding early neornithine evolution. Despite this, many of the taxa have not been studied in detail. Previous research has identified different taxa assigned to some of the major clades of modern birds, including Apodiformes [8,9], Pelecaniformes [10], Coliiformes [8], Galliformes [11],

Coraciiformes [12], and Psittaciformes [13]. In addition, members of the extinct Messelornithidae [14,15] and Lithornithidae [16] have been reported. Putative charadriiform birds have also been recovered from this formation, such as the exquisitely preserved holotype specimen of *Scandiavis mikkelseni* Bertelli et al. [17]. However, charadriiform affinities were recovered for *S. mikkelseni* with low support in the phylogenetic analyses of Bertelli et al. [17], and it was placed as either a charadriiform or gruiform in the results of Musser and Clarke [18].

In recent decades, advances in non-destructive imaging techniques, including X-ray computed tomography, have increasingly allowed more information to be gained from well-preserved avian specimens [19–23]. These techniques make it possible to visualize otherwise inaccessible skeletal material, but also provide a tool to reconstruct and investigate neural anatomy, by digitally constructing endocranial casts [24,25]. In mammals and birds, the space between the brain and endocranium is so limited that the internal surface of the braincase provides a reasonable approximation of both the size and morphology of the brain. Therefore, virtual endocasts provide relatively accurate estimates of the linear, volumetric and geometric proportions of the brain [21,26,27]. Natural endocasts are extremely rare (e.g., [28,29]). Consequently, the introduction of digital three-dimensional (3D) reconstructions has resulted in significant progress in avian paleoneurology (e.g., [30–41]). However, fossil specimens with three-dimensionally preserved cranial material suitable for CT scanning are comparatively rare and many lineages on the avian evolutionary tree are unrepresented with respect to neural development. As a result, the evolution of the avian brain remains poorly understood. Furthermore, despite its importance for understanding post-Cretaceous avian evolution, specimens from the Fur Formation avifauna have not been studied with regard to brain morphology.

Here, we employ high-resolution computed tomography (µCT) to describe new remains of *Scandiavis mikkelseni* from the Fur Formation. This specimen, NHMD 625345, consists of a fully articulated skull preserved in three dimensions. Analysis of CT data of the new skull allowed visualization and description of the cranium, and reconstruction of the neural anatomy of *Scandiavis* for the first time. We first compared the skull to previously published taxa recovered from the Fur Formation and assigned the skull to *Scandiavis mikkelseni* based on identical features and measurements. A differential diagnosis distinguishing this taxon from the morphologically similar *Nahmavis grandei* Musser and Clarke [18] and an emended description of *Scandiavis* is also provided. The results of this work provide novel information on avian neural development during the early Paleogene. Furthermore, phylogenetic analysis of both osteological and endocranial characters further constrains the affinities of this taxon by more robustly placing both *S. mikkelseni* and *N. grandei* within the Charadriiformes, primarily as stem-Charadriiformes. This new specimen is another example of the excellent preservation of early Eocene avifauna and testifies to the potential of the Fur Formation in providing material that is crucial to better understanding avian evolution.

## 2. Materials and Methods

### 2.1. µCT Scanning and Analysis

CT scanning of the skull of NHMD 625345 was performed at the 4D-Imaging Laboratory, Division of Solid Mechanics, Lund University, Sweden, using a Zeiss XRadia Versa XRM520 (Pleasanton, CA, USA). The specimen was scanned with a source voltage of 160 kV and reconstructed to provide 16-bit TIFF files, with a cubic voxel width of 53.3 µm. The two rock slabs containing the fossil sub-parts were attached to each other during CT scanning to obtain data of the intact skull. The skull and endocranial cavity were segmented and virtually reconstructed from the scan slice data using 3D Slicer 4.11.20200930 (http://www.slicer.org (accessed on 25 November 2021); [42]) without down-sampling. This software was also used for the linear and volumetric measurements.

### 2.2. Comparative Materials

Specimens used for description and phylogenetic analyses came from the Bird Division of FMNH, the Ornithology Department of AMNH, and the Ornithology Department of USNM. Common anglicized equivalents of avian anatomical nomenclature are used throughout, with the exception of osteological terminology from Baumel and Witmer [43], Bertelli et al. [14], Musser and Cracraft [44], and Musser and Clarke [18]. Non-anglicized neuroanatomical terminology, such as telencephalon, mesencephalon, and eminentia sagittalis, follow Breazile and Kuenzel [45]. Specimen numbers for comparative material are provided in Table 1. For comparing neural anatomy, published descriptions, images, and CT-derived 3D models of brains and endocasts of extant as well as fossil avian taxa were used. The taxa and publications used specifically for the phylogenetic analysis are provided in Table 2.

**Table 1.** Specimen numbers of newly added taxa and skeletal specimens used for comparison during fossil description and phylogenetic analyses.

| Group Name | Species Sampled and Specimen Numbers |
| --- | --- |
| Tinamiformes | *Crypturellus undulatus* (AMNH 2751, AMNH 6479), *Tinamus solitarius* (AMNH 21983, USNM 561269, USNM 345133) |
| Galliformes | *Lophura bulweri* (AMNH 10962, AMNH 16532), *Gallus gallus* (AMNH 18555, AMNH 4031, M-12244, USNM 489422) |
| Anseriformes | *Chauna torquata* (M-10449, USNM 646637), *Anas platyrhynchos* (USNM 633396, USNM 610643), *Mergus serrator* (USNM 490105, USNM 634853, USNM 430710) |
| Gaviiformes | *Gavia immer* (AMNH 15919, USNM 501589) |
| Sphenisciformes | *Spheniscus humboldti* (AMNH 4921) |
| Phoenicopteriformes | *Phoenicopterus chilensis* (M-4923, M-5325) |
| Podicipediformes | *Podiceps cristatus* (AMNH 25241, USNM 502553, USNM 560595) |
| Caprimulgiformes | *Caprimulgus carolinensis* (USNM 559607) |
| Otidiformes | *Chlamydotis macqueenii* (USNM 430378, USNM 430485, AMNH 28665) |
| Columbiformes | *Columba livia* (USNM 555707) |
| Opisthocomiformes | *Opisthocomus hoazin* (AMNH 12127, AMNH 24230, USNM 612024, USNM 344066, USNM 344065) |
| Charadriiformes | *Eudromias ruficollis* (AMNH 7013, USNM 322963, USNM 322962), *Jacana jacana* (FMNH 376137, USNM 560148, USNM 614605, USNM 345812), *Haematopus ostralegus* (FMNH 338440, USNM 502440, USNM 560934, AMNH 1681), *Burhinus bistriatus* (AMNH 2630, FMNH 289831, USNM 621089, USNM 626233, USNM 432021), *Charadrius semipalmatus* (AMNH 9963, FMNH 342530, USNM 489728, USNM 489665, USNM 489599), *Vanellus coronatus* (USNM 636688, USNM 636689, AMNH 5243), *Chionis alba* (AMNH 549, AMNH 879, USNM 553253, USNM 490989), *Pluvianus aegyptius* (FMNH 93449, FMNH 291228, USNM 491870, USNM 500294), *Larus atricilla* (M-10469, USNM 560290, USNM 227064), *Recurvirostra avosetta* (USNM 556295, USNM 610452, AMNH 28666), *Stercorarius longicaudus* (USNM 491644, USNM 491643, AMNH 21036), *Thinocorus rumicivorous* (USNM 227504, USNM 491022, AMNH 10143), *Turnix nigricollis* (USNM 488643, USNM 432224, AMNH 1994, AMNH 5381) |
| Gruiformes | *Psophia crepitans* (AMNH 29322, FMNH 338504, USNM 621709, USNM 429974), *Aramus guarauna* (AMNH 24194, NC State 18405, FMNH 376076, USNM 612025, USNM 226809), *Balearica regulorum* (AMNH 10699, USNM 637581, USNM 647263, USNM 631784), *Grus japonensis* (AMNH 1938, AMNH 1718, AMNH 4252), *Podica senegalensis* (AMNH 4148, AMNH 4208, AMNH 5268, USNM 562803), *Heliornis fulica* (FMNH 376129, USNM 623068, USNM 19159, USNM 345807, USNM 321493, YPM 109145), *Heliopais personata* (USNM 534558, USNM 344532), *Sarothrura lugens* (AMNH 2417, AMNH 4235), *Sarothrura pulchra* (FMNH 490196, USNM 291778, USNM 292395, AMNH 4235), *Himantornis haematopus* (AMNH 4183, UNSM 318391), *Habroptila wallacii* (USNM 560793, USNM 557026, USNM 557027, USNM 572365, USNM 560792, USNM 557025), *Gallicrex cinerea* (USNM 319118, USNM 319481, USNM 489266, USNM 292017), *Canirallus oceleus batesi* (AMNH 4151), *Aramides cajanea* (AMNH 4343, AMNH 8637, FMNH 105856, USNM 612270, USNM 612266), *Rallus longirostris* (M-10359, USNM 499648, USNM), *Gallinula chloropus* (AMNH 28451, FMNH 105107, USNM 499259), *Porphyrula martinica* (USNM 611560, USNM 611561) |
| Phaethontiformes | *Phaethon aethereus* (AMNH 28494, FMNH 348136, FMNH 339435, USNM 558044, USNM 525793) |
| Eurypygiformes | *Eurypyga helias* (AMNH 3750, AMNH 4293, FMNH 376130, FMNH 106439, USNM 637209, USNM 623251, USNM 344047), *Rhynochetos jubatus* (AMNH 1326, AMNH 554, FMNH 291228, USNM 612087, USNM 018994) |

**Table 1.** *Cont.*

| Group Name | Species Sampled and Specimen Numbers |
|---|---|
| Cariamiformes | *Cariama cristata* (AMNH 1722, AMNH 8667, AMNH 8646, M-10446, FMNH 105634, USNM 555731, USNM 430173, USNM 631176), *Chunga burmeisteri* (USNM 431487, AMNH 4250) |
| Accipitriformes | *Cathartes burrovianus* (AMNH 1264, USNM 623071) |
| Leptosomiformes | *Leptosomus discolor* (AMNH 10083, USNM 291844, USNM 291845) |
| Extinct Taxa | *Nahmavis grandei* (FMNH PA778), *Scandiavis mikkelseni* ([17] and DK788), *Salmila robusta* [46,47], *Pellornis mikkelseni* (MGUH 29278, DK664, FUM 1681a; [10,15]), *Songzia acutunguis* [48], *Messelornis cristata* (USNM 462392, [14,49]) |

**Table 2.** The sampled taxa and publications for brain and osseous labyrinth characters used for the phylogenetic analysis.

| Species Sampled | Reference |
|---|---|
| *Anas platyrhynchos* | [50,51] |
| *Burhinus oedicnemus* | [52] |
| *Caprimulgus europaeus* | [52,53] |
| *Cathartes aura* | [54,55] |
| *Cathartes sp.* | [53] |
| *Charadrius vociferus* | [56] |
| *Chauna chavaria* | [51] |
| *Columba livia* | [53,57] |
| *Crypturellus obsoletus* | [39] |
| *Eurypyga helias* | [24] |
| *Gallus gallus* | [53,58] |
| *Gavia immer* | [53,59] |
| *Grus grus* | [53] |
| *Haematopus ostralegus* | [53] |
| *Larus argentatus* | [53,56] |
| *Opisthocomus hoazin* | [53] |
| *Phaethon rubricauda* | [30] |
| *Phaethon lepturus* | [53] |
| *Phoenicopterus chilensis* | [52] |
| *Phoenicopterus ruber* | [53] |
| *Pluvianus aegyptius* | [52] |
| *Podiceps cristatus* | [53] |
| *Spheniscus humboldti* | [32,59] |
| *Stercorarius longicaudus* | [56] |
| *Tinamus major peruvianus* | [60] |
| *Vanellus vanellus* | [52] |

### 2.3. Character Matrix

The data matrix is based on that of Musser and Clarke [18], an expansion on Musser and Cracraft [44]. We added 28 additional discrete morphological characters characterizing brain anatomy from Smith and Clarke [56], as the new specimen allowed for description and scoring of brain morphology. Character descriptions are provided in the Appendix A and the data matrix has been made publicly available on Morphobank [61] under Project 3614.

### 2.4. Phylogenetic Analysis

We first performed unconstrained parsimony analyses of the dataset in PAUP [62] Version 4.0a169, build 164 (86) using 10,000 random taxon addition replicates per run. Heuristic search algorithms were used. Tree bisection reconnection branch swapping was employed and minimum branch lengths valued at zero were collapsed. No character weighting was applied. Characters 245, 320, and 688 were ordered, following Bertelli et al. [14] and Musser et al. [15]. Bootstrap analyses were performed using 500 bootstrap replicates each with 10 random taxon addition replicates as in [63]. In addition to unconstrained analyses, constrained analyses following those of Musser and Clarke [18] were performed using

the topology of recent phylogenomic studies [64–66] as backbone constraints. Extinct taxa were unconstrained in all analyses.

*2.5. Institutional Abbreviations*

AMNH, American Museum of Natural History, United States; DK, Danekræ collections, Geological Museum of the University of Copenhagen, Denmark; FMNH, Field Museum of Natural History, United States; FUM, Fur Museum, Museum Salling, Denmark; HLMD, Hessisches Landesmuseum Darmstadt, Germany; M, the Texas Memorial Museum, United States; MGUH, Geological Museum of the University of Copenhagen, Denmark; NHMD, Natural History Museum of Denmark, Denmark; USNM, National Museum of Natural History, Smithsonian Institution, United States.

## 3. Results

*3.1. Systematic Paleontology*

Aves Linnaeus [67].
Neognathae Pycraft [68].
Pan-Charadriiformes Hood et al. [69].
*Scandiavis mikkelseni* Bertelli, Lindow, Dyke and Mayr [17].

### 3.1.1. Holotype Specimen

FU171x, a partial articulated skeleton of an adult individual preserved on a slab. The holotype is missing the zygomatic region of the skull, the left quadrate, the caudal portion of the mandible, the forelimbs and several ribs and pedal phalanges.

### 3.1.2. Referred Specimen

NHMD 625345 consists of a bird skull preserved within a calcium carbonate concretion (Figure 1). The skull is remarkably complete, preserved in three dimensions and shows no signs of deformation. The fossil has been split longitudinally along the medial plane, forming sub-samples that represent the right and left side of the skull, respectively. The cranial cavity has not been filled with secondary deposits but remains as an empty cavity.

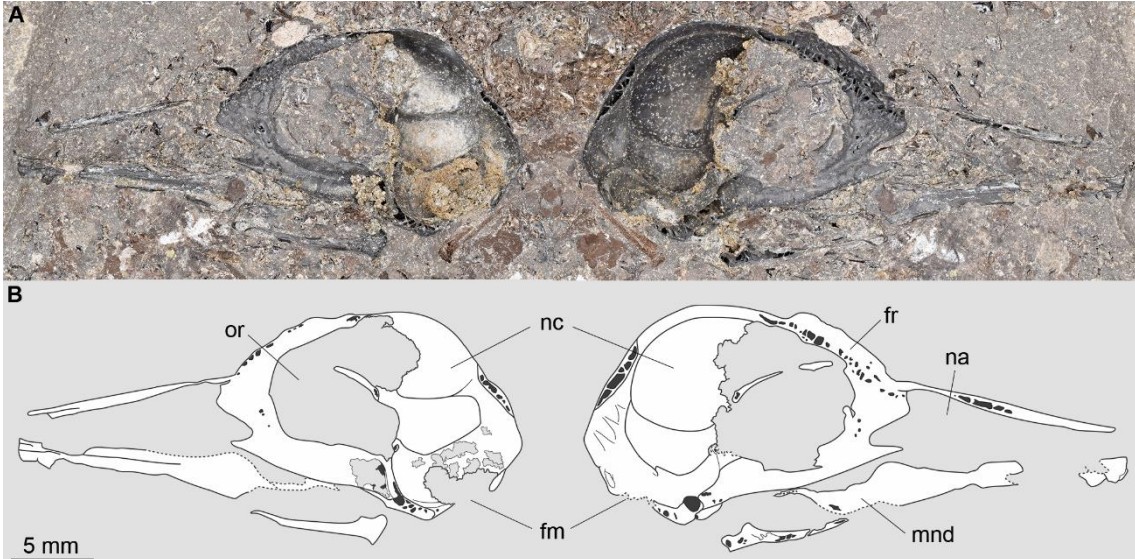

**Figure 1.** Photograph (**A**) and line drawing (**B**) of the right and left side of NHMD 625345 (*Scandiavis mikkelseni*). Bone is unfilled, the matrix is grey, and large voids in the bone are delimited in black. Anatomical abbreviations: fr, frontal bone; fm, foramen magnum; mnd, mandible; na, nares; nc, neural cavity; or, orbit.

### 3.1.3. Measurements

See Table 3.

**Table 3.** Linear measurements in millimeters (mm) of NMHD 625345, taken from the 3D rendering, compared with measurements of *Scandiavis mikkelseni* and *Nahmavis grandei*. Total skull length is measured from the anterior end of the premaxilla to the cerebellar prominence.

| | NHMD 625345 | *Scandiavis mikkelseni* | *Nahmavis grandei* | *Burhinus oedicnemus* | *Chionis alba* |
|---|---|---|---|---|---|
| Total skull length | 47.2 | 49.5 | 67.8 | 93.2 | 70.7 |
| Rostrum length | 22.3 | 24.5 | 35.1 | 50.5 | 31.7 |
| Neurocranium length | 24.9 | 25.0 | 32.7 | 50.8 | 40.8 |

### 3.1.4. Locality

Skarrehage Quarry, Silstrup Member, earliest Eocene, Fur Formation (early Ypresian), Island of Mors, Denmark, 56°56′41.5″ N 8°52′23.6″ E (WGS84). The specimen derives from the stratigraphic level containing carbonate concretions around ash layer +15 of the formation. This layer sits immediately below ash layer +19, which is currently radiometrically dated to ~55.4 Ma [70,71].

### 3.1.5. Referral and Emended Diagnosis

Prior to phylogenetic analysis, we assigned NHMD 625345 to *Scandiavis* based on the following diagnostic characters from Bertelli et al. [17]: (1) skull with long narial openings, (2) pars symphysialis of mandible with a flat ventral surface, and (3) presence of a dorsally recurved processus retroarticularis of the mandible (character 238: state 1 and 239:1 of Musser and Clarke [18]. We additionally assigned NHMD 625345 to *Scandiavis* based on the following combination of characters from Musser and Clarke [18]: a rostrum that is slightly curved ventrally at the apex (2:1), nares that are both rostrally (13:1) and caudally (10:2) rounded, nares that are over half the length of that of the rostrum (11:3), nares that are holorhinal or rostral to the zona flexoria craniofacialis (15:1), presence of furrowing along the midline of the interorbital area (32:2), presence of a supraorbital crest (49:1) that is dorsally projected (50:1), presence of a craniocaudally extensive fonticulus interorbitalis (90:2) that is not confluent with the fonticulus orbitocranialis (93:1), an occipital that is subequal in rostrocaudal position to that of the nuchal crest (158:2), a nuchal crest that is ventral to the dorsal base of the postorbital process (159:1), and a symphysis of the mandible that is less than 1/5 the length of the mandible (216:1). Phylogenetic analysis of NHMD 625345 were additionally performed with NHMD 625345 as a separate taxon. Results placed NHMD 625345 as the sister-taxon of *Scandiavis*.

### 3.1.6. Differential Diagnosis from *Nahmavis grandei*

*Scandiavis mikkelseni* differs from *Nahmavis grandei* in the following combination of character states: (1) nares that are more rostrocaudally and dorsoventrally extensive, (2) a more obtuse antorbital angle that is approximately 90 degrees, (3) a more rostrocaudally truncate cranium, (4) a rostroventrally oriented postorbital process [18], (5) a dorsoventrally wider rostral fenestra of the mandible, (6) a more prominent dorsal mandibular angle, (7) an articular of the mandible that is located markedly ventral to the ramus, (8) a more cranially projected crista cnemialis cranialis of the tibiotarsus [18], (9) a crista cnemialis cranialis with a more rounded distal apex than that of *N. grandei*, (10) presence of a notch along the distal rim of the medial condyle of the tibiotarsus, (11) a more shallow fossa parahypotarsalis lateralis in the tarsometatarsus, and (12) a shorter femur and tibiotarsus.

*3.2. Description*

3.2.1. Cranium

Three-dimensional skeletal remains of NHMD 625345 are visible on the exposed surfaces of the slabs, including the endocranial cavity, the frontal and orbital region, as well as most of the bill and mandible (Figure 1). CT data (see Appendix A) revealed skull morphology hidden by the sediment matrix (Figure 2). The maximum length of the skull is 47.2 mm. The nares are extensive, taking up almost the entire length and height of the rostrum as noted in the diagnosis of *Scandiavis* based on the holotype specimen [17]. This is similar to the condition in *Nahmavis grandei*, *Burhinus*, *Charadrius*, and *Jacana*, although *N. grandei* and the extant taxa exhibit more of the terminal premaxilla that is not perforated by the nares. This is in contrast to *Haematopus* and many Gruiformes including *Psophia*, *Heliornis*, *Sarothrura*, and *Himantornis* that exhibit nares that are approximately half the length of the rostrum. As in *Burhinus* and *N. grandei*, the nares are rostral to the zona flexoria craniofacialis and would have likely been rounded at the caudal terminus, as in holorhinal taxa. All Gruiformes are holorhinal as well. Schizorhinal, caudally acuminate nares that extend caudal to the zona flexoria craniofacialis are present in *Haematopus*, *Jacana* and *Charadrius*. The caudal margins of the nares are not well preserved but were rounded. The rostral terminus of the premaxilla has been pushed dorsally due to diagenesis. However, it is clear that the rostral terminus of the premaxilla is swollen dorsally and recurved as in the holotype. This condition is also present in *N. grandei*, *Jacana*, and *Charadrius*.

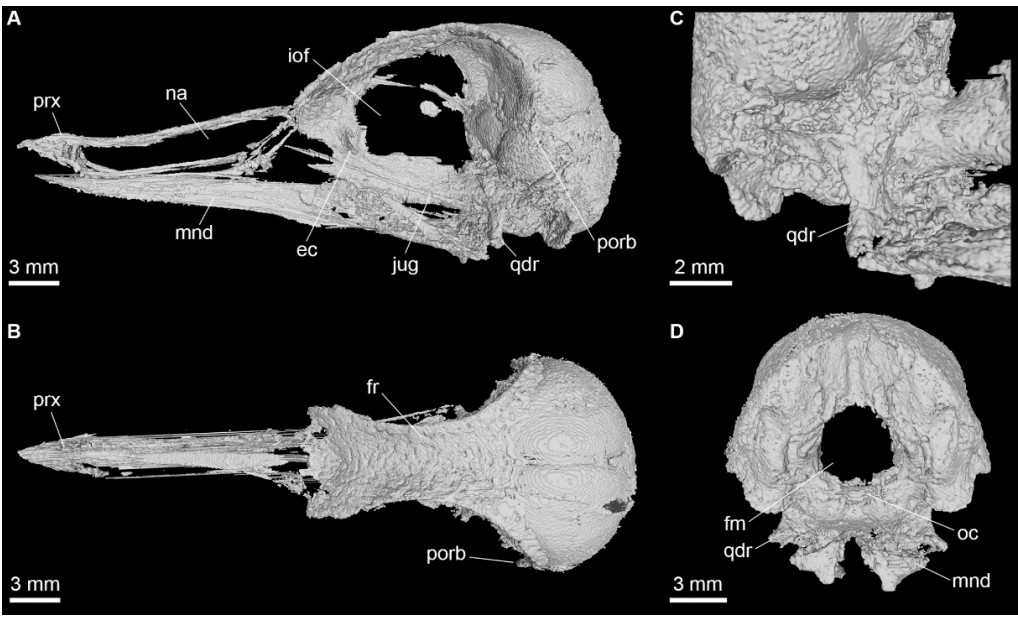

**Figure 2.** µCT visualization of *Scandiavis mikkelseni* (NHMD 625345) in (**A**) lateral, (**B**) dorsal, (**C**) rostrolateral, and (**D**) caudal aspect. (**C**) shows the right quadrate in lateral aspect. Anatomical abbreviations: ec, ectethmoid; fm, foramen magnum; fr, frontal bone, jug, jugal; mnd, mandible; na, nares; oc, occipital condyle; iof, interorbital fonticulus; porb, postorbital process; prx, premaxilla; qdr, quadrate.

The frontals are largely intact and show dorsally projected supraorbital crests. This condition is also present in *N. grandei*, *Jacana*, *Burhinus*, and *Charadrius*. In dorsal aspect the interorbital area is deeply furrowed but without foramina. In lateral aspect, a well-developed ectethmoid is visible and appears to have been ankylosed to the lacrimal as in *Jacana*, *Charadrius*, *Haematopus* and *Burhinus*; however, no lacrimal is preserved. This condition is present in many Charadriiformes and some Gruiformes, such as *Psophia and Himantornis*. *Parts* of the interorbital septum are visible within the orbital cavity; the sclerotic rings are lost. Straight, thin jugals can be observed on both sides of the skull.

The postorbital process is truncate and dorsal to the dorsal apex of the nuchal crest, like in Scandiavis. The latter condition is present in *Charadrius*, *Haematopus*, and *Psophia* but not in *Burhinus*, *Jacana*, *Heliornis*, *Sarothrura*, or *Himantornis*. The zygomatic process is truncate, aciminate at the terminus, and rostrally directed. The occipital region is well preserved and visible in the CT data (Figure 2). The fonticuli occipitalis are not present. This is consistent with the condition in *Charadrius*, *Burhinus*, *Jacana*, *Haematopus*, *Psophia*, *Himantornis*, *Sarothrura*, and *Heliornis*. The foramen magnum is round and the occipital condyle is reniform. The occipital condyle is subequal in rostrocaudal position to the paroccipital processes.

Both quadrates are well preserved. The capitula of the quadrate are moderately spaced and separated by a notch. The medial otic capitulum is deflected caudally, as in Charadriiformes. The capitula of the quadrate are moderately spaced and separated by a notch. The capitulum squamosum is rostrocaudally flattened and the capitulum oticum is mediolaterally elongate as in all examined Charadriiformes. The otic process is slender and recurved along the dorsal margin as in *N. grandei*. Prominent lateral and tympanic cristae are present, as in *N. grandei*, *Haematopus*, *Jacana*, *Himantornis*, *Sarothrura*, *Heliornis*, *and Psophia*. An elongate orbital process that is subequal in length to the otic process is present. It appears to have a blunted, sub-rectangular terminus. This is similar to the condition in *Burhinus*, *Jacana*, and *Haematopus*, although their orbital processes exhibit broader width. It is most similar to the condition in *Heliornis*. A caudal condyle is present as in Neoaves. The caudal condyle does not appear to be confluent with the lateral condyle, unlike the condition in *Nahmavis*. The fovea of the cotyla quadratojugalis appears to be shallow and laterally directed as in most Charadriiformes and Gruiformes. Similarly, as in most Charadriiformes and Gruiformes, a rounded processus lateralis present along the caudodorsal rim projects over the fovea quadratojugalis.

The mandible is complete. The symphysis appears to be truncate, less than $1/5$ of the length of the mandible. This is most like the condition in *Jacana*, *Heliornis*, *Sarothrura*, *Himantornis*, and *Psophia*. The dorsal angle of the mandible is prominent, as in *Scandiavis*, and is more prominent than those of *Haematopus*, *Jacana*, *Burhinus*, and *Charadrius* and the examined Gruiformes. The rostral mandibular fenestrae appear to have been elongate and perforate, like the condition in *Haematopus*, *Jacana*, *Charadrius*, *Burhinus*, and *Psophia* and unlike the slit-like condition in *N. grandei* and examined ralloids. Rounded, perforate caudal mandibular fenestrae are also present. The caudal articular area displays neoavian morphology. A hook-like caudolateral process is present on the articular [15,18]. It appears to form a pseudo-retroarticular process like that of *Scandiavis*. *Haematopus* similarly exhibits a pseudo-retroarticular process, although it is more ventrally directed.

### 3.2.2. Endocranial Anatomy

The reconstructed endocast includes the forebrain, midbrain, hindbrain, and osseous labyrinths as well as some cranial nerves and the arrangement of the carotid artery, but excludes full reconstruction of the olfactory bulbs and pathways of the optic nerve. As such, preservation of NHMD 625345 allowed reconstruction of almost the entire endocranial cast (Figures 3 and 4). Excluding cranial nerves, carotid artery and osseous labyrinths, the total endocranial volume is 1.09 mL. However, this should be considered a minimum value since the optic nerve and the olfactory bulbs were incompletely reconstructed. Unless otherwise stated, anatomical comparisons with extant charadriiform birds refer to brains and endocasts illustrated by Stingelin [52] and Smith and Clark [56].

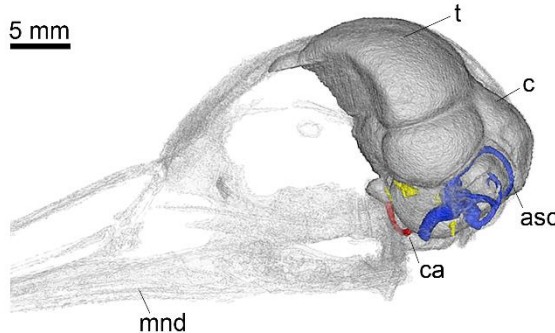

**Figure 3.** Visualization of the skull of *Scandiavis mikkelseni* (NHMD 625345) rendered transparent to show the position of the virtual brain endocast (grey) together with the osseous labyrinth (blue), carotid artery (red), and cranial nerves (yellow). Anatomical abbreviations: asc, anterior semicircular canal; c, cerebellum; ca, carotid artery; mnd, mandible; t, telencephalon.

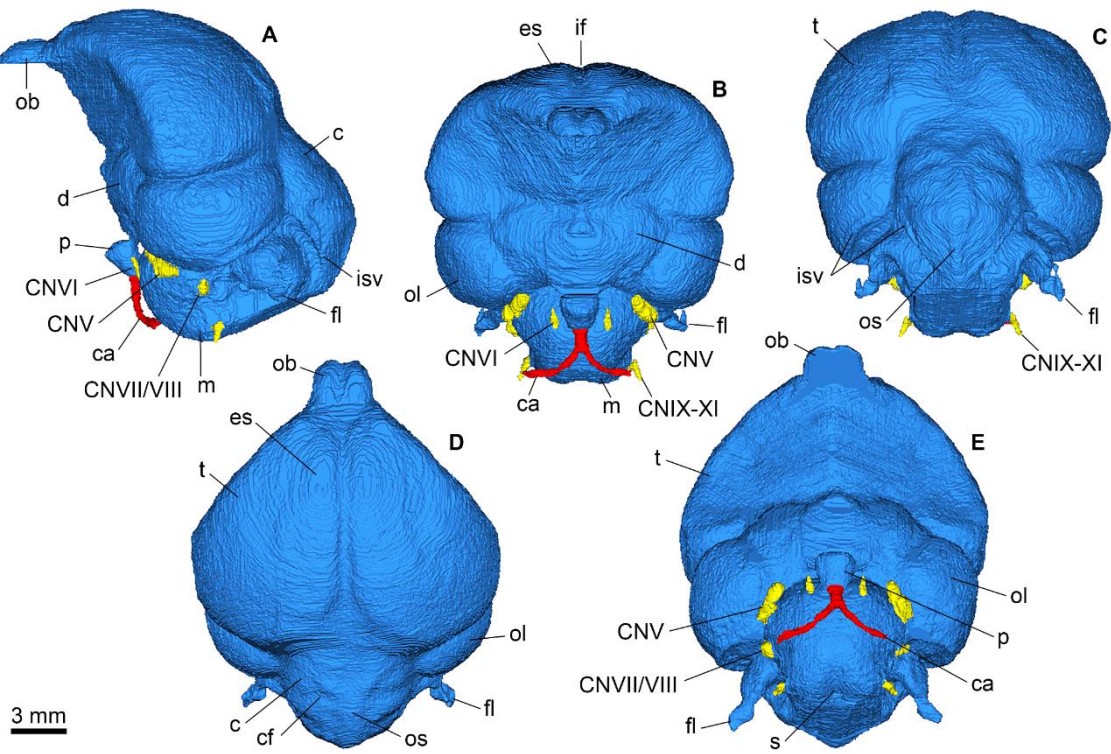

**Figure 4.** Segmented endocast of the cranial cavity of *Scandiavis mikkelseni* (NHMD 625345) in (**A**) left lateral, (**B**) rostral, (**C**) caudal, (**D**) dorsal, and (**E**) ventral aspect. Anatomical abbreviations: c, cerebellum; ca, carotid artery; CNV, trigeminal nerve; CNVI, abducens nerve; CNVII, facial nerve; CNVIII, vestibulocochlear nerve; CNIX, glossopharyngeal nerve; CNX, vagus nerve; CNXI, accessory nerve; d, diencephalon; es, eminentia sagittalis; fl, (cerebellar) flocculus; if, interhemispheric fissure; isv, impression of the semicircular vein; m, medulla oblongata; ob, olfactory bulb; ol, optic lobe; os, occipital sinus; p, pituitary gland; s, sulcus; t, telencephalon.

In *Scandiavis*, the telencephalon is expanded dorsolaterally as in modern birds, and largely stacked on top of the mesencephalon and rhombencephalon in lateral view. The relative dimensions and positions of brain regions, including the olfactory lobes, are highly congruent with certain charadriiform taxa such as *Pluvialis* and *Charadrius*. The brain axis is relatively vertically oriented, but the position of the horizontal semicircular canal of the osseous labyrinth indicates a more ventrally tilted in vivo head posture in *Scandiavis* than in *Charadrius*. In dorsal view, the two telencephalic hemispheres are anteriorly tapered and form a spade-like structure as in extant charadriiforms (Figure 4D). In caudal view, the

telencephalon shows a similar outline to one of the relatively complete natural endocast from the early Eocene of the German Baltic coast figured by Hoch [29]. However, the optic lobes appear relatively smaller in that specimen, and the cerebellum looks more rounded and has more distinct foliation.

*Scandiavis* also shows the presence of eminentia sagittalis or wulst; an expansion of the dorsal surface of the telencephalic hemispheres found in extant birds. Based on its position on the dorsal telencephalon, Stingelin [52] described two different morphotypes. Type A is rostrally positioned, and is found in several of the previously described Eocene taxa (*Odontopteryx*, *Halcyornis*, *Numenius*, and *Prophaethon*; [30,31]). In contrast, Type B is more caudally positioned, but also includes intermediate forms where the eminentia sagittalis is centrally located on the telencephalic hemisphere. In *Scandiavis*, it is relatively weakly developed dorsally (Figure 4B), but broad and rostrocaudally extended without contacting the olfactory bulb nor the cerebellum (Figure 4D). As such, *Scandiavis* is best described as possessing an intermediate Type B development. The telencephalon has a well-defined interhemispheric fissure, stretching from a rostral position to just dorsally to the contact telencephalon-cerebellum. In extant charadriiforms, the wulst shows considerable variation [56]. In dorsal projection, the wulst of *Scandiavis* resembles both that of aquatic foraging taxa, including alcids, and that of terrestrially foraging taxa, such as *Charadrius* and *Stiltia*. However, the rostrocaudal development of the wulst present in *Scandiavis* can only be found in the latter two. Its lateral and dorsal expansion is somewhat similar to *Prophaethon*, but its morphology is otherwise different from that of other Paleogene taxa for which wulst development is available [28,30–32,36,41,59].

In *Scandiavis*, the olfactory bulbs have left an impression on the internal surface of the frontals, but their ventral and lateral margins are not defined by bone, and the segmentation was thus stopped at the lower limits of the ossified margins in accordance with Balanoff et al. [24]. The olfactory bulbs are positioned at the rostralmost apex of the telencephalon without clear bifurcation and demarcated from the telencephalon by a depression (Figure 4A). The dorsal surface displays a shallow inter-lobe sulcus. The relative size of the olfactory bulbs in birds is known to correlate to olfactory capabilities [34,72,73]. Overall, the olfactory bulbs of *Scandiavis* are small compared to some extant avian taxa as well as fossil birds such as *Halcyornis*. Nonetheless, the relative proportions are similar to, e.g., *Pluvianus* and *Charadrius*, and well within the size range of living birds. The lack of clear outlines in this area means the extent of the olfactory nerve (CNI) could not be determined.

The opening for the optic nerve (CNII) consists of a single central relatively wide foramen. As such, it conforms to the Type 2 optic nerve exit sensu Hall et al. [74]. This opening reflects the position of the optic chiasm rather than the size of the optic nerve, and is therefore considered unreliable for estimating the size of the nerve bundle. As a result, the optic nerves were not possible to determine. This type of nerve exit is present in different orders of birds [74]. However, a large optic tract is typical for Charadriiformes [56], and most families within this clade are characterized by a Type 2 optic foramen [74].

The endocast reveals no evidence of a pineal organ, indicating that it did not leave an impression on the skull roof (in similarity with extant birds). The pituitary fossa is readily distinguishable, and the pituitary gland is slightly cone-shaped (Figure 4B). The bony tunnels that housed the carotid artery are preserved. The two arteries merge and form a longitudinal vessel before entering the pituitary gland fossa as a single artery. As such, it conforms to the type I described by Baumel and Gerchman [75]. Ventrally, they bifurcate into two narrow tunnels that curve gently and extend caudolaterally (Figure 4E).

The optic lobes of *Scandiavis* are of considerable size in relation to the telencephalon, and their relative size appears to exceed that of most Charadriiformes and other Paleogene taxa such as *Odontopteryx*, *Prophaethon*, and *Halcyornis*. In lateral view, the optic lobes are globular and slightly lunate. The apparent contact between the optic lobe and the cerebellum is wide, similarly to *Charadrius* and *Stiltia*, whereas the caudal portion of the optic lobe is tapered in most other charadriiform taxa. The optic lobes are visible in dorsal

view in *Scandiavis* as well as basal taxa such as *Charadrius, Pluvianus, Pluvialis, Vanellus,* and *Burhinus*. This character is notably missing from other Charadriiformes. The semicircular vein is impressed on the caudo-ventral surface of the mesencephalon and extends onto the lateral surface of the cerebellum (Figure 4C).

In lateral view, the medulla is globe-shaped and extends along the rostral half of the optic lobes. It bears a defined ventral sulcus and displays clear depressions on the lateral surfaces where the cochleae are positioned. The trigeminal nerve (CNV) is easily distinguished (Figure 4B). Although the three branches cannot be distinguished with certainty, it appears that at least $V_1$ is clearly delimited. The abducens nerve (CNVI) is small and narrow, exiting from the rostralmost end of the medulla (Figure 4B). The facial nerve (CNVII) and the vestibulocochlear nerve (CNVIII) are difficult to distinguish in this dataset, but one or both appear to be represented by a relatively small protrusion on the lateral medulla (Figure 4A). The glossopharyngeal (CNIX), vagus (CNX), and accessory (CNXI) nerves do not appear to differentiate before leaving the skull. In *Scandiavis*, the cerebellum is oval-shaped in caudal view, but tapers caudally (Figure 4C). It bears at least three distinct folia along with an impression of an occipital sinus, which is distinguishable primarily in the caudal region. The cerebellum is heavily restricted dorsally in lateral view and shows only very limited contact with the caudal margin of the telencephalon (Figure 4A), similar to *Vanellus, Pluvialis,* and *Pluvianus*. The cerebellum is significantly more expanded in most other charadriiforms and in many other birds, including *Fulica* and *Balearica*, the only gruiform taxa for which published brain morphology were found [33,50,55]. The impression of the semicircular vein (present on the caudo-ventral surface of the optic lobe) extends to the cerebellum, where it can be traced ventrally along the lateral surface (Figure 4A).

The flocculus is relatively large and projects from the lateral surface of the cerebellum through the arch of the anterior semicircular canal. The base exhibits some torsion, from where it is directed caudolaterally before tapering distally. Within Charadriiformes, this elongated morphology is similar to those of wing-propelled diving Alcinae and bears similarities to *Larus philadelphia* [33], and is significantly different from the reduced and truncated state in the terrestrially foraging *Charadrius* and *Stiltia*. In terms of relative mediolateral length, shape of the base, degree of tapering distally, and size, it closely resembles that of the more aquatic foraging gruiform *Fulica americana*. The reconstruction of the flocculus of *Scandiavis* shows that it may have fenestration distally, but the CT data was difficult to interpret in this region and the state of this character therefore remains uncertain.

In the endocast of the osseous labyrinth of *Scandiavis*, all semicircular canals as well as the cochlear duct are intact and preserved in detail on both sides of the skull. The structure is positioned largely ventral to the optic lobe (Figure 3). All canals are relatively long and narrow with well-defined ampullae (Figure 5). As in living birds, the anterior semicircular canal is more expanded than the others, and in addition distinctly medially angled. The cochlear duct is relatively long and somewhat arched. The distal tip is swollen as in *Stiltia* and *Charadrius* rather than tapered as in all other charadriiforms. Overall, the labyrinth endocast appears nearly identical to the two aforementioned taxa, but with a less pronounced swelling of the cochlear tip than in *Charadrius*. In comparison with previously described labyrinths of Eocene taxa (*Odontopteryx, Prophaethon* and *Halcyornis*), the semicircular canals are both relatively long and thin. The overall morphology of the inner ear is also significantly different from those observed in fossil stem penguins [32,36,59].

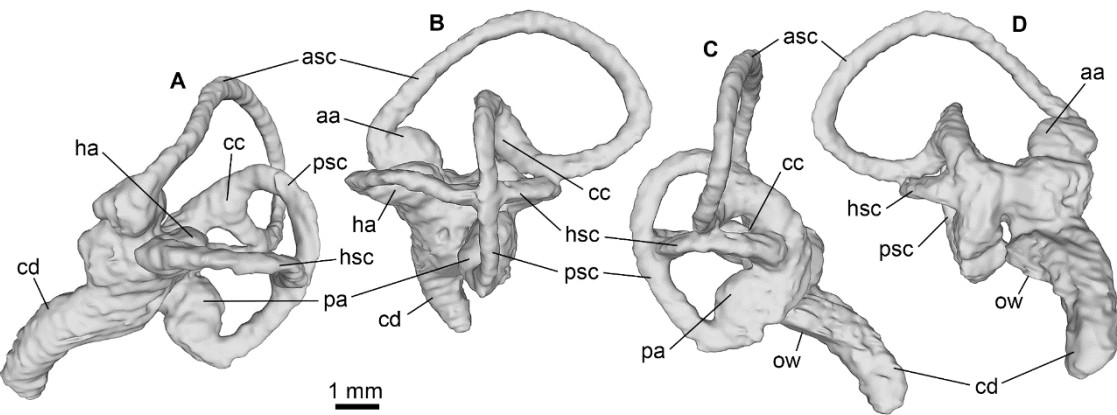

**Figure 5.** Virtual reconstruction of the left osseous labyrinth of *Scandiavis mikkelseni* (NHMD 625345) in (**A**) rostrolateral, (**B**) caudal, (**C**) caudomedial, and (**D**) medial views. Anatomical abbreviations: aa, anterior ampulla; asc, anterior semicircular canal; cc, common crus; cd, cochlear duct; ha, horizontal ampulla; hsc, horizontal semicircular canal; ow, oval window; pa, posterior ampulla; psc, semicircular canal.

### 3.3. Phylogenetic Results

Unconstrained parsimony analysis resulted in 237 most parsimonious trees (MPTs) of 5237 steps. This analysis recovered a paraphyletic Gruiformes with respect to a monophyletic Charadriiformes, with *Nahmavis grandei* and *Scandiavis mikkelseni* being collapsed into a polytomy of included Charadriiformes (CI = 0.168, RI = 0.472, RC = 0.079, HI = 0.832).

Analysis applying a backbone constraint representing major subclade relationships of Kimball et al. [66] recovered 56 MPTs of 5335 steps (Figure 6; CI = 0.165, RI = 0.460, RC = 0.076, HI = 0.835) and an *S. mikkelseni* + (*N. grandei* + *Turnix nigricollis*) clade as the sister group of all included charadriiform taxa. Synapomorphies recovered for critical groups across all constrained analyses are displayed in Table A1. *Pellornis mikkelseni* Bertelli et al. [14] was recovered as a messelornithid across all analyses, consistent with prior placements [14,15]. Messelornithidae (*P. mikkelseni* + *Messelornis cristata* Hesse [76]) was placed as the sister taxon of Ralloidea under the Kimball et al. [66] and Reddy et al. [65] constraints. These results are consistent with those of Musser and Clarke [18] with the exception of the Prum et al. [64] constraints causing collapse of Messelornithidae within a polytomy containing extant Ralloidea and *Songzia acutunguis*.

Analyses using a backbone constraint based on the major clade relationships of Prum et al. [64] recovered 24 MPTs of 5331 steps (Figure 7A; CI = 0.165, RI = 0.461, RC = 0.076, HI = 0.835), and employing a Reddy et al. [65] constraint yielded 56 MPTs of 5345 steps (Figure 7B; CI = 0.165, RI = 0.459, RC = 0.076, HI = 0.835). These analyses did not recover Charadriiformes and Gruiformes as sister groups as in the Kimball et al. [66] constrained analyses. Within the results constrained using Reddy et al. [65], *N. grandei* is placed as the sister taxon of a *S. mikkelseni* + Charadriiformes as a part of Pan-Charadriiformes. Analyses employing the Prum et al. [64] constraint resulted in an *N. grandei* + *S. mikkelseni* as sister taxon of *Pluvianus aegyptius*. This group is then collapsed into a polytomy of Charadriiformes. Bootstrap support for placement of extinct taxa was less than 50% across all analyses, with the exception of Messelornithidae earning a 94% or higher bootstrap score in each result.

Placement of *Salmila robusta* Mayr [46] and both Eocene humeri remain unchanged from the results of Musser and Clarke [18] with the exception of their placement in the tree resulting from analyses employing the Reddy et al. [65] constraint. In Musser and Clarke [18], *Salmila* is the sister-taxon of *Eurypyga helias* + *Rhynochetos jubatus*, whereas it is placed the sister-taxon of *Opisthocomus hoazin* + *Leptosomus discolor* in our results. Our results place IGM 100/1435 as the sister-taxon of *Chionis alba*, whereas it is collapsed into a polytomy of Charadriiformes in Musser and Clarke [18].

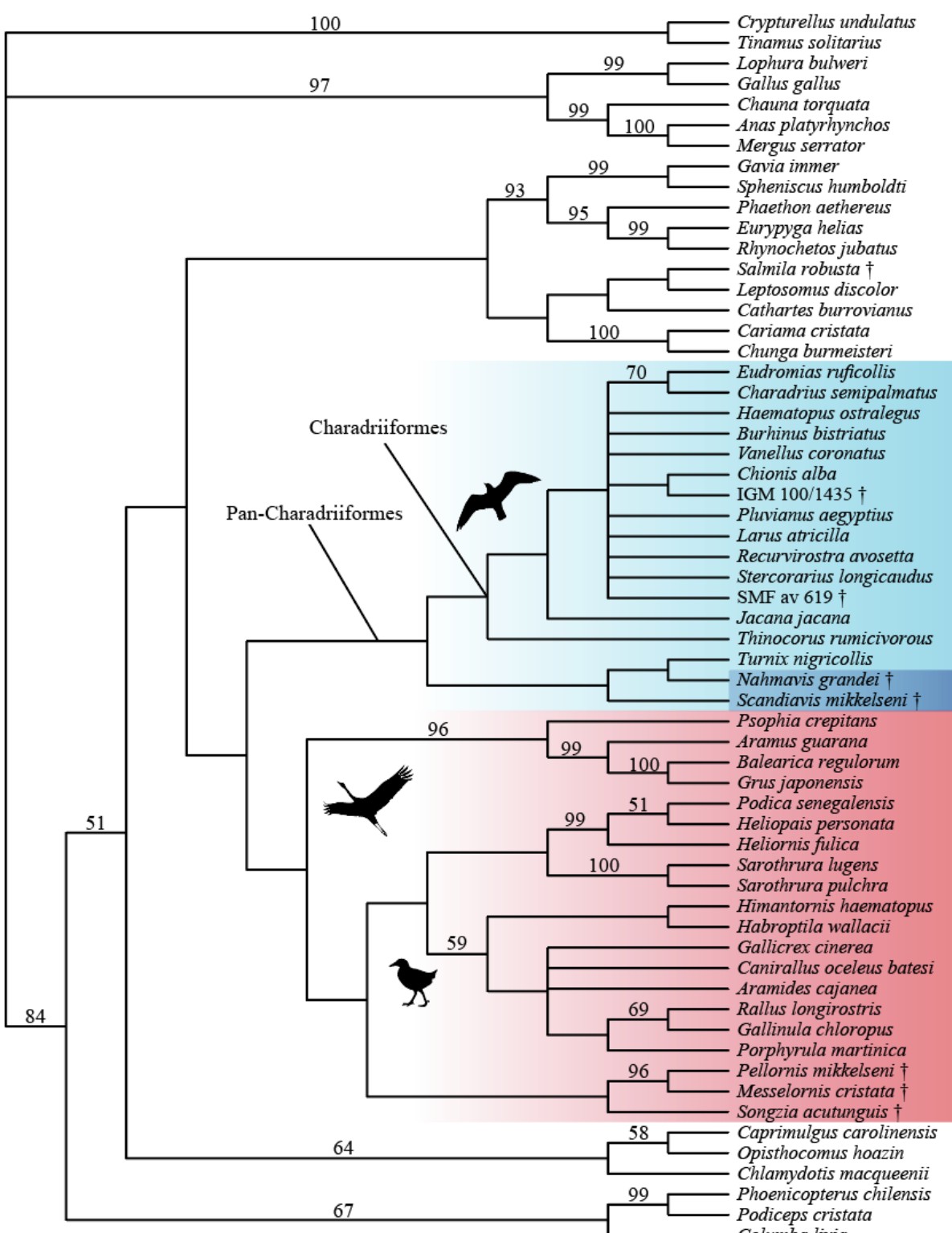

**Figure 6.** Strict consensus tree of 5335 steps recovered from analyses using the most recent Kimball et al. [66] tree as a molecular backbone constraint (CI = 0.168, RI = 0.472, RC = 0.079, HI = 0.832). Charadriiform, ralloid, and gruoid silhouettes are placed at the crowns of their respective clades. All extinct taxa are denoted with daggers. Gruiformes are highlighted in red, Charadriiformes are highlighted in blue, and *Scandiavis mikkelseni* and *Nahmavis grandei* are highlighted in dark blue. Bootstrap support values greater than 50% are denoted above branches.

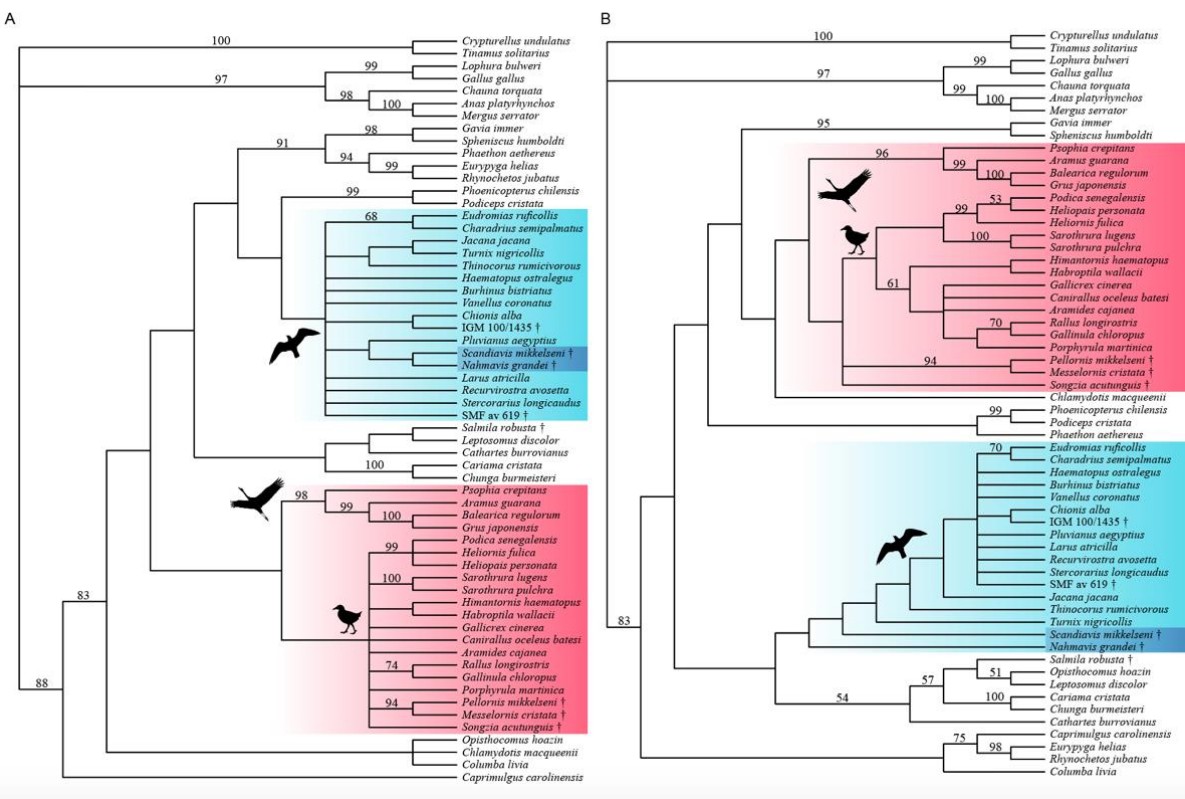

**Figure 7.** Strict consensus trees recovered from heuristic parsimony analyses using (**A**) Prum et al. [64] (24 MPTs, 5331 steps, CI = 0.165, RI = 0.461, RC = 0.076, HI = 0.835) and (**B**) Reddy et al. [65] (56 MPTs, 5345 steps, CI = 0.165, RI = 0.459, RC = 0.076, HI = 0.835) trees as molecular backbone constraints. Charadriiform, gruoid, and ralloid silhouettes are placed at the crowns of their respective clades. Charadriiformes are highlighted in blue and Gruiformes are highlighted in red, with *Scandiavis mikkelseni* and *Nahmavis grandei* being highlighted in dark blue. All extinct taxa are denoted with daggers. Bootstrap values greater than 50% are denoted above branches.

Nine unambiguous and five ambiguous optimized synapomorphies with CI < 1.0 support placement of *S. mikkelseni* with Charadriiformes using the Kimball et al. [66] constraint (Figure 6). The rostral apex of the premaxilla is not dorsally inflated (character 4:state 1). This is similar to the condition in most included Charadriiformes; it is inflated in most included Gruiformes (4:2). The apex of the postorbital process is oriented rostroventrally as in most Charadriiformes (55:1), whereas it is angled ventrally in most Gruiformes (55:2). The supraorbital process of the lacrimal is tapered toward the caudal apex as in most included Charadriiformes (68:1). It is broad in most included Gruiformes (68:2). The ectethmoid is ankylosed to the lacrimal (86:1), a feature that is more common in Charadriiformes than in Gruiformes (typically unankylosed, 86:2). The fonticulus orbitocranialis is caudally extensive and continues along the coronal plane of the squamosal region (91:2). This feature is common in both Gruiformes and Charadriiformes. The lateral condyle of the quadrate terminates well ventral to the caudal condyle as in most Charadriiformes (206:2). Gruiformes typically exhibit a lateral condyle that is only slightly ventrally positioned with respect to the caudal condyle (206:1). The cervical vertebrae are not heterogeneously elongate, as in most Charadriiformes (253:2); most Gruiformes present heterogeneous vertebrae that are relatively elongated (253:1). The torus dorsalis of section II of the cervical vertebrae is concave as in most included Charadriiformes (261:2), whereas it is convex in most included Gruiformes (261:1). The caudalmost presacral vertebrae exhibit deep lateral excavations which are also present in all included Charadriiformes (279:2). These excavations are not present in included Gruiformes (279:1). The spina interna rostri of the sternum is present as in most included Charadriiformes (299:1); it is lost in all included Gruiformes (299:2). The iliac blades and synsacrum of the pelvis are unfused as in almost

all included Charadriiformes (498:2), whereas those of included Gruiformes are fused (498:1). Pedal digit I: phalanx 1 is about half the length of III:1 as in most included Charadriiformes (691:2). Most included Gruiformes present a I:1 that is about half the length of III:1 (691:1). Three common crus of the semicircular canals are present as in *Charadrius*, *Larus*, *Haematopus*, and *Stercorarius* (719:1). *Grus* conversely only exhibits two common crus (719:0). The cochlear curvature of the endosseous labyrinth in lateral aspect is curved as in *Charadrius*, *Larus*, and *Stercorarius* (720:0). It is straight in *Grus* (720:1). As in Musser and Clarke [18], placement of *Turnix nigricollis* as the sister-taxon of *N. grandei* is not consistent with recent phylogenies, which recover *Turnix* as the sister-taxon of a group containing *Glareola*, *Larus*, and *Dromas* [64–66]. Incongruous placement of this taxon in our results is likely due to a lack of constraining taxon relationships within major subclades and a need for more taxon and character sampling. *Turnix* is also known to be problematic in morphological analyses; Livezey [77] similarly recovered *Turnix* as the sister-group to all other Charadriiformes, and Mayr [78] recovered Turnicidae as either within a polytomy in Charadriiformes or as the sister-taxon of a clade containing Jacanidae, Scolopacidae, Rostratulidae, Thinocoridae, and Pedionomidae. Synapomorphies for additional constraint analyses can be found in the Appendix A.

## 4. Discussion

Recovery of additional material and new phylogenetic analysis of *Scandiavis mikkelseni* more robustly establishes both *S. mikkelseni* and *Nahmavis grandei* as charadriiform, and most consistently places them as stem-charadriiform taxa. This is critical to reconstructing phylogenetic relationships of Paleogene fossils, as previous studies produced conflicting results as to whether either taxon was charadriiform or gruiform [17,18]. While it remains unclear whether *S. mikkelseni* or *N. grandei* is more basal, both taxa come from the early Eocene and may represent important divergence date calibration fossils for Pan-Charadriiformes.

The relative dimension of a brain region in vertebrates is proportional to the relative importance of that particular region in a species [79,80], allowing inferences to be made about the neural development and sensory capabilities of extinct taxa. The endocast of *Scandiavis* thus provides valuable new insights into avian neural architecture in the early Paleogene. It shares general features with modern birds, such as laterally expanded mesencephalic and telencephalic lobes, and a more vertically flexed brain axis as in many extant birds [81]. Comparisons with extant and Paleogene taxa reveal that this endocast shows numerous similarities with extant basal crown charadriiforms, and clearly possesses the apomorphies that have been proposed to characterize the endocranium of charadriiform birds. This includes more dorsally positioned olfactory bulbs, an anteriorly tapered telencephalon, relatively wider cerebella, and a larger optic tract [56].

The relative dimensions of the brain regions of *Scandiavis* and their positions relative to each other, including the olfactory lobes, are congruent with taxa such as *Pluvialis*, *Vanellus* and *Charadrius*. In particular, there is a striking resemblance between the endocasts of *Scandiavis* and *Charadrius vociferus,* which share all identical character states relating to both the brain and the osseous labyrinth with the exception of the floccular region (characters 711–713). The flocculus in *Scandiavis* does not exhibit the reduced state seen in *Charadrius*. Instead, it is relatively large and elongated, resembling that of *Fulica*. The flocculus has been proposed to play a pivotal role in gaze stabilization and swift head movements, facilitating acrobatic maneuverability during flight [19,82]. However, floccular volumes have also been found to be unreliable for inferences on flight ability and ecology in extant taxa [83,84]. In contrast to *Fulica*, *Scandiavis* may have fenestration distally (character 712), but this was scored as a missing character due to difficulties interpreting the CT data in this region. This may indicate that a large flocculus may be ancestral for Charadriiformes + Gruiformes if this sister-relationship is correct. Intriguingly, finfoots are some of the most basal gruiforms and have a similar ecology to that of *Fulica*. The endocast of *Scandiavis* additionally exhibits one endocranial character state that supports an 'aquatic charadriiform clade' including the

Pan-Alcidae, Stercorariidae, Sternidae, Laridae, and Rynchopidae in Smith and Clarke [56]: a telencephalon with < 50% of its caudal margin in contact with the cerebellum. However, this character state is also present in several basal charadriifom taxa including *Charadrius, Burhinus, Pluvianus* and *Vanellus.*

The eminentia sagittalis is a dorsal telencephalic expansion unique to the brains of birds. It is commonly linked to visual field processing and tends to be well-developed in species that rely heavily on vision [85,86], but is also generally involved in somatosensory perception [85]. The eminentia sagittalis is found in all extant birds, but morphology and development greatly vary. This structure has been observed in several Paleogene birds (e.g., [30,31,59,72]) and recently also in a Cretaceous specimen [87]. Although *Scandiavis* possessed a relatively weakly developed eminentia sagittalis, it is within the range of development observed among extant birds. The dorsal expansion appears at the lower end of the scale, but still mirrors that of *Charadrius vociferus* and closely related taxa. The overall morphology is different from that of other Paleogene taxa and thus adds to a growing range of morphologies of the eminentia sagittalis within Aves by the early Eocene, suggesting that the diversification of this structure may have been initiated much earlier [31]. In addition, *Scandiavis* possessed large optic lobes. This region receives and processes much of the retinal input [88], and may thus suggest a need for optical input.

The morphology of the osseous labyrinth has been hypothesized to reflect flight style in modern birds, and is therefore commonly targeted in paleontological studies for generating ecological interpretations. Relatively long and thin semicircular canals often occur in taxa that engage in 'acrobatic flight' (e.g., *Larus* and *Columba*) whereas shorter and broader canals (e.g., *Anas* and *Gallus*) are normally associated with straight-line flight [30,81,89,90]. Additionally, the ampullae, cross-sectional shape, and angles between the semicircular canals have been used to infer flying ability [30,81,89,90]. The relatively long and thin semicircular canals of *Scandiavis* as well as the close resemblance to terrestrially foraging charariiform birds may thus suggest locomotory abilities and maneuverability similar to these taxa. The ampullae are also relatively well-developed and the anterior semicircular canal is distinctly inclined medially, all conditions seen in birds regarded as good fliers [81,89]. However, potential relationships between labyrinth morphology and locomotory modes in birds were recently questioned. Benson et al. [53] suggested that avian labyrinth shape is determined more by spatial constraints in the braincase than flying style. This in turn was questioned for not adequately capturing the relevant patterns of locomotion [91], and at present relationships remain unclear [92].

Recovery of *S. mikkelseni* and *N. grandei* as stem-Charadriiformes presents a similar biogeographic pattern as that of Paleogene Ralloidea (Gruiformes: rails, finfoots and flufftails) [14,15,18,93], with Eocene representatives being located in Europe and North America. Messelornithidae are currently the only robustly placed stem gruiform taxa and comprise *Pellornis mikkelseni* of the earliest Eocene Fur Formation in Denmark, the *Messelornis nearctica* [94] in the Eocene Green River Formation of North America [94–97], *Messelornis cristata* [76] of the Eocene Messel Formation of Germany, *Messelornis russeli* [98] of the Paleocene of France, and *Itardiornis hessae* [98] of the late Eocene to early Oligocene Quercy fissure fillings of France. Such fossil evidence suggests two possible biogeographic hypotheses if Aves originated in West Gondwana: the North American Gateway hypothesis [99] or a Laurasian Gateway hypothesis. If Gruiformes and Charadriiformes originated in West Gondwana and were present in South America by the end of the middle Paleocene (~59.2 Ma), these clades could have dispersed through and diversified in North America coincident with the first appearances of Laurasian metatherian and placental mammals in South America [100,101] during the Paleocene to early middle Eocene (the North American Gateway hypothesis [99]). These avifaunas would then have spread to Europe by the end of the Paleocene and earliest Eocene ~56 to 53 Ma, reaching Europe via a North Atlantic corridor with land connections along northeastern North America, Greenland and Europe [102,103]. Sea-floor spreading between northeastern Greenland and Europe would have created an ocean barrier around 53 to 52 Ma [102,104]. Conversely, Gruiformes and

Charadriiformes could have dispersed through North America and crossed the Bering land bridge, which would have been emergent during much of the Tertiary [105,106]. Eocene–Oligocene fossils of stem-Gruoidea (Gruiformes: trumpeters, cranes and limp-kins; [107]) in North America and Asia and a putative ralloid (*Songzia* [48,108]) in China may support this hypothesis for Gruiformes. Charadriiform fossils from Asia comprise the pan-charadriiform humeri from the earliest Eocene of Mongolia [69] and *Jiliniornis huadianensis* from the middle Eocene of China [109–111]. At the same time, presence of these fossils in Asia could still be consistent with the North American Gateway hypothesis with these avifaunas expanding first into Europe and then Asia [99], especially as both charadriiform fossils from Asia were recovered within crown Charadriiformes [18,109] and a putative specimen of *Songzia* has been recovered from the Paleocene of France [112]. If these avifaunas did not originate in Gondwana, fossil evidence may be indicative of stem clades originating in Eurasia and rapidly expanding into North America by the end of the early Eocene [18]. More fossil evidence from Gondwana and Asia is ultimately necessary to more robustly support one of these hypotheses or a non-Gondwanan based hypothesis.

In conclusion, *Scandiavis mikkelseni* is an important Paleogene fossil for understanding evolutionary relationships, identifying divergence dates and biogeography, and better understanding the evolution of charadriiform and possibly gruiform brain structure. The brain of *Scandiavis* is largely similar to those of basal extant charadriiforms, and it probably also possessed similar sensory capabilities. This endocast demonstrates that by the early Eocene, avian brain development was close to that of extant birds. Confirmation of *Scandiavis* and *Nahmavis grandei* as charadriiform presents a biogeographic pattern similar to that within the fossil record of Gruiformes [14,15,18,93]. This evidence is consistent with several biogeographic hypotheses, and further evidence supporting the sister-relationship of these two important clades is thus necessary to better understand their biogeographic histories.

**Author Contributions:** Conceptualization, M.H., G.M. and S.A.H.; Data curation, M.H., G.M. and S.A.H.; Formal analysis, G.M.; Funding acquisition, M.H. and G.M.; Investigation, M.H. and G.M.; Methodology, M.H., G.M. and S.A.H.; Project administration, M.H., G.M. and S.A.H.; Resources, M.H., G.M. and S.A.H.; Software, M.H. and G.M.; Supervision, J.A.C.; Validation, M.H., G.M. and J.A.C.; Visualization, M.H. and S.A.H.; Writing—original draft, M.H. and G.M.; Writing—review and editing, M.H., G.M., S.A.H., and J.A.C. All authors have read and agreed to the published version of the manuscript.

**Funding:** Financial support for this project was provided by a National Science Foundation GRFP award (to G.M., Grant number DGE-16-4486), the Jackson School of Geosciences (G.M. and J.A.C.), and a grant (to M.H.) from The Royal Physiographic Society in Lund.

**Institutional Review Board Statement:** Not applicable.

**Informed Consent Statement:** Not applicable.

**Data Availability Statement:** The phylogenetic and CT datasets along with stl files presented in this study can be found on Morphobank at the following link: http://morphobank.org/permalink/?P4049 (accessed on 25 November 2021).

**Acknowledgments:** We thank Bent Lindow (NHMD) and Bo Schultz (FUM) for access to NHMD 625345, invaluable discussion and feedback, and for providing stratigraphic information regarding the specimen. We would also like to thank Randolph De La Garza for assisting with photographing the fossil, and Johan Lindgren and Christopher Torres for constructive comments and discussions. We additionally thank the editor and anonymous reviewers for thoughtful feedback.

**Conflicts of Interest:** The authors declare that the research was conducted in the absence of any commercial or financial relationships that could be construed as a potential conflict of interest.

## Appendix A

Character Descriptions

State (0) is reserved for absence. Citations are provided where significant overlap with a previously created character has occurred and/or where the character has been

modified based on assessment of previously created characters. Citations of previously created characters are not meant to represent a comprehensive list of character overlap, and largely comprise the characters (and citations therein) of Musser and Cracraft [44], due to our building on this dataset. Characters are anglicized as much as possible.

*Skull: Rostrum*

1.   Premaxilla, dorsal aspect of rostral apex, distinct pair of foramina:  absent (0); present (1); present, with associated rostral sulcus (2). Musser and Cracraft [44], character 1.

2. Premaxilla, rostral apex, curvature: slightly curved ventrally (1); markedly curved inferiorly, hooked (2); lacking inferior curvature so that bill appears to be flat or recurved superiorly (3). Musser and Cracraft [44], character 4.

3. Premaxilla, rostral apex, exceptionally mediolaterally wide: no (1); yes (2).

4. Premaxilla, processus frontalis, dorsally inflated: no (1); yes (2).

5.  Premaxilla, processus frontalis, caudal terminus and associated suture between premaxilla and nasals: triangular (1); rounded (2). Noncomparable where suture obscured by nasal and/or frontal.

6. Rostrum, curvature, not including anterior terminus of premaxilla: lacking inferior curvature so that bill appears to be flat or recurved superiorly (1); curved inferiorly (2); curved extremely inferiorly, distal half of rostrum essentially perpendicular to proximal half (3); distal half of rostrum recurved dorsally (4).

7.  Rostrum, dorsoventrally compressed so that maxillary process of nasal bar and ventral bar of rostrum almost at same level as processus frontalis of premaxilla: no (1); yes (2). Musser and Cracraft [44], character 10; Livezey and Zusi [95], character 280.

8. Rostrum, ventral aspect, tomial crest, caudomedial extent and ventral enclosure of rostrum and consequent shape of ventromedial fenestra: almost completely fused along midline to border of antorbital angle so that rostrum essentially closed by tomial crest ventrally, slit-like fenestra along midline may be visible in ventral aspect (1); fused along about 1/2 of midline or less near rostral apex so that pair of cylindrical cavities at apex visible but separated toward antorbital fenestra, creating "leaf-shaped" fenestra that is wider near antorbital fenestra (2); ventral enclosure of rostrum lost completely (3); like state 2 but over 1/2 of rostral portion is enclosed, pair of concavities concealed by tomial crest (4); completely closed ventrally by inflated bone (5).

9. Rostrum, tomial crest, lateral margins, ventral extension: essentially none, tomial crest at same level of ventral face of rostrum (1); marked caudal to apex (2); marked at rostral apex (3). Not applicable if rostrum not closed ventrally by tomial crest.

10.  Nares, caudal margin: acuminate (1); rounded (2).  Musser and Cracraft [44], character 6.

11. Nares, length as compared to total length of rostrum: less than 1/2 of rostrum (1); 1/2 of rostrum (2); over 1/2 length of rostrum (3).

12.  Nares, nasal sulcus beginning from rostral limit of nares: present, complete to terminal end of rostrum (1); lost or truncated (2). Musser and Cracraft [44], character 9; Livezey and Zusi [95], character 272.

13. Nares, rostral margin: rounded (1); acuminate (2).

14. Nares, internal bony septum ankylosed to ventral face of dorsal bar along midline: absent (0); present (1).

15.  Nares, caudal terminal end, caudal margin relative to caudal margin of zona flexoria craniofacialis (defined here as the caudal margin of processus frontalis of premaxilla), site: rostral or subequal to rostral margin of zona flexoria craniofacialis, well within margins of flexor site (1); reach caudal margin or extend beyond caudal margin of zona flexoria craniofacialis so that premaxillary and maxillary processes of nasals essentially separated (2). Musser and Cracraft [44], character 13; Livezey and Zusi [95], character 341; Mayr and Clarke [63], character 6.

16. Nares, caudal margin, thickness of dorsal limit of maxillary process of nasals compared to thickness of ventral limit: subequal in rostrocaudal width, narial bar often thin and rod like (1); wider at dorsal end (2); wider at ventral end (3).

17. Nasals, maxillary processes, ventral termini, lack of fusion to ventral rostrum: present (1); lost, fused (2).

18. Nasals, maxillary processes: rostrocaudally flattened (1); mediolaterally flattened (2); both, narial bars mediolaterally flattened rostrally and rostrocaudally flattened caudally (3).

19. Zona flexoria craniofacialis, depth: slight to moderate (1); deep, concave and fissure-like (2).

20. Zona flexoria craniofacialis, mediolateral alignment of caudal margin of processus frontalis of premaxilla and caudal margin of premaxillary (and maxillary, if separated as in schizorhiny) processes of nasal bone: absent, processus frontalis rostral to premaxillary processes of nasal bone (1); present, caudal termini of processes aligned mediolaterally (2); lost, caudal margin of processus frontalis of nasal bone extends further caudally than those of premaxillary processes of nasal bone (3). Not comparable when frontonasal suture indiscernible or where caudal termini of sutures are obscured by frontal bone. Musser and Cracraft [44], character 14; Livezey and Zusi [95], character 603.

21. Nasals and/or frontal, (typically) subtriangular depression just caudal to caudal limit of processus frontalis of premaxilla, status (regardless of furrowing of interorbital area): present (1); lost (2).

22. Nasals and/or frontal, rostral depression (if present), craniocaudal extent: limited within rostral half of orbit (1); extends caudally beyond rostral half of orbit (2). Noncomparable where rostral depression absent.

23. Frontal, dorsal face caudal to zona flexoria craniofacialis, dorsal inflation, status and magnitude: absent (1); present, extreme, extends caudally along frontal bone (2).

24. Frontal, rostrally protruding so that processus frontalis of premaxilla (and sometimes premaxillary processes of nasal bone) is obscured: no (1); yes, small, 1–2 rounded rostral bulb(s) protruding dorsal to proccessus frontalis of nasal bone (2); yes, mediolaterally wide, rostrally protruding lamina (3); yes, extremely dorsally inflated frontal bone protrudes dorsal to proccessus frontalis of nasal bone (4).

25. Rostrum, dorsal face, midline length from the apex of the premaxilla to the zona flexoria craniofacialis relative to that of the cranium: 0.40–2.00 (1); 2.01–3.00 (2). Musser and Cracraft [44], character 2; Livezey and Zusi [95], character 260.

26. Antorbital angle, general shape: subtriangular (1); ovoid, craniocaudally elongate (2).

27. Antorbital angle, angle created by ankylosis of maxillary process of nasal and maxilla: large, essentially 90 degrees (1); small, approximately 45 degrees (2). Noncomparable if antorbital fenestra is ovoid.

28. Mesethmoid, bifurcation and ankylosis to maxillary processes of nasals so that mesethmoid is inflated and visible within caudal portion of nares: no (1); yes (2).

29. Frontonasal suture, elevated dorsally: no (1); yes (2).

30. Orbital margin, salt gland sulcus: absent or non-distinct (0); present (1). Musser and Cracraft [44], character 17; Mayr and Clarke [63], character 25; Livezey [113], character 72.

31. Orbital margin, salt gland sulcus (if present), depth: shallow (1); deep, typically mediolaterally expansive (2). Noncomparable where absent.

32. Interorbital area, midline: convex and not furrowed (1); furrowed along midline, concave (2). Musser and Cracraft [44], character 18; Livezey [113], character 70.

33. Cranium, dorsal aspect, fronto-parietal area, pair of prominent, horn-like protuberances: absent (0); present (1). Musser and Cracraft [44], character 19.

34. Cranium (postorbital braincase), general form: craniocaudally elongate (1); craniocaudally compressed (2). Musser and Cracraft [44], character 16; Livezey and Zusi [95], character 7.

35. Frontoparietal suture, status in adults: present (1); lost (2). Musser and Cracraft [44], character 20; Livezey and Zusi [95], character 213.

36. Maxilla, tomial crest, tomial angle, extension of caudal terminal end as distinct tubercle or short process caudal to jugomaxillary suture and lateral to, and variably free from, arc of jugum: absent (0); present (1).

37. Maxilla, tomial crest, tomial angle, extension of caudal terminal end as distinct tubercle or short process caudal to jugomaxillary suture and lateral to, and variably free from, arc of jugum (if present), shape: triangular (1); subrectangular (2). Noncomparable where absent. Musser and Cracraft [44], character 21; Livezey and Zusi [95], character 408.

*Skull: Cranium*

38. Maxillopalatine processes, fused along midline: yes (1); no (2).

39. Maxillopalatine processes, shape of caudal terminus: acuminate (1); rounded (2).

40. Maxillopalatine processes, orientation of flattening: dorsoventrally flattened (1); mediolaterally flattened (2).

41. Maxillopalatine processes, craniocaudal elongation: present, extreme (1); lost (2); present, moderate (3).

42. Maxillopalatine processes, ventral margin, dorsoventral location relative to palatines: ventral or subequal to palatines (1); dorsal to palatines (2).

43. Maxillopalatine processes, extensive ankylosis to and/or contact with palatines: present (1); lost (2).

44. Maxillopalatine processes, enclosure of lateral (when mediolaterally flattened) or dorsal (when dorsoventrally flattened) face so that inflated, cone-like structure is created: not enclosed (1); partially enclosed (2); completely enclosed (3). Musser and Cracraft [44], character 22; Livezey and Zusi [95], character 420.

45. Maxillojugal pons, ventrolateral fenestra enclosed by caudal strut that is ankylosed to maxillopalatine processes and jugal: not enclosed (1); enclosed by bony projection (2). Musser and Cracraft [44], character 12; Livezey and Zusi [95], character 290.

46. Maxillojugal pons, craniocaudal attachment point of medial margin of strut: within rostral half of maxillopalatine processes (1); within caudal half of maxillopalatine processes (2). Noncomparable where maxillojugal pons absent.

47. Maxillojugal pons, caudal strut (if present), shape: craniocaudally thick, medial margin at least half craniocaudal length of lateral margin (1); tapered, lateral margin being craniocaudally more elongate than medial margin, medial portion of strut typically twisted rostrally (2). Noncomparable where maxillojugal pons absent.

48. Maxillojugal pons, ventrolateral fenestra, shape: small, circular (1); craniocaudally elongate, ovoid (2). Noncomparable where maxillojugal pons absent.

49. Orbital margin, crista-like orbital margin or supraorbital crest (see Musser and Cracraft [44]): absent, orbital margins not laterally extended and are mostly or completely rounded (0); present (1). Musser and Cracraft [44], character 24.

50. Orbital margin, supraorbital crest (if present): full, projecting dorsolaterally with a sharp, crista-like margin, creates a sharp and prominent supraorbital crest that extends from the postorbital process to the lacrimal (1); partial, like state only present along caudal half of orbit (2); crista-like margin prominent and present but projected ventrally/laterally, does not extend dorsal to frontal bone as in states 2 and 3 (3). Noncomparable where absent.

51. Orbital margin, pair of large, circular foramina near zona flexoria craniofacialis, typically within salt gland sulcus: absent (0); present (1).

52. Orbital margin, large pores: absent (0); present (1). Musser and Cracraft [44], character 23.

53. Orbital margin, additional large foramina along orbital margin: absent (0); present (1).

54. Postorbital process, length: truncate (1); extremely elongate, subequal in length to height of quadrate (2). Musser and Cracraft [44], character 26.

55. Postorbital process, orientation of apex: facing rostroventrally (1); facing ventrally (2).

56. Postorbital process, concavity: convex (1); concave along caudal length of process (2). Musser and Cracraft [44], character 27.

57. Interorbital septum, mediolateral thickness: thin, in some places translucent (1); thick, opaque throughout (2). Musser and Cracraft [44], character 28.

58. Lacrimofrontal suture: lost or indistinguishable (1); present (2).

59. Lacrimofrontal suture (if present), craniocaudal extent: incomplete, present in only rostral portion of lacrimal (1); present throughout dorsal margin of orbital process of lacrimal (2). Noncomparable where lacrimofrontal suture is lost or indistinguishable.

60. Lacrimal, site of ankylosis to nasals: ankylosed to both maxillary processes and frontal process of nasals and/or frontal bone (1); ankylosed to maxillary processes of nasals only (2); ankylosed to frontal process of nasals (3).

61. Lacrimals, site of ankylosis to nasals relative to zona flexoria craniofacialis (ZFC): at same level of ZFC (1); caudal to ZFC (2); rostral to ZFC (3).

62. Lacrimal, ankylosed to jugal or touching jugal bar: absent (0); present (1). Musser and Cracraft [44], character 30; Livezey and Zusi [95], character 195; Mayr and Clarke [63], character 12.

63. Lacrimal, ankylosis to maxillopalatine processes/touching maxillopalatine processes: absent (0); present (1).

64. Lacrimal, supraorbital process, length: truncate (1); elongate (2).

65. Lacrimal, especially supraorbital process, inflated: no (1); yes (2).

66. Lacrimal, supraorbital process, general mediolateral width: narrow (1); wide (2).

67. Lacrimal, supraorbital process, gap present between interorbital area and caudal terminus of supraorbital process: no (1); yes (2). Musser and Cracraft [44], character 31; Livezey and Zusi [95], character 206.

68. Lacrimal, supraorbital process, tapered toward caudal apex: no (1); yes (2).

69. Lacrimal, supraorbital process, orientation of flattening: flattened dorsoventrally or craniocaudally (in line with coronal and/or transverse planes) (1); flattened mediolaterally (in line with parasagittal plane) (2); flattened ventrolaterally (3).

70. Lacrimal, rostral terminus of head of lacrimal, shape of apex: rounded or subrectangular (1); acuminate (2).

71. Lacrimal, rostral terminus of head of lacrimal, ankylosed to nasals: yes (1); no (2).

72. Lacrimal, rostral terminus of head of lacrimal, ankylosed to mesethmoid: no (1); yes (2).

73. Lacrimal, rostral terminus of head of lacrimal, extends rostrally with terminus medial to nasals: no (1); yes (2).

74. Lacrimal, additional rostral process along rostral margin of body ventral to rostral terminus of lacrimal head: absent (0); present (1).

75. Lacrimal, additional rostral process (if present), shape: rounded (1); acuminate (2); subrectangular (3). Noncomparable where absent and for *Phoenicopterus* due to incomplete ossification, and noncomparable where region fused to bill or mesethmoid or jugal.

76. Lacrimal, additional rostral process (if present), ankylosed to mesethmoid via bony strut: no (1); yes (2). Noncomparable where additional rostral process absent.

77. Lacrimal, body/orbital process, foramen: absent (0); present (1). Musser and Cracraft [44], character 32; Livezey and Zusi [95], character 197.

78. Lacrimal, descending process: wide and robust (1); narrow (2). Musser and Cracraft [44], character 33; Clarke et al. [114], character 11.

79. Lacrimal, descending process, formen or foramina: absent (0); present (1).

80. Lacrimal, descending process, convexity: concave (1); convex (2).

81. Lacrimal, descending process: elongate (1); truncate (2).

82. Lacrimal, ventral apex, shape: rounded or subrectangular (1); acuminate (2).

83. Lacrimal, ventral apex (or apices): terminal end(s) projects ventrally and/or caudally (1); terminal end(s) projects cranially (2). Musser and Cracraft [44], character 36.

84. Ectethmoid, pneumatic foramen (exclusive of foramen orbitonasale laterale): absent (0); present, one (1); present, 2 or more (2). Noncomparable in absence of ectethmoid. Musser and Cracraft [44], character 37.

85. Ectethmoid, size: absent (0); well developed (1); poorly developed (2). Musser and Cracraft [44], character 38; Livezey and Zusi [95], character 189; Mayr and Clarke [63], character 14.

86. Ectethmoid, ankylosed to lacrimal: present (1); lost (2). Noncomparable where absent. Musser and Cracraft [44], character 44.

87. Ectehmoid, foramen orbitonasalis medialis: v-shaped (1); small, circular (2); elongate, ovoid (3); lost (4). Noncomparable when absent due to lack of ankylosis of ectehmoid to ventral portion of antorbital area.

88. Olfactory sulcus, status and depth: shallow (1); deep (2).

89. Fonticulus interorbitalis, status: present (1); lost, only optic foramen present (2).

90. Fonticulus interorbitalis, craniocaudal extent: craniocaudally limited to caudal half of interorbital septum (1); craniocaudally extensive (2). Noncomparable where absent.

91. Fonticulus orbitocranialis, caudal extent: limited or essentially absent (1); extensive, continues along coronal plane of squamosal region (2).

92. Optic nerve foramen, completely confluent with closest lateral foramen/foramina (e.g., oculomotor nerve foramen, ophthalmic nerve foramen): no (1); yes (2).

93. Fonticulus interorbitalis and fonticulus orbitocranialis, confluent: no (1); yes (2). Noncomparable where absent.

94. Jugal, pronounced lateral bowing (convexity): absent, essentially straight (0); present and curved (1). Musser and Cracraft [44], character 42; Livezey and Zusi [95], character 298.

95. Jugal (especially maxillary portion), pronounced ventral bowing (convexity): absent (0); present (1). Musser and Cracraft [44], character 43; Livezey and Zusi [95], character 299.

96. Jugal, orientation of compression: absent, quadratojugal bar rounded (0); present, mediolateral compression evident so that bar is thin mediolaterally and thick dorsoventrally (1); present, dorsoventral compression evident so that bar is thin dorsoventrally and wide mediolaterally (2).

97. Jugal, slight medial bowing (concavity): absent (0); present (1).

98. Vomers, mediolaterally wide: yes (1); no (2). Mayr and Clarke [63], character 20.

99. Vomers, forming a midline, narrow, and dorsoventrally high lamella: yes (1); no (2). Mayr and Clarke [63], character 21.

100. Vomers, caudal ends not fused, more or less deeply cleft: no (1); yes (2). Mayr and Clarke [63], character 19.

101. Palatines, contact with premaxilla or maxilla: palatines have contact with maxillae only (1); palatines have contact with premaxillae (2). Musser and Cracraft [44], character 45; Cracraft and Clarke [115], character 8.

102. Palatines, long and thin, especially anteriorly, poorly developed caudally, and widely separated anteriorly: absent (0); present (1).

103. Palatines, contact each other along midline along ventromedial margin: yes (1); no (2).

104. Palatines, ventral height of medial and lateral crests relative to each other: height of lateral crest greater than that of medial crest (1); heights subequal or crests barely visible (2); height of medial crest is greater than that of lateral crest (3). Musser and Cracraft [44], character 46.

105. Palatines, inflated: no (1); yes (2).

106. Palatines, lateral portion: well-developed (1); rudimentary or vestigial (2). Musser and Cracraft [44], character 47; Livezey and Zusi [95], character 447; Mayr and Clarke [63], character 16.

107. Palatines, lateral portion, marked caudolateral orientation: absent (0); present (1).

108. Palatines, ventrally raised oblique crest that extends from lateral to medial margins of ventral face: absent (0); present (1). Musser and Cracraft [44], character 50; Livezey and Zusi [95], character 457.

109. Palatines, ventrally raised oblique crest (if present), rostrocaudal location: located at level of caudolateral angle (1); located further rostrally, typically at rostrocaudal midpoint of palatines (2). Noncomparable where absent.

110. Palatines, caudomedial angle, pneumatic foramen: absent (0); present (1). Noncomparable where absent. Musser and Cracraft [44], character 51.

111. Palatines, choanalis, ventral lamella, medial separation of bilateral lamellae: moderate (1); great (2). Musser and Cracraft [44], character 53; Livezey and Zusi [95], character 444.

112. Palatines, choanalis, ventral lamella, rostrocaudal position of caudomedial angle relative to that of caudolateral angle: coincident (1); rostral (2); caudal (3). Not comparable where caudolateral angle and/or caudomedial angle absent. Musser and Cracraft [44], character 55.

113. Palatines, caudolateral angle: absent (0); present (1).

114. Palatines, caudolateral angle, position relative to area of ankylosis with pterygoids: rostral (1); subequal (2); caudal (3). Not comparable where caudolateral angle absent.

115. Palatines, caudolateral angle, shape: rounded (1); acuminate (2). Not comparable where caudolateral angle absent.

116. Palatines, pterygoid process, pneumatic foramen: present (1); lost (2). Not comparable in absence of pterygoid process. Musser and Cracraft [44], character 57.

117. Pterygopalatine juncture, form: syndesmosis and pterygo-palatina propria, extensive rostrocaudally, caudal terminus approaching processus quadraticus pterygoidei (1); articulatio mesipterygo-palatina, with rudimentary gomphosis intrapterygoidea (2); articulatio pterygo-palatina simplex (3); articulatio mesipterygo-palatina, with complete gomphosis intrapterygoidea (4). Musser and Cracraft [44], character 58; Livezey and Zusi [95], character 601.

118. Palatines and pterygoids: sutured (1); segmented (articulated) (2). Musser and Cracraft [44], character 59; Cracraft and Clarke [115], character 6.

119. Pterygoid, facies articularis palatina, dorsoventral site of pterygopalatine juncture relative to parasphenoid rostrum (regardless of articulation via basipterygoid processes): slightly ventral, articulatio pterygo-rostroparasphenoidalis absent (1); on rostrum, articulatio pterygo-rostroparasphenoidalis present (2); markedly ventral, articulation of pterygo-rostroparasphenoidalis absent (3). Musser and Cracraft [44], character 61; Livezey and Zusi [95], character 480.

120. Pterygoid, rostral end markedly widened: absent (0); present (1). Noncomparable for Paleognathae. Bertelli et al. [14], character 9; Mayr [116].

121. Pterygoid, dorsomedial margin, craniocaudally elongate crista that is well projected dorsally and/or medially, may or may not have a facet for articulation with basipterygoid processes, rostrocaudal length: absent (0); present (1).

122. Pterygoid, dorsomedial margin, craniocaudally elongate crista that projects dorsally and/or medially (if present), rostrocaudal length: truncate, typically the length of roughly 1/4–1/2 of corpus (1); elongate, makes up entire length of pterygoid (2). Noncomparable where crista absent.

123. Pterygoid, dorsolateral margin, concavity: lost or slight (1); deep (2).

124. Pterygoid, ventromedial concavity: lost or slight (1); deep (2).

125. Pterygoid, area immediately rostromedial to area of articulation with quadrate, thin, accessory projection located on the dorsal face: absent (0); present (1).

126. Pterygoid, robust process that projects from medial margin of pterygoid medially towards basisphenoid for articulation with basipterygoid processes: absent (0); present (1).

127. Basisphenoid, basipterygoid processes: absent (0); present (1). Musser and Cracraft [44], character 64; Cracraft and Clarke [115], character 33.

128. Basisphenoid, basipterygoid processes, location: well rostral to hypoglossal canals (typically well rostral to basioccipital) (1); just rostral to hypoglossal canals (typically within margin of basioccipital) (2). Noncomparable where absent.

129. Basisphenoid, basipterygoid processes, ankylosed to pterygoids: yes (1); no (2). Noncomparable where absent.

130. Basisphenoid, basipterygoid processes, length: elongate (1); truncate (2). Noncomparable where absent.

131. Basisphenoid, basipterygoid processes, facet for articulation with pterygoids, shape: ovoid (1); acuminate (2). Noncomparable where absent. Musser and Cracraft [44], character 93; Mayr and Clarke [63], character 24.

132. Basiparasphenoid plate is inflated, rounded and broad: absent (0); present (1). Mayr and Clarke [63], character 26; Cracraft and Clarke [115]; Cracraft [117].

133. Basiparasphenoid plate, well-marked depression on posterolateral side of basiosphenoid plate containing numerous foramina for arteries and cranial nerves: absent (0); present (1). Mayr and Clarke [63], character 26; Cracraft and Clarke [115]; Cracraft [117].

134. Quadratopterygoid juncture: articulatio complex, involving both broad contact on facies medialis of processus orbitalis supplementary to condylus pterygoideus quadraticum (0); articulatio duplex, moderate dorsal extension on medial face of orbital process combined with condylus pterygoideus (1); articulatio simplex, retaining vestigial contact on basis of processus orbitalis in addition to primary articulatio with condylus pterygoideus (2); articulatio simplex, virtually limited to condylus pterygoideus (3). Musser and Cracraft [44], character 65; Livezey and Zusi [95], character 600; Mayr and Clarke [63], character 26.

135. Temporal fossa: mostly only visible from lateral view, terminal end located ventrocranially to nuchal crest (1); edges of temporal fossae almost meet or do meet above nuchal crest (2). Musser and Cracraft [44], character 66; Bertelli et al. [14], character 19.

136. Temporal fossa, crest: shallow, at same level of top of skull (1); consists of a distinctive crest that is raised prominently (2). Musser and Cracraft [44], character 67.

137. Zygomatic process, status: present (1); lost (2). Mayr and Clarke [63], character 33.

138. Zygomatic process, length: truncate (1); elongate, reaches level of base of orbital process of quadrate (2). Noncomparable where absent. Musser and Cracraft [44], character 68; Livezey and Zusi [95], character 146.

139. Zygomatic process, ankylosed to postorbital process: no (1); yes (2). Noncomparable where absent.

140. Zygomatic process, divided: no (1); yes (2). Noncomparable where absent or if aponeurosis present. Musser and Cracraft [44], character 69.

141. Zygomatic process, dorsal and ventral branches, relative rostrocaudal position if divided: dorsal branch more elongate cranially than ventral branch (1); subequal (2). Noncomparable where absent or if aponeurosis present.

142. Zygomatic process, articular notch for lateral head of quadrate: notch facing cranially (1); notch angled ventrally (2). Noncomparable in absence of the zygomatic process. Musser and Cracraft [44], character 70.

143. Zygomatic process, ossifications (aponeurosis): absent (0); present (1). Noncomparable where absent.

144. Zygomatic process, rostrocaudal orientation: at level of postorbital process (1); significantly caudal to postorbital process (2). Noncomparable where absent.

145. Processus suprameaticus: absent as distinct processus, in most or all cases the homologous bone evidently continuous as rostral margin of meatus acusticus externus (0); present as variably prominent postorbital process (1). Musser and Cracraft [44], character 71; Livezey and Zusi [95], character 143.

146. Processus suprameaticus (if present), shape: subrectangular (1); acuminate (2). Noncomparable where absent.

147. Occipital, foramen dorsomediana, cf. foramen (ostium): absent, foramina bilaterally symmetrical within occipital region (0); present, distinctly dorsal, often proximate to

transverse nuchal crest (1). Musser and Cracraft [44], character 72; Livezey and Zusi [95], character 77.

148. Occipital, transverse nuchal crest, dorsal portion: shallow or barely visible (1); distinct and prominent, raised dorsally (2). Musser and Cracraft [44], character 73.

149. Occipital, fonticuli occipitalis: absent (0); present, large, ovoid and perforate (1). Musser and Cracraft [94], character 74; Mayr and Clarke [63], character 27; Livezey [118], character 9; Ericson [119], character 1.

150. Occipital, foramen v. occipitalis externae: present or very distinctly visible (1); absent or smoothed to the point of being hardly or not visible (2). Musser and Cracraft [44], character 76.

151. Occipital, supraoccipital eminence: right and left portions of occipital complex indistinguishable or have very faint, smoothed division (1); occipital complex distinctly separated by median nuchal crest, appearance of a line or extremely bulbous process dividing the complex (2). Musser and Cracraft [44], character 77.

152. Occipital, foramen magnum: round or dorsoventrally ovoid (1); lateromedially elongate (2). Musser and Cracraft [44], character 78; Livezey and Zusi [95], character 27.

153. Occipital, foramen n. abducentis: round (1); craniocaudally elongate (2). Musser and Cracraft [44], character 79.

154. Occipital, occipital condyle, form: essentially circular (1); distinctly bilobate or reniform, lobes partitioned by medial condylar notch, lateromedially elongate (2); essentially round but flattened along ventral margin of foramen magnum (3). Musser and Cracraft [44], character 80; Livezey and Zusi [95], character 21.

155. Occipital, occipital condyle, rostrocaudal position relative to exoccipital, processus paroccipitalis: rostral (1); approximately equal or caudal (2). Musser and Cracraft [44], character 81; Livezey and Zusi [95], character 24.

156. Occipital, subcondylar fossa: deep (1); shallow (2); lost (3). Musser and Cracraft [44], character 82.

157. Occipital, hypoglossal nerve foramen: circular (1); elongate (2). Musser and Cracraft [44], character 83.

158. Occipital, regardless of supraoccipital eminence or exoccipital, rostrocaudal position relative to nuchal crest: caudal (1); subequal or rostral (2).

159. Occipital, dorsal apex of nuchal crest, dorsoventral position: subequal or ventral to dorsal base of postorbital process (1); dorsal to postorbital process (2).

160. Occipital, basioccipital and basiparasphenoid: medially compressed (1); laterally splayed (2). Musser and Cracraft [44], character 86.

161. Occipital, fossa parabasalis: absent (0); present (1). Musser and Cracraft [44], character 87; Livezey and Zusi [95], character 120.

162. Occipital, fossa parabasalis (if present), depth: shallow (1); deep (2). Noncomparable where absent. Musser and Cracraft [44], character 87; Livezey and Zusi [95], character 120.

163. Occipital, basioccipital, crista basilaris transversa: absent or barely visible (0); present (1).

164. Basiparasphenoid, pneumatic foramen in center: absent (0); present (1). Musser and Cracraft [44], character 92.

165. Occipital, basioccipital, concave: no, convex (1); yes (2).

166. Basiparasphenoid, processus medialis parasphenoidalis, alignment: lateral (1); ventral (2).

167. Basiparasphenoid, eustacian tubes fossa: deep (1); shallow (2). Noncomparable where basisphenoid covering homologous site. Musser and Cracraft [44], character 94.

168. Pila otica, status: present (1) lost, only cotyla quadrati otici present so that recessus tympanicus completely exposed (2).

169. Squamosal, laterally splayed: no (1); yes (2).

170. Dorsally projecting process on quadrate anterior to quadrate-prootic articulation (attachment for m. adductor mandibulae externum, pars profundus): absent or very poorly developed (0); present and well developed (1).

171. Quadrate, otic process, lateral face: convex or flat (1); slightly concave (2); deeply concave (3). Musser and Cracraft [44], character 96.

172. Quadrate, otic process, medial head, markedly extended caudally compared to lateral head: no, positions subequal (1); yes (2). Noncomparable where heads not well separated.

173. Quadrate, otic process, lateral head, rostrocaudally compressed: no (1); yes (2). Noncomparable where heads not well separated.

174. Quadrate, medial condyle: present (1); essentially absent, linear (2).

175. Quadrate, lateral capitulum, rostrocaudal position relative to quadratojugal articulation: quadratojugal articulation rostral (1); subequal (2); caudal to lateral head (3).

176. Quadrate, tuberculum subcapitulare under lateral capitulum, status: absent (0); present (1).

177. Quadrate, tuberculum subcapitulare (if present), prominence: prominent, tuberculate (1); diminutive, barely raised scar (2). Noncomparable where absent.

178. Quadrate, tuberculum subcapitulare (if present), ankylosis to zygomatic process: no (1); yes (2). Noncomparable where absent.

179. Quadrate, otic process, crista tympanica, prominence: absent or indiscernible (1); present, shallow (2); present, extremely prominent crista (3).

180. Quadrate, otic process, crista tympanica, ventral apex, dorsoventral location: within dorsal half of quadrate or at midpoint (1); within ventral half of quadrate (2). Not comparable where crista absent.

181. Quadrate, otic process, crista tympanica, mediolateral location relative to lateral capitulum: along midline of lateral capitulum (1); encompasses entire lateral capitulum (2). Not comparable where crista absent.

182. Quadrate, otic process, cotylae: cotylae confluent (1); cotylae completely separated (2). Musser and Cracraft [44], characters 97–98; Livezey and Zusi [95], characters 150 and 551; Mayr and Clarke [63], character 34.

183. Quadrate, otic process, caudal face dorsal to crista tympanica (or homologous site), deeply concave: no (1); yes (2).

184. Quadrate, otic process (caudal aspect), mediolateral bowing toward ventral end: no, linear (1); yes, lateral bowing (2).

185. Quadrate, otic process, capitulum, medial face, pneumatic foramen: present (1); lost (2). Musser and Cracraft [44], character 99; Livezey and Zusi [95], character 508.

186. Quadrate, otic process, capitulum, medial face, pneumatic foramen (if present), location with respect to crista medialis and crista tympanica: rostral to crista medialis and crista tympanicum (1); rostral to crista tympanicum but caudal to crista medialis (2); like state 1 but located further ventrally (3). Noncomparable where foramen absent.

187. Quadrate, otic process, capitulum, medial face, pneumatic foramen, dorsoventral location: present within dorsal 2/3 (1); present within ventral third (2). Noncomparable where foramen absent.

188. Quadrate, otic process, capitulum, caudal face, pneumatic foramen, status: present (1); lost (2). Musser and Cracraft [44], character 100; Livezey and Zusi [95], character 554.

189. Quadrate, otic process, capitulum, caudal face, pneumatic foramen, shape: circular (1); elongate and ovoid (2). Noncomparable where foramen absent.

190. Quadrate, orbital process, medial face, depressio protractoris, form: flattened or only slightly concave (1); deeply concave (2). Noncomparable where orbital process absent.

191. Quadrate, orbital process, status: present (1); lost (2). Musser and Cracraft [44], character 102; Livezey and Zusi [95], character 532.

192. Quadrate, orbital process, length relative to otic process: subequal (1); shorter than otic process (2); longer than otic process (3). Noncomparable where orbital process absent. Musser and Cracraft [44], character 103; Livezey and Zusi [95], character 534.

193. Quadrate, orbital process, medially bowed: yes (1); no (2). Noncomparable where orbital process absent.

194. Quadrate, orbital process, crista orbitalis, prominence: shallow (1); prominent (2). Noncomparable where orbital process absent.

195. Quadrate, orbital process, shape of terminus: terminus subequal or wider than orbital process (1); exceptionally tapered toward terminus (2). Noncomparable where orbital process absent. Musser and Cracraft [44], character 104–105; Bertelli et al. [14], character 25; Livezey and Zusi [95], character 535 and 538.

196. Quadrate, medial condyle, orientation: aligned cranio-caudally (1); aligned medio-laterally (2). Musser and Cracraft [44], character 107.

197. Quadrate, medial condyle, shape: convex and bulbous, undivided (1); bilobate, 2 lobes approximately equal in size (2); bilobate, lateral lobe distinctly smaller than medial lobe (3). Musser and Cracraft [44], character 108; Livezey and Zusi [95], character 518.

198. Quadrate, mandibular process, facies articulais pterygoidea (ventral face in those taxa having two): facies articularis, with slight anteromedial eminentia on basis (1); condylar, tubercular, or jugosublinear (2). Musser and Cracraft [44], character 109; Livezey and Zusi [95], character 523.

199. Quadrate, caudal condyle and mandibular articulation: present (1); lost (2).

200. Quadrate, caudal condyle, shape (caudal aspect): dorsoventrally compressed (1); bulbous (2). Noncomparable where caudal condyle absent.

201. Quadrate, lateral condyle, location relative to articulation with quadratojugal: caudal (1); rostral (2); subequal (3).

202. Quadrate, lateral condyle, relatively large and oriented lateromedially: absent (0); present (1).

203. Quadrate, lateral condyle, size compared to caudal condyle: smaller than caudal condyle (1); larger than caudal condyle (2); subequal (3). Noncomparable where caudal condyle absent.

204. Quadrate, caudal condyle, orientation (caudal aspect): medial margin ventral to lateral margin (1); horizontal (2); lateral margin ventral to medial margin (3). Noncomparable where caudal condyle absent.

205. Quadrate, caudal condyle, confluence with lateral condyle: present (1); lost (2). Noncomparable where caudal condyle absent.

206. Quadrate, lateral condyle, dorsoventral position relative to caudal condyle regardless of confluence: caudal condyle positioned only slightly dorsal to lateral condyle (1); lateral condyle terminates well ventral to caudal condyle (2). Noncomparable where caudal condyle absent.

207. Quadrate, facies articularis quadratojugalis: fovea and cotyla (1); incisura—concave, troughlike, raised margin lacking entirely or at least in two, geometrically opposing points (2). Musser and Cracraft [44], character 114; Livezey and Zusi [95], character 511.

208. Quadrate, lateral margin of processus lateralis dorsal to cotyla quadratojugalis, prominent tubercle present that projects rostrolaterally past cotyla quadratojugalis: absent (0); present (1). Noncomparable where cotyla/fovea absent. Musser and Cracraft [44], character 40.

209. Quadrate, caudal condyle, medial to medial head of quadrate: no (1); yes (2). Noncomparable where cotyla/fovea absent.

210. Quadrate, facies articularis quadratojugalis, cotyla quadratojugalis, orientation: oriented laterally (1); oriented rostrally (2); oriented dorsally (3). Noncomparable where cotyla/fovea absent.

211. Quadrate, facies articularis quadratojugalis, cotyla quadratojugalis, exceptional dorsoventral thickening of the rostroventral margin of cotyla: absent (0); present (1). Noncomparable where cotyla/fovea absent.

212. Quadrate, lateral aspect, cotyla quadratojugalis and processus lateralis, caudodorsally oriented, elongate flange that is separate from lateral and caudal condyles: absent (0); present (1).

*Skull: Mandible*

213. Articular surface, having single centrally located ridge oriented rostrocaudally: absent (0); present (1).

214. Articular surface, posteromedial and lateral walls: absent (0); present (1).

215. Symphysis, lamina ventral to foramina neurovascularis, craniocaudal alignment with dorsal lamina of symphysis: subequal, dorsal and ventral lamina craniocaudally aligned (1); ventral lamina extends caudal to dorsal lamina (2). Musser and Cracraft [44], character 115; Livezey and Zusi [95], character 654.

216. Symphysis, length as proportion of total length of mandible (not including retroarticular process if present): short, less than 1/5 (1); intermediate, between 1/5 and 1/3 (2); long, between 1/3 and 1/2 (3). Musser and Cracraft [44], character 116; Livezey and Zusi [95], character 676.

217. Ramus, general form: medially compressed (1); laterally splayed so that ramus on each side is mediolaterally oriented (2). Musser and Cracraft [44], character 117; Livezey and Zusi [95], character 280.

218. Symphysis, orientation: straight or dorsally oriented (1); apex ventrally recurved (2).

219. Ramus, portion rostral to coronoid process, curvature: present, variably but distinctly decurved (1); obsolete, i.e., virtually straight (2). Musser and Cracraft [44], character 119; Livezey and Zusi [95], character 673.

220. Coronoid process, shape: rounded (1); subrectangular (2).

221. Tuberculum m. adductor mandibulae externus, pars articularis, caput externus (see tuberc. m. AME in Livezey and Zusi 2006), laterally extensive: absent (0); present (1).

222. Ventral angle: absent or indistinct (0); present, marked (1). Musser and Cracraft [44], character 120; Livezey and Zusi [95], character 681.

223. Fenestra caudalis mandibulae, size: small, essentially absent (1); large and perforate (2).

224. Fenestra rostralis mandibulae, size: essentially absent, thin and slit-like (0); substantial and transverse, completely or largely perforate (1). Musser and Cracraft [44], character 121; Livezey and Zusi [95], character 684.

225. Surangular, dorsoventrally thick: no (1); yes (2).

226. Surangular and dentary, dorsoventrally thicker than articular area/angular: no (1); yes (2).

227. Angulus mandibulae, dorsally prominent: no (1); yes (2). Noncomparable for Anseriformes.

228. Prearticular: extends to rostral margin of fenestra rostralis mandibulae (1); only extends to caudal margin of fenestra rostralis mandibulae (2).

229. Area of articulation with quadrate, lateral cotyle, mediolaterally compressed: no (1); yes (2).

230. Retroarticular process (if present), dorsoventral thickening: absent, processes thin and elongate (1); present, processes dorsoventrally thick and mediolaterally flattened (2).

231. Dentary, caudal development: strongly forked (1); weakly forked posteriorly into dorsal and ventral rami (2). Musser and Cracraft [44], character 123; Cracraft and Clarke [115], character 4.

232. Area of articulation with quadrate, lateral cotyle, orientation of rostral margin compared to that of caudal margin: oriented laterally (1); oriented medially (2). Musser and Cracraft [44], character 124.

233. Area of articulation with quadrate, internal articular process, pneumatic foramen: absent (0); present (1).

234. Area of articulation with quadrate, fossa articularis quadratica, sulcus intercotylaris, pneumatic foramina: absent (0); present (1). Musser and Cracraft [44], character 126; Livezey and Zusi [95], character 696.

235. Area of articulation with quadrate, cotylae fossae articularis, tuberculum intercotylare: variably tuberculate with intervening depressions (1); single, centrally positioned, rostrocaudally oriented jugum (2).

236. Area of articulation with quadrate, internal articular process: truncate (1); elongate (2).

237. Area of articulation with quadrate, processus lateralis mandibulae, size: small, diminutive (1); prominent (2). Note: not to be confused with cotyla lateralis, it is located rostral to this feature.

238. Area of articulation with quadrate, dorsally projecting hook-like accessory projection present caudolateral to cotyla caudalis: absent (0); present, dorsoventrally elongate and hook-like (1); present but truncate and rounded, sometimes present as a pedestal-like projection with a flattened dorsal surface (2). Noncomparable where articulation for caudal condyle of quadrate is absent in mandible. Note: This character is not to be confused with a true retroarticular process seen in e.g., galloanserines and Phoenicopteridae. The difference is that while the hook-like process seen in Neoaves (e.g., Gruiformes) is only present caudolateral to the caudal condyle (between area of articulation of caudal and lateral condyles of quadrate), the true retroarticular process is medial to or within the margin of the caudal cotyla or medial to the cotyla lateralis where the articulation for the caudal condyle is absent. The true retroarticular process must incorporate medial lamina of the mandible. Some Sphenisciformes appear to have a true retroarticular process, but that is actually the area of the caudal fossa of the mandible that has been elevated and flattened dorsally. Musser and Cracraft [44], character 129; Bertelli et al. [14], character 26.

239. Dorsally projecting hook-like accessory projection present caudolateral to cotyla caudalis (if present), extreme caudal elongation of all dorsoventral lamina: absent or only slight caudal projection present (0); present, creates mediolaterally flattened "pseudo processus retroarticularius" (1). See note above. Noncomparable where absent.

240. True retroarticular process: absent (0); present (1).

241. True retroarticular process (if present), medial margin and impressio m. depressor mandibulae, hollow: no (1); yes (2). Noncomparable where absent.

242. Area of articulation with quadrate, extreme dorsoventral flattening: not present, area dorsoventrally robust (1); present (2).

243. Area of fossa caudalis/impressio m. depressor mandibulae, ventral apex, rostrocaudal position: apex subequal in rostrocaudal position to caudal margin of area of articulation with quadrate (1); rostral, apex rostral to caudal margin of area of articulation with quadrate (2); caudal, apex extended caudally so that caudal fossa visible in dorsal aspect (3).

244. Fossa caudalis, status and depth: absent (0); present, shallow (1); present, deep (2). Noncomparable where hollow.

*Vertebrae*

245. Number of presacral vertebrae: 20–22 (1); 18–19 (2); 23 or more (3). Mayr and Clarke [63], character 55.

*Vertebrae: Cervical Vertebrae*

246. Atlas, processus ventralis corporis, location of caudal apex: caudal (1); ventrocaudal (2).

247. Atlas, fossa condyloidea, shape: a complete, circular facet (1); lateromedially elongate with a subrectangular facet, does not form circle (2).

248. Axis, spinous process, terminus caudal to corpus: no (1); yes (2).

249. Axis, ventral process: essentially absent (0); present, represented by variably thick, rounded or subangular crista (1); present, represented by a ventrally elongated, caudally

deflected, rounded spina (2); present, represented by a ventrally elongated, bilaterally compressed, craniocaudally restricted lamina (3). Musser and Cracraft [44], character 133; Livezey and Zusi [95], character 783.

250. Axis, processus costalis: present, typically vestigial, i.e., represented only by head (1); lost (2). Musser and Cracraft [44], character 134; Livezey and Zusi [95], character 787; Mayr and Clarke [63], character 50.

251. Axis, foramen transversarium: absent (0); present (1). Note: related in part to prominence of processus costalis. Musser and Cracraft [44], character 135; Mayr and Clarke [63], character 49; Livezey [113], character 104.

252. Axis, lateral lamina of arcus, pneumatic foramen: absent (0); present (1). Musser and Cracraft [44], character 136; Mayr and Clarke [63], character 48; Livezey [113], character 107.

253. Cervical vertebrae, marked heterogeneity of form involving relative elongation of intermediate elements (regardless of length): present (1); lost (2). Musser and Cracraft [44], character 137; Livezey [113], character 111.

254. Cervical vertebrae, corpus, dorsoventrally compressed: no (1); yes (2).

255. Cervical vertebrae, corpus, extremely craniocaudally elongate: no (1); yes (2).

256. Cervical vertebrae, section I, arcus interzygopophysialis lateralis: absent on all elements (0); present (1). Musser and Cracraft [44], character 138; Livezey [113], character 806.

257. Cervical vertebrae, section I, arcus vertebrae, spinous processes, craniocaudal location of apices: close to cranial margin or midpoint of vertebrae (1); reaching caudal margin of vertebrae (2).

258. Cervical vertebrae, section I, arcus vertebrae, spinous processes, shape: subrectangular (1); rounded (2); bifurcated (3).

259. Cervical vertebrae, section I, arcus vertebrae, spinous processes, dorsal lengths: dorsoventrally truncate (1); dorsoventrally elongate (2).

260. Cervical vertebrae, section II, arcus vertebrae, spinous processes, craniocaudal locations of apices: close to cranial margin of vertebrae (1); at midpoint of vertebrae (2).

261. Cervical vertebrae, section II, zygapophysis caudalis, dorsal portion (torus dorsalis): convex (1); concave (2).

262. Cervical vertebrae, section II, zygapophysis caudalis: craniocaudally elongate (1); craniocaudally truncate (2).

263. Cervical vertebrae, section II, costal processes: present throughout section II (1); present in cranial elements but absent in caudal elements of section II (2); absent throughout section II (3). Musser and Cracraft [44], character 139; Livezey [113], character 820.

264. Cervical vertebrae, section II, ansa costotransversaria: as craniocaudally elongate as at approximately half of vertebra (1); craniocaudally short so that ansa is as elongate as 1/3 of vertebra or less (2); like state 1 but toward cranial elements can be essentially as craniocaudally elongate as entire corpus (3).

265. Cervical vertebrae, section II, costal processes (if present), ansa costotransversaria, foramen: absent (0); present (1).

*Vertebrae: Thoracic Vertebrae*

266. Thoracic vertebrae, corpus, lateral face, pneumatic foramen: absent (0); present (1). Musser and Cracraft [44], character 143; Livezey and Zusi [95], character 850.

267. Thoracic vertebrae, corpus, bilateral compression manifested by virtually laminar structure of corpus between facies articulares craniali et caudalis: no, corpus cylindrical (1); present (2). Musser and Cracraft [44], character 144; Livezey and Zusi [95], character 858.

268. Thoracic vertebrae, dorsal margins of spinous processes, overlapping craniocaudally (excluding those within notarium, if present): no (1); yes (2).

269. Thoracic vertebrae, dorsal surface and spinous processes, extensive aponeurosis ossificans: no (1); yes (2).

270. Thoracic vertebrae, dorsal lamina of arcus, recessus dorsocranialis pneumatici: absent (0); present (1). Musser and Cracraft [44], character 145; Livezey and Zusi [95], character 866.

271. Thoracic vertebrae, dorsal lamina of arcus, fovea interzygopophysialis, pneumatic foramina: absent (0); present (1). Musser and Cracraft [44], character 146; Livezey and Zusi [95], character 867.

272. Thoracic vertebrae, transverse processes, craniocaudal gradual but distinct lateral elongation of processes in seratium among presynsacral elements: absent, transverse processes essentially of uniform width throughout the presynsacral elements or moderately reversed grade (0); present, transverse processes distinctly increasing in width throughout the presynsacral elements (1). Musser and Cracraft [44], character 147; Livezey and Zusi [95], character 875.

273. Thoracic and sacral vertebrae, caudalmost elements (cranial to synsacral vertebrae), facies articularis caudalis (of penultimate element) and facies articularis cranialis (of ultimate element), type: series completely heterocoelous (1); at least part of series with subround, central articular surfaces (e.g., amphicoelous/opisthocoelous) that lack the dorsoventral compression and saddle-shaped articular surface seen in heterocoelous vertebrae (2). Musser and Cracraft [44], character 148; Livezey and Zusi [95], character 888; Mayr and Clarke [63], character 57.

274. Thoracic vertebrae, notarium: present (1); lost (2). Musser and Cracraft [44], character 151; Livezey and Zusi [95], character 892; Mayr and Clarke [63], character 55.

275. Thoracic vertebrae, notarium (if present), number of incorporated vertebrae: 4 (1); 3 (2). Noncomparable where absent. Musser and Cracraft [44], character 151; Livezey and Zusi [95], character 892.

276. Thoracic vertebrae, notarium (if present), corpus notarii, crista ventralis notarii, fenestrae intercristales: typically present, ligamenta intercristales of adjacent vertebrae notarii (often robustly) ossified (1); typically lost, cristae ventrales demarcated ventrally by incisurae cristae, ligamenta intercristales of adjacent vertebrae notarii not ossified (2). Not comparable for taxa without a notarium. Musser and Cracraft [44], character 152; Livezey and Zusi [95], character 895.

277. Thoracic vertebrae, notarium (if present), arcus notarii, foramina (intervertebralia) interarcuales: present, small, diameter approximately equal to that of fovea costalis (1); present, obsolete, reduced to minute perforations or evidently absent (2). Not comparable for taxa without a notarium. Musser and Cracraft [44], character 153; Livezey and Zusi [95], character 896.

278. Thoracic vertebrae, ventral processes, widely bifurcated (lateral termini extending laterally beyond costal facets): no (1); yes (2).

279. Caudalmost presacral vertebrae with deep lateral excavations: no (1); yes (2). Mayr and Clarke [63], character 58.

*Vertebrae: Synsacrum and Caudal Vertebrae*

280. Synsacrum (maximally including thoracic vertebrae): 14–18 (1); 10–13 (2). Musser and Cracraft [44], character 154; Livezey and Zusi [95], character 897; Mayr and Clarke [63], character 91.

281. Synsacrum, cranialmost vertebrae caudales synsacri, lateral faces of vertebrae, penetration and visibility through foramen ilioischiadicum (lateral perspective): absent (0); present (1). Musser and Cracraft [44], character 158; Livezey and Zusi [95], character 933.

282. Free caudal vertebrae, number (including pygostyle): 5–6 (1); 7–8 (2); 9–10 (3).

283. Pygostyle, fused to last free caudal via spinous process of caudal vertebra(e): no (1); yes (2).

284. Pygostyle, fused to last free caudal(s) via ventral processes: no (1); yes (2).

285. Pygostyle, foramen in lateral lamina: absent (0); present (1).

286. Free caudal vertebrae, spinous processes, mediolaterally wide bifurcation throughout series, status: present (1); lost (2).

287. Free caudal vertebrae, ventral processes, status: present (1); lost (2).

288. Free caudal vertebrae, ventral processes (if present), general orientation of apex: cranioventral (1); cranial (2).

289. Free caudal vertebrae, spinous process, form: rounded (1); subrectangular (2).

290. Free caudal vertebrae, transverse processes: truncate (1); elongate (2).

291. Pygostyle, apex, shape: subrectangular, tapered (1); rounded (2); hamate (3); spatulate (4).

292. Pygostyle, apex, dorsoventral departure from major axis of pygostyle: absent (0); dorsal (1); ventral (2). Noncomparable if spatulate.

293. Pygostyle, craniocaudal length: elongate (1); truncate (2).

*Sternum*

294. Spina externa rostri: absent or obsolete (0); present (1). Musser and Cracraft [44], character 159; Livezey and Zusi [95], character 1157; Mayr and Clarke [63], character 70.

295. Spina externa rostri, fused to furcula: no (1); yes (2).

296. Spina externa rostri, length: truncate, shorter than craniolateral processes (1); elongate, longer than craniolateral processes (2). Noncomparable where spina externa absent. Musser and Cracraft [44], character 159; Livezey and Zusi [95], character 1157; Mayr and Clarke [63], character 70.

297. Spina externa rostri, shape: fan-shaped (1); spine-like (2). Mayr and Clarke [63], character 70.

298. Spina externa rostri, completely bifurcated: no (1); yes (2).

299. Spina interna rostri: present (1); lost (2). Bertelli et al. [14], character 45.

300. Spina interna rostri, fused to spina externa rostri: yes (1); no (2). Noncomparable where spina interna and spina externa absent. Bertelli et al. [14], character 45.

301. Spina interna rostri (if present), pneumatic foramen: absent (0); present (1). Noncomparable where spina interna absent.

302. Spina externa rostri (if present), pneumatic foramen: absent (0); present (1). Noncomparable where spina externa absent.

303. Spina externa rostri (if present), crest along ventral midline: absent (0); present (1). Noncomparable where spina externa absent.

304. Spina externa rostri (if present), invagination of dorsal margin: present (1); lost (2).

305. Spina externa rostri (if present), exceptionally mediolaterally wide: no (1); yes (2).

306. Spina externa rostri (if present), lateral faces deeply concave: no (1); yes (2).

307. Depressiones (sulcus) articulares coracoidei, dorsal lip: reaches dorsally (1); reaches ventrally (2). Musser and Cracraft [44], character 160.

308. Depressiones (sulcus) articulares coracoidei, ventral lip: reaches ventrally (1); reaches dorsally (2). Musser and Cracraft [44], character 162.

309. Depressiones (sulcus) articulares coracoidei, crossed: no (1); yes (2). Mayr and Clarke [63], character 69; Ericson [119], character 34.

310. Processus craniolateralis: extremely long, creates fenestra (1); prominent but more truncate than state 1 (2); extremely truncate, barely projected (3). Musser and Cracraft [44], character 164; Livezey and Zusi [95], character 1141.

311. Processus craniolateralis with respect to major axis of carina, angle: extending laterally (1); extending rostrally (2). Musser and Cracraft [44], character 165; Livezey and Zusi [95], character 1142.

312. Processus craniolateralis, apex, shape: trapezoidal (1); rounded (2); cruciate (3).

313. Processus craniolateralis, apex, width compared to width of base: subequal (1); apex narrower than base (2).

314. Ventral face of sternum, base of processi craniolateralis and/or processus craniolateralis proprius, impressio origii m. sternocoracoidei: present (1); lost or extremely shallow (2). Musser and Cracraft [44], character 166; Livezey and Zusi [95], character 1149.

315. Visceral face of sternum, sulcus medianus sterni immediately caudal to cranial margin, pneumatic foramen and/or depression exclusive of pneumatic pores: absent (0);

present, undivided, enclosing pneumatic pores and os spongiosum (1); present, divided medially by osseus lamina or trabecula, enclosing pneumatic pores and os spongiosum (2). Musser and Cracraft [44], character 167; Livezey and Zusi [95], character 1108.

316. Visceral face of sternum, sulcus medianus sterni and/or pneumatic foramen: begins within margin of or is immediately caudal to pila coracoidea (1); significantly caudal to pila coracoidea, approximately caudal to base of carina (2). Musser and Cracraft [44], character 168; Livezey and Zusi [95], character 1109.

317. Visceral face of sternum, pneumatic pores, exclusive of those included within pneumatic foramen: absent (0); present, at cranial margin (1); present, along medial sulcus (2); present, within both cranial margin and medial sulcus (3). Musser and Cracraft [44], character 169; Livezey and Zusi [95], character 1110.

318. Ventral face, facies muscularis, sulcus ventrolateralis (longitudinal trough on ventral surface of element immediately medial to processus costales): absent or indistinct (1); distinct, typically for length of costal margin (2). Musser and Cracraft [44], character 171; Livezey [113], character 153.

319. Costal margin, craniocaudal length relative to that of entire sternum along median axis: less than $\frac{1}{4}$ (1); between 1/4 and $\frac{3}{4}$ (2); greater than $\frac{3}{4}$ (3). Musser and Cracraft [44], character 172; Livezey [113], character 1114.

320. Costal margin, articular surfaces for sternal ribs: 4 or less (1); 5–7 (2); 8–9 (3). Musser and Cracraft [44], character 173; Mayr and Clarke [63], character 71.

321. Costal margin, articular surfaces for sternal ribs, location: on processus craniolateralis and body (1); on body only (2); on processus craniolateralis only (3).

322. Costal margin, linea intermuscularis dorsolateralis: absent (0); present (1).

323. Costal margin, linea intermuscularis dorsolateralis, mediolateral location of caudal terminus: medial only to trabecula caudolateralis (if present) (1); medial to trabecula cauolateralis and intermediana (if present) but lateral to mediana (2); medial to lateral margin of trabecula mediana (3). Noncomparable where linea intermuscularis dorsolateralis absent.

324. Costal margin, linea intermuscularis dorsolateralis, caudal extent: within rostral half of sternum only (1); extends to caudal margin of sternum or well within caudal half of sternum (2). Noncomparable where linea intermuscularis dorsolateralis absent.

325. Carina, cranial margin, lateral crest: present (1); lost (2). Noncomparable for taxa without a carina. Musser and Cracraft [44], character 176; Livezey and Zusi [95], character 1207.

326. Carina, cranial margin, sulcus carinae: absent (0); present (1). Not comparable for taxa without a carina. Musser and Cracraft [44], character 177; Livezey and Zusi [95], character 1213.

327. Carina, sulcus carinae or homologous site, pneumatic foramen: absent (0); present, variable size and shape (1). Not comparable for taxa without a carina. Musser and Cracraft [44], character 178; Livezey and Zusi [95], character 1218.

328. Carina, sulcus carinae or homologous site, crista complete along midline: present (1); lost (2). Noncomparable where carina absent.

329. Carina, apex, site relative to spina externa rostri or most proximal point of sternum: subequal to caudal location of spina externa rostri or homologous site (1); located caudally (2); located extremely cranially (3). Noncomparable where carina absent. Musser and Cracraft [44], character 179; Livezey and Zusi [95], character 1198.

330. Carina, apex, fused to furcula: no (1); yes (2). Noncomparable where carina absent. Musser and Cracraft [44], character 180; Livezey and Zusi [95], character 1200.

331. Carina, apex, facet for articulation with furcula: absent (0); present, as incisura or concavity (1). Noncomparable where carina absent. Musser and Cracraft [44], character 180; Livezey and Zusi [95], character 1200.

332. Carina, cranial margin, relative widths of dorsal and ventral portions: dorsal margin wider (1); ventral margin wider (2); width of both portions subequal (3). Noncomparable where carina absent. Musser and Cracraft [44], character 181.



333. Carina, maximal depth ventral and normal to body of sternum, facies muscularis, relative to minimal width of body of sternum (exclusive of processes laterales, if present) across points on costal margin directly lateral to that of maximal depth of carina: height is equal to or greater than width of body, not including processus craniolateralis (1); height is less than width of body (2). Noncomparable where carina absent. Musser and Cracraft [44], character 182; Livezey and Zusi [95], character 1199.

334. Carina, apex, shape: rounded (1); acuminate (2); trapezoidal (3); bifurcated (4). Noncomparable where carina absent.

335. Carina, recurvature of cranial margin: absent (0); present (1). Noncomparable where carina absent.

336. Carina, apex and ventral keel, reinforced with thickened jugum: no (1); yes (2). Noncomparable where carina absent.

337. Carina, linea intermuscularis ventromedialis, status: present (1); lost (2). Noncomparable where carina absent.

338. Carina, lateral aspect, exceptionally concave: no (1); yes (2). Noncomparable where carina absent.

339. Carina, caudal margin, continues to caudal margin of trabecula mediana: yes (1); no (2). Noncomparable where carina absent.

340. Sternum, caudal margin: with 4 or more notches/fenestrae (1); with 2 notches/ fenestrae (2); without notches/fenestrae (3); with 3 notches/fenestrae (4). Musser and Cracraft [44], character 183; Livezey and Zusi [95], character 1182; Mayr and Clarke [63], character 73.

341. Incisura and fenestra caudolateralis (if present), cranial extent (length of fenestra measured using body of sternum, not trabecula lateralis): elongate—length of incisura and fenestra greater than 2/3 craniocaudal length of corpus sterni, approaching terminis caudalis of processus costales sterni (1); intermediate—length of incisura and fenestra between 1/3 and 2/3 of craniocaudal length of corpus sterni (2); abbreviate—length of incisura and fenestra less than 1/3 craniocaudal length of corpus sterni (3). Noncomparable where incisura (fenestra) and/or trabecula absent. Musser and Cracraft [44], character 184; Livezey and Zusi [95], character 1183.

342. Processus caudolateralis (if present), orientation relative to body of sternum as reflected (in part) by angle defined by incisura caudolateralis, cranial vertex of angle: angle undefined, processus is parallel to costal margin and "vertex" is ellipsoidal (1); laterally splayed, approximately 45 degrees (2); angle acute, less than 15 degrees (3). Noncomparable where incisura (fenestra) and/or trabecula absent. Musser and Cracraft [44], character 185; Livezey and Zusi [95], character 1184.

343. Processus caudolateralis (if present), space between process and body of sternum: typical, wide (1); extremely narrow so that almost no space is present between sternum body and trabecula caudolateralis (2). Noncomparable where incisura (fenestra) and/or trabecula absent.

344. Trabecula lateralis, status: absent (0); present (1).

345. Trabecula caudolateralis (if present), shape of caudal terminal margin: rounded or (sub)rectangular (1); (sub)acuminate (2); cruciate, with transverse pila (3). Noncomparable where presence of trabecula uncertain or known to be absent. Musser and Cracraft [44], character 186; Livezey and Zusi [95], character 1185.

346. Trabecula caudolateralis (if present), orientation of caudal terminal margin: caudal, does not deviate from trabecula lateralis axis (1); medial (2); lateral (3). Noncomparable where presence of trabecula uncertain or known to be absent.

347. Trabecula mediana, thickened, raised and bifurcated crista that connects to the ventral margin of the carina and extends to the caudolateral margin of trabecula mediana: absent (0); present (1). Noncomparable where carina absent.

348. Dorsal lip, labrum interna sterni, pair of rostrally oriented projections: absent (0); present (1). Musser and Cracraft [44], character 161.

349. Dorsal lip, labrum interna sterni, pair of rostrally oriented projections (if present), shape: rounded (1); acuminate (2). Noncomparable where processes absent. Musser and Cracraft [44], character 187.

350. Dorsal lip, labrum interna sterni, pair of rostrally oriented projections (if present), raised dorsally with accompanying raised jugae along visceral margin of sternum: no (1); yes (2). Noncomparable where processes absent.

351. Dorsal lip, labrum interna sterni, pair of rostrally oriented projections (if present and raised dorsally), sulcus present along midline: absent (0); present (1). Noncomparable if not raised along visceral face.

352. Dorsal lip, labrum interna sterni, pair of rostrally oriented projections (if present), mediolateral position: gap present between 2 processes (1); 2 processes contact each other (2). Noncomparable where processes absent.

353. Dorsal lip, labrum interna sterni, pair of rostrally oriented projections (if present), mediolateral width: wide (1); narrow (2). Noncomparable where processes absent.

354. Incisura and fenestra intermediana, relative cranial extent: rudimentary, incisura is a minor concavity (1); moderately elongate, length of incisura and fenestra less than 1/3 craniocaudal length of corpus sterni (2). Noncomparable where incisura (fenestra) and/or trabecula absent. Musser and Cracraft [44], character 188; Livezey and Zusi [95], character 1188.

355. Trabecula mediana, margin and caudal terminus, tapering: untapered or weakly tapered (1); distinctly tapered (2).

356. Trabecula mediana, caudal terminus, shape: subrectangular (1); rounded or acuminate (2); cruciate with rounded transverse pila extending laterally beyond mediana (3); cruciate with elongate, acuminate transverse pila extending laterally beyond mediana (4). Musser and Cracraft [44], character 189; Livezey and Zusi [95], character 1191.

357. Trabecula mediana or intermediana, fused to trabecula lateralis: no (1); yes (2). Noncomparable where trabeculae absent.

358. Trabecula mediana, fused to trabecula intermediana: no (1); yes (2). Noncomparable where trabeculae absent.

359. Trabecula mediana, mediolateral width: mediolaterally narrow, essentially confined to midline of sternum (1); mediolaterally wide (2).

360. Trabecula caudolateralis and mediana, relative caudal extents: mediana > caudolateralis (1); caudolateralis > mediana (2). Noncomparable if no notches or fenestrae present. Musser and Cracraft [44], character 190; Livezey and Zusi [95], character 1192.

361. Trabecula intermediana, relative caudal extent: subequal in length to mediana (1); subequal in length to caudolateralis (2); between lengths of craniolateralis and mediana (3). Noncomparable if trabecula intermediana absent. Musser and Cracraft [44], character 190; Livezey and Zusi [95], character 1192.

362. Trabecula intermediana, shape of terminus: round or subrectangular (1); acuminate (2); cruciate (3). Noncomparable if trabecula intermediana absent.

363. Trabecula intermediana, orientation of caudal terminal end: caudal, does not deviate from trabecula intermediana axis (1); medial (2); lateral (3). Noncomparable if trabecula intermediana absent.

364. Trabecula intermediana, orientation relative to body of sternum as reflected (in part) by angle defined by incisura caudolateralis, cranial vertex of angle: angle undefined, processus is parallel to trabecula mediana and "vertex" is ellipsoidal (1); laterally splayed, approximately 45 degrees (2). Noncomparable if trabecula intermediana absent.

365. Sternum, tracheal convolution: absent (0); present (1).

*Ribs*

366. Thoracic ribs, medial face, pneumatic foramina between capitulae: present (1); lost (2). Musser and Cracraft [44], character 191; Livezey [113], character 132.

367. Thoracic ribs, dorsal portion, craniocaudally thickened: yes (1); no (2).

368. Uncinate processes on thoracic ribs: absent (0); present (1).

369. Uncinate processes on thoracic ribs, fusion to ribs: fused (1); unfused (2). Noncomparable where absent.

370. Uncinate processes on thoracic ribs, shape of terminus: subrectangular (1); rounded (2). Noncomparable where absent.

371. Uncinate processes on thoracic ribs, angle of articulation: oriented caudally (1); oriented dorsocaudally (2). Noncomparable where absent.

372. Uncinate processes, angulus basalaris processi: present (1); absent on all processes (2). Noncomparable where absent.

373. Uncinate processes on thoracic ribs, length: truncate (1); elongate (2). Noncomparable where absent.

*Shoulder girdle: Coracoid*

374. Coracoid, recessus infra-acrocoracoideus: shallow (1); extremely deep (2).

375. Coracoid, insertion of ligamentum acrocoracoprocoracoideum strongly ventromedially protruding: no, essentially coplanar with craniocaudal axis of shaft of coracoid (1); yes (2). Musser and Cracraft [44], character 193; Livezey and Zusi [95], character 1267.

376. Coracoid, facies articularis clavicularis overhanging sulcus supracoracoideus or homologous site: no (1); yes (2). Mayr and Clarke [63], character 64.

377. Coracoid, facies articularis clavicularis, if overhanging sulcus supracoracoideus, exceptionally mediocaudally elongate and hamate (ventral aspect): no (1); yes (2). Noncomparable where not overhanging.

378. Coracoid, facies articularis clavicularis, if overhanging sulcus supracoracoideus, (medial aspect): subhorizontal (1); diagonal, with most proximal portion being located along dorsal margin (2). Noncomparable where not overhanging.

379. Coracoid processus acrocoracoideus, impressio ligamenti acrocoracohumeralis: deep (1); shallow (2). Musser and Cracraft [44], character 195; Livezey and Zusi [95], character 1276.

380. Omalis of coracoid, fused to furcula: no (1); yes (2).

381. Processus procoracoideus: present (1); lost (2). Musser and Cracraft [44], character 197; Livezey and Zusi [95], character 1283.

382. Processus procoracoideus, medial length: truncate (1); elongate (2). Noncomparable where processus procoracoideus absent. Musser and Cracraft [44], character 197; Livezey and Zusi [95], character 1283.

383. Processus procoracoideus, fused to head of coracoid: no (1); yes (2).

384. Processus procoracoideus, ankyolosed to furcula: no (1); yes (2).

385. Processus procoracoideus, terminus, shape: no spatulation or expansion present (1); spatulation and expansion of terminus present (2); tapering present (3). Noncomparable where processus procoracoideus absent or fused to head of coracoid.

386. Processus procoracoideus, terminus, caudally projecting, acuminate flange: absent (0); present (1). Noncomparable where processus procoracoideus absent or fused to head of coracoid.

387. Processus procoracoideus, tuberculum apicalis procoracoidei, ventral curvature: absent or slight (1); present (2). Noncomparable where processus procoracoideus absent.

388. Processus procoracoideus, extending distally as a craniocaudally elongate, sharp, crest along midline of shaft (crista procoracoidei of Livezey [113]): no (1); yes (2). Noncomparable where processus procoracoideus absent.

389. Processus procoracoideus, if extending as a craniocaudally elongate crista, additional foramen located cranially: no (1); yes (2). Noncomparable where processus procoracoideus absent or craniocaudally truncate.

390. Processus procoracoideus, if extending as a craniocaudally elongate crista, extreme medioventral projection of crista, even along distal margin: no (1); yes (2). Noncomparable where processus procoracoideus absent or craniocaudally truncate.

391. Scapular cotyla, form: shallow (1); deep, cuplike (2). Bertelli et al. [14], character 37.

392. Scapular cotyla, large pneumatic foramen or foramina: absent (0); present (1).

393. Coracoid shaft, foramen nervi supracoracoidei: absent (0); present (1). Mayr and Clarke [63], character 63.

394. Processus procoracoideus, pneumatic foramen directly below facies articularis scapularis which does not penetrate shaft: absent (0); present (1). Musser and Cracraft [44], character 198; Livezey and Zusi [95], character 1286; Mayr and Clarke [63], character 66.

395. Coracoid shaft, general form sensu length relative to width of facies articularis sternalis (not including additional projections): elongate—length between 3 and 4× width (1); typically proportioned—length between 2 and 3× width (2); truncate (less than 2× width) (3). Musser and Cracraft [44], character 199; Livezey and Zusi [95], character 1292.

396. Sternal coracoid, impressio musculi sternocoracoidei on dorsal surface of extremitas sternalis: shallow (1); deep (2). Musser and Cracraft [44], character 200; Livezey and Zusi [95], character 1294; Mayr and Clarke [63], character 67.

397. Sternal coracoid, impressio musculi sternocoracoidei, sulcus m. sternocoracoideus: absent, area smooth (0); present, defined ridges present (1).

398. Sternal coracoid, dorsal surface, impressio m. sternocoracoidei, pneumatic foramen, status: absent (0); present (1). Musser and Cracraft [44], character 201; Mayr and Clarke [63], character 67.

399. Sternal coracoid, crista articularis sternalis, cranial recurvature: absent (0); present (1).

400. Sternal coracoid, crista articularis sternalis, ventral recurvature: absent (0); present (1).

401. Sternal coracoid, medial angle: rounded (1); acuminate (2).

402. Sternal coracoid, medial angle, additional cranially oriented projection along cranial margin: absent (0); present (1). Noncomparable where crista extends along length of coracoid.

403. Sternal coracoid, medial angle, additional cranially oriented projection along cranial margin (if present), shape of apex: rounded (1); acuminate (2). Noncomparable where processus procoracoideus absent or craniocaudally truncate. Musser and Cracraft [44], character 202; Livezey [113], character 194.

404. Sternal coracoid, medial angle, additional cranially oriented projection along cranial margin (if present), craniocaudal length: craniocaudally truncate (1); craniocaudally elongate (2). Noncomparable where processus procoracoideus absent or craniocaudally truncate.

405. Sternal coracoid, processus lateralis, length: truncate (1); elongate (2).

406. Sternal coracoid, processus lateralis, shape of terminus: rounded or subrectangular (1); acuminate (2).

407. Sternal coracoid, processus lateralis, recurvature of cranial margin: absent or negligible (0); present (1).

408. Sternal coracoid, processus lateralis, recurvature of caudal margin: absent or negligible (0); present (1).

409. Sternal coracoid, processus lateralis, apex, orientation: lateral (1); rostral (2); caudal (3). Musser and Cracraft [44], character 203; Livezey and Zusi [95], character 1303.

410. Sternal coracoid, processus lateralis, apex, craniocaudal location: well cranial to lateral angle (1); just above lateral angle or subequal (2).

411. Sternal coracoid, lateral angle, mediolateral position relative to shaft of coracoid: medial to lateral margin of shaft (1); lateral to lateral margin of shaft (2).

412. Sternal coracoid, ventral aspect, distolateral corpus (just medial to lateral angle), deeply concave: no (1); yes (2).

413. Sternal portion of coracoid, ventral aspect, distomedial corpus (just lateral to medial angle), deeply concave: no (1); yes (2).

414. Sternal coracoid, facies articularis sternalis, labrum externa, general cranial extent of cranial margin (and correlated cranial expanse) relative to those of labrum interna:

former approximately equal to latter, producing facies articularis of dorsoventrally equal expanse (1); former distinctly caudal to latter, producing internally (dorsally) angled facies articularis (2); former significantly cranial to the latter, producing externally (ventrally) angled facies articularis (3).

415. Sternal coracoid, facies articularis sternalis, labrum externa, labrum along medial angle ventrocranially angled: no (1); yes (2). Musser and Cracraft [44], character 205; Livezey and Zusi [95], character 1314.

*Shoulder girdle: Furcula*

416. General shape: laterally splayed, "u-shaped" (1); mediolaterally compressed, "v-shaped" (2).

417. Apophysis, status: absent (0); present (1).

418. Apophysis if present, caudal prominence: diminutive, barely raised (1); extremely elongate (2); intermediate (3). Noncomparable where apophysis absent.

419. Apophysis (if present), continues ventrally beyond symphysis and is dorsoventrally elongate: no (1); yes (2). Noncomparable where apophysis absent.

420. Apophysis (if present), mediolaterally wide: no (1); yes (2). Noncomparable where apophysis absent.

421. Processus interclavicularis dorsalis: absent (0); present (1).

422. Scapus clavicle, mediolateral width: extremely thin (1); intermediate (2); extremely thick (3).

423. Scapus clavicle, ventral curvature: absent (0); present (1).

424. Scapus clavicle, pneumatic foramen on lateral face: absent (0); present (1).

425. Scapus clavicle, craniocaudal length: truncate (1); elongate (2).

426. Scapus clavicle, direction of compression: craniocaudal (1); mediolateral (2).

427. Symphysis, orientation of compression: craniocaudally flattened (1); slightly dorsoventrally flattened (2); completely dorsoventrally flattened (3).

428. Processus acrocoracoideus, length: truncate (1); elongate (2).

429. Processus acromialis, length: truncate (1); elongate (2).

430. Processus acromialis, dorsocaudal orientation: dorsal (1); slightly caudal (2); caudal, at a 90-degree angle to scapus clavicle (3).

431. Processus acromialis, mediolateral orientation: does not deviate from axis of scapus clavicle (1); lateral (2); medial (3).

432. Scapus clavicle, lateral recurvature: absent (0); present (1).

*Shoulder Girdle: Scapula*

433. Collum, medial face, pointed, ventrally oriented projection: absent (0); present (1). Musser and Cracraft [44], character 206; see also Musser et al. [44], character 41 for further discussion.

434. Collum, medial face, large pneumatic foramen: present (1); lost (2). Mayr [120], character 1248.

435. Collum, pronounced ventromedial bowing: absent (0); present (1).

436. Scapus, monotonic ventral curvature general to scapus, regardless of deviation of terminal end: straight to moderate, body and distal margin of scapula is slightly to moderately convex (1); pronounced, body and distal margin of scapula conspicuously convex (2). Musser and Cracraft [44], character 207; Livezey and Zusi [95], character 1260.

437. Scapus, lateral face, concavitas longitudinalis: present, concave throughout (1); lost, essentially planar throughout or shallow concavitas limited to cranial and medial portion (2). Musser and Cracraft [44], character 208; Livezey and Zusi [95], character 1257.

438. Scapus, ventrolateral face, tubercle of variable size located cranially, often accompanied by pitted crest trailing distally: present (1); lost (2). Musser and Cracraft [44], character 209.

439. Dorsal angle: prominent, deviates from body of scapula (1); small or lost, does not interrupt curvature of body of scapula (2).

440. Dorsal angle of scapula, apex, if discernible: cranial to midpoint of shaft (1); at midpoint of shaft (2); caudal to midpoint of shaft (3).

441. Terminal margin, shape: rounded or subrectangular (1); acuminate (2).

442. Terminal margin, narrowing and/or expansion (regardless of prominence of dorsal angle): none (1); spatulation and expansion of terminus present (2); tapering present (3). Musser and Cracraft [44], character 210; Livezey and Zusi [95], character 1264.

443. Terminal margin, ventral deviation from scapus: no (1); yes (2).

444. Terminal margin, tip, dorsal orientation of terminus: no (1); yes (2).

*Forelimb: Humerus*

445. Tuberculum ventrale and crista m. scapulohumeralis caudalis, general form and caudal prominence: smoothed and often rounded, not very prominent (1); extremely prominent and laterally flattened, projects well beyond rest of bone caudally (2). Musser and Cracraft [44], character 211; Livezey and Zusi [95], character 1364.

446. Tuberculum ventrale, proximal elevation regardless of fossa pneumotricipitalis exposure: located inferior to tuberculum dorsale, projects cranially (1); subequal in elevation to that of tuberculum dorsale, projects cranially (2); elevated proximally to (above) tuberculum dorsale, projects cranially or cranioproximally (3). Musser and Cracraft [44], character 213.

447. Tuberculum dorsale, proximodistally elongate: no (1); yes, tuberculum dorsale at least 1.5 times as long as it is wide (2). Musser et al. [44], character 65.

448. Incisura capitis, depth: extremely deep and prominent (1); shallow (2). Musser and Cracraft [44], character 215; Livezey and Zusi [95], character 1358.

449. Incisura capitis, distal closure: not closed by additional structures distally (1); enclosed by distal projection of caput humeri (2); closed by transverse ridge (3). Musser and Cracraft [44], character 221; Bertelli et al. [14], character 54.

450. Fossa pneumotricipitalis ventralis: pneumatic (1); apneumatic (2). Musser and Cracraft [44], character 216; Livezey and Zusi [95], character 1414–1415; Mayr and Clarke [63], character 77.

451. Fossa pneumotricipitalis (fossa pneumotricipitalis dorsalis): absent (0); present (1).

452. Fossa pneumotricipitalis dorsalis (if present), depth: shallow (1); deep and concave (2). Noncomparable where absent.

453. Proximal portion of humerus, caudal surface, fossa pneumotricipitalis, crus dorsale fossae: dorsoventrally narrow (1); dorsoventrally broad (2). Musser and Cracraft [44], character 219; Livezey [113], character 202.

454. Head, craniocaudally flattened: no, globose (1); yes (2). Musser and Cracraft [44], character 220.

455. Head, distally reaching apex near ventral tubercle: present (1); lost (2).

456. Head, orientation: horizontal or dorsoventral (1); diagonal (2)

457. Impressio coracobrachialis: absent or shallow (1); deep (2). Musser and Cracraft [44], character 222; Livezey and Zusi [95], character 1428.

458. Sulcus ligamentosus transversus, dorsoventral length: abbreviate (1); elongate, at least half dorsoventral width of humeral head (2). Musser and Cracraft [44], character 223; Livezey and Zusi [95], character 1431.

459. Sulcus ligamentosus transversus, depth: shallow (1); deep (2). Musser and Cracraft [44], character 223; Livezey and Zusi [95], character 1431.

460. Bicipital crest, proximodistal length relative to that of deltopectoral crest: less than 1/2 length of deltopectoral crest (1); over 1/2 length of deltopectoral crest (2). Musser and Cracraft [44], character 231; Livezey and Zusi [95], character 1383.

461. Bicipital crest, terminus on body of humerus, ventral margin: abruptly discontinued proximally on corpus humeri, ventral margin (1); gradually continued by shallow, low but distinct jugum along corpus humeri, ventral margin (2) Musser and Cracraft [44], character 227; Livezey and Zusi [95], character 1413.

462. Head of humerus, separation from crista deltopectoralis and tuberculum dorsale ("externus"): pronounced, caput offset and well distinguished from tuberculum dorsale (1); diminished, caput low and poorly distinguished adjacent features (2). Musser and Cracraft [44], character 228; Livezey and Zusi [95], character 1354.

463. Crista deltopectoralis, orientation relative to head of humerus: cranial (1); craniolateral (2). Musser and Cracraft [44], character 229; Livezey and Zusi [95], character 1374.

464. Crista deltopectoralis, shape: rounded (1); trapezoidal (2).

465. Crista deltopectoralis, comparative proximodistal length relative to that of body of humerus: great, well developed and extending at least 1/3 length of corpus humeri (1); small, diminutive, extending less than 1/3 length of corpus humeri (2). Musser and Cracraft [44], character 230; Livezey and Zusi [95], character 1382.

466. Proximal humerus, long and narrow accessory scar for m. supracoracoideus: absent (0); present (1). Musser et al. [44], character 61.

467. Proximal section of shaft with triangular cross section: absent (0); present (1). Musser and Cracraft [44], character 232; Bertelli et al. [14], character 56.

468. Caudal surface of proximal shaft with centrally positioned, raised muscle attachment scar for m. latissimus dorsi, pars caudalis: absent (0); present (1). Mayr (2013), character 20.

469. Fossa olecrani, depth: limited depth (1); present, markedly deep (2). Musser and Cracraft [44], character 236; Livezey and Zusi [95], character 1482.

470. Processus supracondylaris dorsalis greatly elongated proximodistally: no (1); yes (2). Musser and Cracraft [44], character 241; Bertelli et al. [14], character 57; Livezey and Zusi [95], character 1467.

471. Flexor process, proximodistally elongate: no (1); yes (2).

472. Sulcus tendinis m. scapulotricipitalis: absent (0); present, weakly defined, typically broad, shallow and truncate (1); present, conspicuously defined, usually narrow, deep and elongate (2). Not comparable for Spheniscidae. Musser and Cracraft [44], character 237; Livezey and Zusi [95], character 1488.

473. Fossa m. brachialis, dorsoventral position relative to median axis of humerus: ventral (1); medial (2). Musser and Cracraft [44], character 239; Livezey and Zusi [95], character 1460.

474. Impression of brachialis anticus (brachial depression): shallow and small, ovoid, or brachial depression nonexistent (1); deep and part of brachial depression, distal portion of humerus appears to have been "scooped" out (2). Musser and Cracraft [44], character 240; Livezey and Zusi [95], character 1456; Mayr and Clarke [63], character 80.

475. Condylus dorsalis humeri, proximal extent relative to distal margin of fossae m. brachialis: condylus distal to distal terminus of fossa and typically separated from distal margin of fossae m. brachialis by smooth area of bone from latter (1); condylus (proximal margin) typically extending at least proximal to distal margin of fossae m. brachialis (2); condylus markedly proximal to fossa, including extreme proximal condylorum dorsalis and ventralis (3). Musser and Cracraft [44], character 244; Livezey and Zusi [95], character 1447.

476. Condylus dorsalis humeri, proximal margin: rounded (1); acuminate (2). Musser and Cracraft [44], character 243; Livezey and Zusi [95], character 1451.

*Forelimb: Ulna*

477. Ulna greatly exceeding humerus in length: no (1); yes (2). Mayr and Clarke [63], character 82.

478. Processus cotylaris dorsalis, pronounced ventral orientation such that apex processi is approximately coplanar with dorsal face of ulna: absent, apex of process variably elevated dorsally to body of ulna (0); present (1). Not comparable for Spheniscidae. Musser and Cracraft [44], character 245; Livezey and Zusi [95], character 1492.

479. Processus cotylaris dorsalis, facies articularis relative to that of cotyla ventralis: less expansive (1); subequal (2). Not comparable for Spheniscidae. Musser and Cracraft [44], character 246; Livezey and Zusi [95], character 1496.

480. Processus cotylaris dorsalis, crista intercotylaris: crista rudimentary but evident despite more typically conformed cotylae (1); variably prominent, cotylae dorsalis and ventralis distinct (2). Musser and Cracraft [44], character 247; Livezey and Zusi [95], character 1497.

481. Impressio insertii m. brachialis: absent or shallow (1); deep (2). Musser and Cracraft [44], character 248; Livezey and Zusi [95], character 1502.

482. Incisura radialis: absent or indistinct (0); present and variably pronounced (1). Musser and Cracraft [44], character 249; Livezey and Zusi [95], character 1505.

483. Trochlea humeroulnaris: no articular facet visible or exhibits shallow articular facet (1); exhibits deep articular facet (2).

484. Impressio scapulotricipitalis, depth: shallow (1); deep (2).

485. Olecranon process, proximal prominence: prominent (1); limited, at about level of cotylae (2).

486. Distal ulna, dorsal aspect, tab-like distally oriented projection along ventromedial margin of labrum condyli dorsalis: absent (0); present (1).

487. Olecranon fossa, depth: shallow (1); deep (2).

488. Depressio radialis, depth: shallow (1); deep (2).

489. Sulcus tendinis, depth: shallow (1); extremely deep and pit-like (2).

*Forelimb: Carpometacarpus*

490. Trochlea carpalis, sulcus trochlearis: shallow, rounded in cranial or caudal view, or is somewhat deep laterally but not cranially (1); deep, subangular in cranial or caudal view (2). Musser and Cracraft [44], character 251; Livezey [113], character 236.

491. Marked ridge between processus pisiformis and trochlea carpalis: absent (0); present (1). Livezey [113], character 242; Mayr (2013), character 24.

492. Metacarpal III strongly bowed, delimiting a large spatium intermetacarpale: no (1); yes (2). Mayr and Clarke [63], character 85.

493. Trochlea carpalis, dorsal aspect, labrum dorsalis, proximal terminus of dorsal rim of trochlea: weakly angular or rounded (1); strongly angular, almost pointed, elongated proximally (2). Musser and Cracraft [44], character 252; Livezey [113], character 238.

494. Metacarpal III, ventral aspect, tuberculum intermetacarpalis (small tubercle on metacarpal III immediately distal to synostosis metacarpalis proximalis): elongate (1); distinct and rounded (2); obsolete (3). Musser and Cracraft [44], character 253; Livezey [113], character 244.

495. Synostosis metacarpalis distalis, dorsal aspect, sulcus interosseus ventralis: present, but shallow (1); lost (2); present, deep (3). Musser and Cracraft [44], character 254; Livezey [113], character 253.

*Forelimb: Manual Phalanges*

496. Manual digit II: phalanx 1, index process: absent or barely protruding (1); present, prominent (2).

497. Manual II:1: not perforated (1); perforated (2). Bertelli et al. [14], character 60; Strauch (1978), character 49.

*Pelvis*

498. Iliac blades and synsacrum: fused (1); unfused (2). Bertelli et al. (2013), character 178.

499. Thoracic vertebrae, if fused within preacetabular synsacrum and iliac blades, degree of fusion of transverse processes of thoracic vertebrae to iliac blades: transverse processes of 3 caudalmost thoracic vertebrae incompletely fused to preacetabular iliac blades (1); transverse processes of 1–2 caudalmost thoracic vertebrae incompletely fused to preacetabular iliac blades (2); transverse processes of all thoracic vertebrae completely fused to preacetabular iliac blades (3); transverse processes of all thoracic vertebrae incompletely fused to preacetabular iliac blades (4) Note: Here, incomplete fusion is defined by a lack

of fusion across entire or almost entire ventral suture. Noncomparable where iliac blades completely unfused to synsacrum. Musser and Cracraft [44], character 281.

500. Preacetabular and postacetabular ilium, dorsal face, relative craniocaudal lengths, indexed by ratio of length of former divided by length of latter: preacetabular ilium greater in craniocaudal length than postacetabular ilium, ratio significantly greater than unity (1); subequal (2); postacetabular ilium greater in craniocaudal length than preacetabular ilium, ratio less than unity (3). Musser and Cracraft [44], character 255; Livezey and Zusi [95], character 1890.

501. Preacetabular ilium, dorsal face, dorsomedial margin, dorsal iliac crest, dorsomedial synostosis forming carina iliaca dorsales: present (1); lost (2). Musser and Cracraft [44], character 256; Livezey and Zusi [95], character 1814; Mayr and Clarke [63], character 792.

502. Preacetabular ilium and synsacrum, dorsal margin of separated preacetabular ilia, if not fused into carina iliaca dorsales: subequal to height of crista spinosa synsacri and postacetabular synsacrum (1); ventrally positioned so that dorsal edge of preacetabular ilia are well ventral to crista spinosa synsacri (2). Noncomparable where carina iliaca dorsales present.

503. Carina iliaca dorsales (if present) extent of dorsal fossae created on either side of crista spinosa synsacra: moderate, approximately half or less the length of the carina (1); extremely elongate, extending proximally over half of the carina, typically extending proximolaterally (2); essentially absent (3). Noncomparable where carina iliaca dorsales absent.

504. Preacetabular synsacrum, dorsal prominence of crista spinosa synsacra or carina iliaca dorsales relative to postacetabular synsacrum: subequal in dorsoventral height (1); dorsally prominent and elevated above level of postacetabular synsacrum (2). Musser and Cracraft [44], character 257; Livezey and Zusi [95], character 1819.

505. Preacteabular ilium, lateral face, angle respective to transverse plane of synsacrum: oblique or subvertical (1); subhorizontal (2). Musser and Cracraft [44], character 258; Livezey and Zusi [95], character 1823.

506. Preacetabular ilium, shape of anterior terminus: subrectangular, does not narrow cranially (1); rounded (sometimes exhibits incomplete ossification) (2); trapezoidal, narrows cranially (3); triangular, narrows cranially toward apex (4).

507. Preacetabular ilium, anterior terminus, extends anteriorly well beyond proximal margin of synsacrum: no (1); yes (2).

508. Preacetabular tubercles, status: present (1); lost (2). Musser and Cracraft [44], character 259; Livezey and Zusi [95], character 1810; Mayr and Clarke [63], character 93.

509. Preacetabular tubercles, length: elongate (1); diminutive, close to body of pelvis (2). Noncomparable where preacetabulear tubercles absent. Musser and Cracraft [44], character 259; Livezey and Zusi [95], character 1810; Mayr and Clarke [63], character 93.

510. Preacetabular tubercles, shape of termini: subrectangular (1); acuminate (2). Noncomparable where preacetabulear tubercles absent. Musser and Cracraft [44], character 259; Livezey and Zusi [95], character 1810.

511. Acetabular ilium, dorsal face, interacetabular width relative to synsacral length: great, exceeds 1/2 synsacral length (1); moderate, 1/2 to 1/3 synsacral length (2); small, approximately 1/4 synsacral length (3); extremely small, approximately 1/6 synsacral length (4). Musser and Cracraft [44], character 260; Livezey and Zusi [95], character 1845.

512. Acetabular and postacetabular dorsal synsacrum, osseous sheet covering fenestrae intertransversae synsacrales: yes (1); lost or incomplete along medial or lateral margins (2).

513. Antitrochanter, lateral face: raised dorsally so that it is more visible and robust (1); depressed ventrally so that it is less visible and somewhat obscures the acetabulum (2). Musser and Cracraft [44], character 263.

514. Antitrochanter, craniolaterally oriented semicircular prominence on cranial margin of antitrochanter that thickens ventrally: absent (0); present (1). Musser and Cracraft [44], character 264.

515. Ilioischiadic foramen: absent (0); oblong (1); essentially circular (2). Musser and Cracraft [44], character 265; Livezey and Zusi [95], character 1791.

516. Obturator foramen, closed caudally: no (1); yes (2). Musser and Cracraft [44], character 268; Livezey and Zusi [95], character 1783.

517. Obturator foramen, shape: circular, often with wider dorsoventral margin (1); elongate, ovoid (2). Musser and Cracraft [44], character 268; Livezey and Zusi [95], character 1783.

518. Obturator foramen, axis of orientation: dorsal cranially and ventral caudally (1); craniocaudal or horizontal (2); ventral cranially and dorsal caudally (3). Musser and Cracraft [44], character 269.

519. Vertex craniolateralis ilii, dorsally protrudes above antitrochanteric sulcus: no (1); yes (2).

520. Vertex craniolateralis ilii, if dorsally protruding above antitrochanteric sulcus, shape: crest-like (1); elongate and tab-like, typically laterally directed (2). Noncomparable where not dorsally protruding.

521. Vertex caudolateralis ilii: tab-like laterally directed (1); a simple crest (2); lost (3); a ventrally directed flange (4). Noncomparable for Paleognathae. Musser and Cracraft [44], character 272; Livezey [113], character 286.

522. Postacetabular ilium and ischium, lateral face, fusion: unfused posteriorly (0); fused posteriorly, ilioischiatic fenestra is closed (1). Musser and Cracraft [44], character 273; Cracraft and Clarke [115], character 27.

523. Postacetabular ilium and ischium, lateral face, synchondrosis ilioischiadica: extremely smooth, barely visible or absent (1); line-like and distinctly etched craniocaudally or caudoventrally (2). Noncomparable for Paleognathae. Musser and Cracraft [44], character 275.

524. Postacetabular ilium and ischiium, lateral face, deep incisura in caudal margin: absent (0); present (1). Noncomparable for Paleognathae. Musser and Cracraft [44], character 276; Bertelli et al. [14], character 65; Livezey and Zusi [95], character 1787.

525. Caudal extent of caudal terminus of postacetabular ilium relative to that of ischium: ilium distinctly cranial to ischium (1); ischium distinctly cranial to ilium (2); subequal (3). Noncomparable for Paleognathae. Musser and Cracraft [44], character 278; Livezey and Zusi [95], character 1892.

526. Preacetabular synsacrum, ventral face, large spine protruding from cranial portion: absent (0); present (1). Musser and Cracraft [44], character 279.

527. Acetabular synsacrum, ankylosis of acetabular transverse process (often synostotic) to acetabular area beyond lateral margin of synsacrum: absent, transverse process does not ankylose to acetabular area (0); present, ankyloses to dorsal margin of acetabulum, often within dorsocaudal margin of the acetabulum (1); present, ankyloses to craniodorsal margin of ilioishciadic foramen with variable separation between acetabulum and ankylosis, accompanied by sessile concavity between ankylosis and margin of acetabulum (2); present, ankyloses to ventral site between caudal acetabular margin and cranial margin of ilioischiadic foramen, ankylosed to pila that extends from ankyloses between the ilioischiadic foramen toward the dorsal margin of the obturator foramen (3). Noncomparable for Paleognathae. Musser and Cracraft [44], character 280.

528. Preacetabular and acetabular synsacrum, ventral face, ventral sulcus of synsacrum: present (1); lost or barely discernible (2). Musser and Cracraft [44], character 282.

529. Postacetabular ischium, ventral face, foramen in oblique iliac crest: present (1); lost (2). Musser and Cracraft [44], character 283.

530. Postacetabular portion of synsacrum, dorsoventral position: postacetabular portion of synsacrum depressed dorsally (1); postacetabular portion of synsacrum raised ventrally (2). Musser and Cracraft [44], character 284; Livezey and Zusi [95], character 1793.

531. Fossa renalis, depth: shallow or lost (1); deep (2). Musser and Cracraft [44], character 284; Livezey and Zusi [95], character 1793.

532. Lamina of ischium anteriorly extensive so as to create pons for recessus caudalis fossae: yes (1); no (2). Noncomparable for Paleognathae. Musser and Cracraft [44], character 286; Livezey and Zusi [95], character 1802.

533. Ischial lamina creating pons for recessus caudalis fossae (if present), visible through ilioischiadic foramen in lateral aspect: no (1); yes (2). Noncomparable where recessus caudalis fossae absent.

534. Recessus caudalis fossae (if present): shallow (1); deep (2). Noncomparable where absent. Musser and Cracraft [44], character 286; Livezey and Zusi [95], character 1802; Mayr and Clarke [63], character 95.

535. Deep ovoid fossa along caudomedial length of postacetabular ilium and ischium forming recessus caudalis fossae (if present): absent (0); present (1). Noncomparable where absent. Musser and Cracraft [44], character 287; Livezey and Zusi [95], character 7427.

536. Caudal margin of postacetabular ilium and in relation to caudal portion of synsacrum: caudal portion of synsacrum separating from and continuing caudally past caudal margin of ilium and splaying laterally towards apex (1); caudal portion of synsacrum fused to caudal margin of ilium completely (2); caudal portion of synsacrum separating from and continuing caudally past caudal margin of ilium and tapered toward apex (3). Musser and Cracraft [44], character 289.

537. Caudalmost synsacral vertebrae, fused to costal processes of closest proximal vertebra: no (1); yes (2).

538. Caudalmost synsacral vertebrae, if fused to costal processes of closest proximal vertebra, number of fused vertebrae present: 2 (1); 3 or more (2). Noncomparable where unfused to costal processes.

539. Iliosynsacral incisura, shape: semicircular (1); subrectangular (2).

540. Processus marginis caudalis, shape of terminus: subrectangular (1); acuminate (2); rounded, tab-like (3). Musser and Cracraft [44], character 291; Livezey and Zusi [95], character 1808.

541. Processus marginis caudalis, termini invaginated on caudodorsal surface so that termini are medially oriented: no (1); yes (2). Musser and Cracraft [44], character 292.

542. Processus terminalis ischii, shape: acuminate (1); subrectangular (2); rounded (3). Musser et al. [44], character 82.

543. Processus terminalis ischii, ankylosed to pubis: no (1); yes (2).

544. Spatium ischiopubicum, dorsoventral width: wide (1); narrow, ischium and pubis nearly in contact (2).

545. Scapus pubis, exclusive of marked departures in apex: recurved, dorsal margin variably concave (1); essentially straight or slightly sigmoid, approximately aligned with major axis of apex of pubis (2); decurved, dorsal margin convex (3). Musser and Cracraft [44], character 294; Livezey and Zusi [95], character 1928.

546. Scapus and apex of pubis, length of extension beyond ischium: pubis not extending beyond caudal margin of ischium or, if it does, extended portion makes up less than one third of entire length of pubis (1); portion extending beyond caudal portion of ischium makes up at least one fourth that of entire length of pubis or more (2). Musser and Cracraft [44], character 295; Livezey and Zusi [95], character 2006.

547. Apex pubis, dorsoventral orientation: projecting dorsally (1); projecting ventrally (2). Musser and Cracraft [44], character 296; Livezey and Zusi [95], character 1945.

548. Apex pubis, dorsomedial form: dorsoventrally spatulate and cruciate (1); spatulation negligible, apex pubis subparallel along ventral and dorsal margins, terminal end subrectangular or acuminate (2). Note: Not to be confused with dorsal angle (can be seen in Rallidae) that is a point of attachment between pubis and ischium. Musser and Cracraft [44], character 297; Livezey and Zusi [95], character 1940.

549. Apex pubis, mediolateral curvature: absent or slight (1); present, prominent departure from craniocaudal axis, apex of pubis medial to caudal ischium (2); present, apex of pubis curves lateral to caudal ischium (3).

*Ossified Tendons*

    550. Intratendinous ossification in hindlimbs: absent (0); present (1). Musser and Cracraft [44], character 298; Bertelli et al. [14], character 80.

    551. Intratendinous ossification in forelimbs: absent (0); present (1).

*Femur*

    552. Crista trochanteris, markedly projected cranially: yes (1); no (2). Musser and Cracraft [44], character 299; Mayr and Clarke [63], character 97.

    553. Crista trochanteris, caudal face distal to facies articularis antitrochantericus: thickening developed (1); lacking distinct, distal thickening (2). Musser and Cracraft [44], character 300; Livezey [113], character 301.

    554. Fossa trochantericus: lost or shallow (1); present, deep and extends across entire width of facies articularis antitrochantericus (2). Musser and Cracraft [44], character 301; Livezey and Zusi [95], character 1978.

    555. Collum femoris, facies articularis antitrochanterica, caudal margin: located at 90-degree angle to femoral body (1); projecting proximally (2). Musser and Cracraft [44], character 303; Livezey and Zusi [95], character 1975.

    556. Proximal femur, pneumatic foramen on craniolateral side: absent (0); present (1). Mayr and Clarke [63], character 98.

    557. Sulcus patellaris, cranial aspect, ovate accessory subfossa in proximal region: absent (0); present (1). Musser and Cracraft [44], character 305; Livezey and Zusi [95], character 2039.

    558. Fossa poplitea: shallow, weakly delimited (1); deep (2). Musser and Cracraft [44], character 306; Livezey and Zusi [95], character 2047.

    559. Fossa poplitea, pneumatic foramen: small or absent (1); present, large (2). Musser and Cracraft [44], character 307; Livezey and Zusi [95], character 2048.

    560. Impressio ligamentum collateralis lateralis, depth: shallow (1); deep (2).

    561. Epicondylus medialis, prominence of tubercle: prominently elevated, typically tubercular (1); not prominently elevated, grading essentially smoothly (2). Musser and Cracraft [44], character 311; Livezey [113], character 308.

    562. Condylus lateralis, crista tibiofibularis, distal extent relative to condylus medialis: subequal to condylus medialis (1); distinctly and significantly distal to condylus medialis (2). Musser and Cracraft [44], character 312; Livezey and Zusi [95], character 2012.

    563. Shaft, cranial recurvature, status: absent (0); present (1).

    564. Condylus lateralis, caudal aspect, lateromedial width relative to that of condylus medialis: subequal (1); lateral condyle significantly wider (2); lateral condyle significantly narrower (3). Musser and Cracraft [44], character 314; Livezey and Zusi [95], character 2027.

    565. Condylus lateralis, relative proximodistal position of crista tibiofibularis and trochlea fibularis: crista tibiofibularis crest located more proximally than trochlea fibularis (1); subequal in proximal position (2); trochlea fibularis more proximal (3). Musser and Cracraft [44], character 315; Livezey and Zusi [95], character 2010.

    566. Condylus lateralis, relative caudal prominence of trochlea fibularis and crista tibiofibularis: caudal prominence subequal (1); crista tibiofibularis more prominent than trochlea fibularis (2).

    567. Condylus medialis, caudal aspect, general form: spherical, distal portion greatly rounded, extremely convex (1); triangular, distal portion somewhat rounded (2). Musser and Cracraft [44], character 316; Livezey and Zusi [95], character 2030.

    568. Condylus medialis, medial aspect, cranial face of body, proximal terminus: abrupt, subperpendicular or acuminate (1); comparatively gradual (2). Musser and Cracraft [44], character 317; Livezey and Zusi [95], character 2032.

    569. Crista supracondylaris medialis, medial aspect, form relative to condylus medialis and facies caudalis corporis: absent or rudimentary, condylus not extended by crista, caused by one distinct (often abrupt) angle or incision above the medial condyle, rarely also a second is present (comparatively proximal), interrupting an otherwise gradual curving

crest (1); present and prominent, medial condyle continued by crest without interruption by angle or incision above the medial condyle (2). Musser and Cracraft [44], character 318; Livezey and Zusi [95], character 2042.

570. Fovea tendineus m. tibialis cranialis, depth: shallow (1); deep (2).

571. Fovea tendineus m. tibialis cranialis, distal aspect, location: located cranially (1); located caudally (2). Musser and Cracraft [44], character 320; Livezey and Zusi [95], character 7428.

572. Femur, length compared to that of tibiotarsus: 1/2 length of tibiotarsus (1); over 1/2 length of tibiotarsus (2); less than 1/2 length of tibiotarsus (3).

*Tibiotarsus and Fibula*

573. Cnemial fossa retropatellaris: shallow (1); deep (2). Musser and Cracraft [44], character 321; Livezey and Zusi [95], character 2112.

574. Cnemial fossa retropatellaris, shape: circular (1); elongate and ovoid (2).

575. Cnemial fossa retropatellaris or homologous site, recessus and/or pneumatic foramina (pores): no (1); yes (2). Musser and Cracraft [44], character 322; Livezey and Zusi [95], character 2113.

576. Lateral fossa retropatellaris: shallow (1); deep (2).

577. Lateral fossa retropatellaris or homologous site, recessus and/or pneumatic foramina (pores): no (1); yes (2).

578. Incisura in caudal aspect of interarticular area delimiting facies articularis medialis: present (1); lost (2). Musser and Cracraft [44], character 323; Livezey and Zusi [95], character 7429.

579. Pneumatic foramen underneath jugum between facies articularis lateralis and area interarticularis (typically found distal to incisura, if present): present, 1 (1); present, 2 or more (2); lost (3). Musser and Cracraft [44], character 324.

580. Fossae retropatellares, angle relative to facies articularis medialis and lateralis: on same plane as articular area (1); proximocranially oriented between 45 and 80 degrees (2); proximocranially oriented at approximately 90 degrees so that fossae are perpendicular to articular areas (3).

581. Crista cnemialis and crista patellaris, apex, proximodistal height relative to head of tibiotarsus: at same height of fossae retropatellares or only slightly elevated above fossae (1); well projected proximally to fossae (2); extremely proximally projected, at least 3 times the length of the fossae above the fossae (3).

582. Crista cnemialis lateralis apex, proximodistal height relative to head of tibiotarsus: essentially at level of fossae retropatellares (1); proximal to fossae retropatellares (2); extremely proximally projected, at least 3 times the length of the fossae above the fossae (3); extremely distally projected (4).

583. Crista cnemialis lateralis and cranialis, distal apices (distal angles of cristae, not attachment locations on shaft of tibiotarsus), relative proximodistal heights: subequal (1); apex of crista cnemialis lateralis located proximally to that of crista cnemialis cranialis (2); apex of crista cnemialis cranialis proximal to that of crista cnemialis lateralis (3).

584. Crista cnemialis cranialis, pronounced lateral curvature (especially proximally): present (1); lost (2). Musser and Cracraft [44], character 329; Livezey and Zusi [95], character 2082.

585. Crista cnemialis lateralis, shape: hamate but intermediate, but hook rounded and does not protrude beyond jugum of lateral crista (1); more prominent and hamate, acuminate hook protrudes distolaterally beyond lateral crest jugum (2); rounded, tuberculate (3).

586. Crista cnemialis cranialis, especially cranially projected: no, but is still well projected (1); yes (2); no, extremely reduced (3).

587. Crista cnemialis cranialis, distal apex, shape: rounded (1); acuminate (2).

588. Crista cnemialis cranialis, apex of crista cnemialis cranialis and crista patellaris or apex of crista cnemialis cranialis only, shape: apex lost (0); rounded (1); acuminate (2).

589. Crista cnemialis cranialis, concavity of lateral and medial surfaces: both surfaces concave (1); medial surface convex, lateral surface concave (2); both surfaces convex (3); medial surface concave, lateral surface convex (4). Musser and Cracraft [44], character 328; Livezey and Zusi [95], character 2078.

590. Crista cnemialis lateralis, concavity of lateral and medial surfaces: both surfaces concave (1); medial surface convex, lateral surface concave (2); both surfaces convex (3); medial surface concave, lateral surface convex (4). Musser and Cracraft [44], character 328; Livezey and Zusi [95], character 2078.

591. Crista cnemialis lateralis, proximal crista, width compared to crista patellaris: extremely wide (1); narrow, subequal in width to crista patellaris (2).

592. Lateral fossa retropatellaris, extremely reduced: no (1); yes (2).

593. Facies articularis medialis: convex (1); concave (2).

594. Crista between fossae retropatellares: absent (0); present (1).

595. Facies articularis lateralis, concavity: absent or small (1); present, large (2).

596. Interarticular area: convex (1); concave (2).

597. Crista cnemialis lateralis, lateral prominence relative to articulated fibula: not lateral to fibular head (1); lateral to fibular head (2). Musser and Cracraft [44], character 331; Livezey and Zusi [95], character 2094.

598. Cristae cnemiales cranialis and lateralis, comparative distal extents on shaft of tibiotarsus: crista cnemialis cranialis terminating distinctly distal to crista cnemialis lateralis (1); cristae subequal in distal extent (typically truncated), both lacking jugae (2). Musser and Cracraft [44], character 332; Livezey and Zusi [95], character 2099.

599. Crista cnemialis cranialis, recurvature of distal margin: absent (0); present (1).

600. Fibula, head, form and position relative to head of tibiotarsus: elongate and extends caudally past interarticular area (1); essentially circular, from proximal view does not extend past interarticular area (2). Musser and Cracraft [44], character 333.

601. Fibular crest, length: over 1/4 length of tibiotarsus (1); about 1/4 length of tibiotarsus (2); less than 1/4 length of tibiotarsus (3).

602. Fibular crest, lateral prominence: little to none (1); prominently projected (2).

603. Fibular crest, if projected laterally, relative lateral prominence of proximal and distal ends: distal end more laterally projected (1); subequal (2). Not comparable where fibular crest not projected laterally.

604. Fibular crest, proximodistal location: just distal to head of tibiotarsus (1); close to midshaft of tibiotarsus (2). Musser and Cracraft [44], character 330; Livezey and Zusi [95], character 2085.

605. Foramen interosseum proximale: sublinear and incisurate, relatively narrow (1); approximately ovate, relatively spacious (2). Musser and Cracraft [44], character 335; Livezey and Zusi [95], character 2128.

606. Foramen interosseum distale, length as compared to length of the tibiotarsus: more than 1/4 length of the tibiotarsus (1); less than 1/4 length of tibiotarsus (2); exactly 1/4 length of tibiotarsus (3). Musser and Cracraft [44], character 336; Livezey and Zusi [95], character 2129.

607. Proximal shaft of tibiotarsus, cranial aspect: rounded (1); craniocaudally flattened and mediolaterally broad (2).

608. Epicondylus medialis: pronounced tubercle (1); lost or diminutive (2). Musser and Cracraft [44], character 337.

609. Epicondylus lateralis: lost or diminutive (1); pronounced tubercle (2).

610. Trochlea cartilaginis tibialis, lateral and medial margins: splayed laterally (1); medially compressed (2); only medial margin splayed laterally (3). Musser and Cracraft [44], character 338.

611. Trochlea cartilaginis tibialis, height: craniocaudally short (1); craniocaudally elongate (2). Musser and Cracraft [44], character 339.

612. Trochlea cartilaginis tibialis, rectangular, medially oriented projection on medial margin: absent (0); present (1). Musser and Cracraft [44], character 340.

613. Condylus lateralis, depressio epicondylaris lateralis: shallow or absent (1); deep (2). Musser and Cracraft [44], character 341.

614. Condylus lateralis, depressio epicondylaris lateralis, accessory depression located along caudal margin of condylus lateralis: absent (0); present (1). Musser and Cracraft [44], character 341.

615. Condylus lateralis, depressio epicondylaris lateralis, accessory depression (if present), depth: shallow (1); deep (2). Noncomparable where depressio epicondylaris lateralis absent.

616. Condylus lateralis, proximocranial rim, tubercle: absent (0); present (1).

617. Condylus medialis, depressio epicondylaris medialis: shallow or absent (1); deep (2).

618. Condylus medialis, depressio epicondylaris medialis, accessory depression located along caudal margin of condylus medialis: absent (0); present (1).

619. Condylus medialis, depressio epicondylaris medialis, accessory depression (if present), depth: shallow (1); deep (2). Noncomparable where depressio epicondylaris absent.

620. Condylus medialis and condylus lateralis, relative positions: pulled in medially (1); spread laterally (2).

621. Medial and lateral condyles, incisura intercondylaris, distal aspect, foveae (sulcus) transcondylares medialis and lateralis: present (1); lost (2). Musser and Cracraft [44], character 344; Livezey and Zusi [95], character 2155.

622. Tuberositas retinaculi extensorius lateralis, ossified retinaculum extensorum where medial portion is almost completely ossified or is completely ossified (retinaculum m. fibularis): absent (0); present (1).

623. Tuberositas retinaculi extensorius lateralis, if ossified retinaculum absent, comparative prominence of lateral and medial cristae: subequal (1); medial crista more prominent (2); lateral crista more prominent (3).

624. Tubercle laterodistal to pons supratendinous: absent (0); present (1). Mayr and Clarke [63], character 101.

625. Tubercle laterodistal to pons supratendinous (if present), prominence: diminutive (1); prominent (2). Noncomparable where tubercle absent. Mayr and Clarke [63], character 101.

626. Crista extending proximally from tubercle laterodistal to pons supratendinous or homologous site: absent (0); present (1).

627. Crista extending proximally from tubercle laterodistal to pons supratendinous or homologous site (if present), length: proximodistally truncate (1); proximodistally elongate (2). Noncomparable where crista absent.

628. Crista extending proximally from tubercle laterodistal to pons supratendinous or homologous site (if present), foramen at base: absent (0); present (1). Noncomparable where crest absent.

629. Tuberositas retinaculi extensorius medialis: absent (0); present (1).

630. Medial and lateral condyles, relative cranial extensions (compare looking at distal surface): subequal (1); medial condyle extended more cranially (2).

631. Medial and lateral condyles, relative mediolateral thickness (compare looking at distal surface): subequal (1); medial condyle mediolaterally thinner (2); lateral condyle thinner (3).

632. Tuberositas retinaculi extensorius medialis (if present), form: tab-like, prominent (1); small and diminutive crest (2). Noncomparable where absent.

633. Tuberositas retinaculi extensorius medialis (if present), craniomedial orientation: oriented medially (1); oriented cranially (2). Noncomparable where absent.

634. Tuberositas retinaculi extensorius medialis (if present), proximodistal location: just above pons supratendinous (1); proximally located (2). Noncomparable where absent.

635. Tuberositas retinaculi extensorius lateralis, proximodistal location: just above pons supratendinous (1); proximally located (2); extends distal to pons supratendinous (3).

636. Tuberositas retinaculi extensorius lateralis, extreme lateral extension: no (1); yes (2).

637. Pons supratendinous, location of distal opening: above medial condyle (1); just lateral to medial condyle but within medial margin of tibiotarsus (2); centered along midline (3).

638. Canalis extensorius, depth: shallow (1); deep (2); extremely deep and cavernous (3).

639. Medial and lateral condyles, relative proximal extents: lateral condyle located more proximally than medial condyle (1); subequal (2); medial condyle located more proximally than lateral condyle (3).

640. Medial condyle, distal rim distinctly notched: no (1); yes (2). Mayr and Clarke [63], character 102.

*Tarsometatarsus*

641. Medial cotyle, medial margin, exceptionally projected proximally and crista-like: no (1); yes (2).

642. Medial cotyle, plantar margin, exceptionally projected proximally and crista-like: no (1); yes, moderately projected to roughly the same level of the proximal margin of the hypotarsal eminence (2); yes, prominently projected proximal to hypotarsal eminence (3).

643. Medial cotyle, proximodistal orientation of dorsal margin: cotyla perpendicular to shaft of tarsometatarsus, proximal surface of cotyla not visible in dorsal aspect (1); dorsal margin of cotyla positioned slightly more distally than that dorsal half of cotyla somewhat visible in dorsal aspect (2); dorsal margin of cotyla positioned much more distally than that of plantar margin so that cotyla is largely visible in dorsal aspect (3).

644. Lateral cotyle, dorsal margin, crista-like margin and/or subtending angle present: yes (1); no, dorsal margin of cotyle essentially absent and cotyla is coplanar with and almost indistinguishable from cotyla corpus (2).

645. Lateral cotyle, depth: deep, concave (1); flat or only slightly concave (2).

646. Lateral cotyle, proximodistal orientation of dorsal margin: dorsal margin of cotyla positioned slightly more distally than that dorsal half of cotyla somewhat visible in dorsal aspect (1); cotyla perpendicular to shaft of tarsometatarsus, proximal surface of cotyla not visible in dorsal aspect (2); dorsal margin of cotyla positioned much more distally than that of plantar margin so that cotyla is largely visible in dorsal aspect (3).

647. Hypotarsal eminence, proximally prominent: no (1); yes (2).

648. Lateral and medial cotylae (dorsal perspective), relative proximal elevation: cotyla lateralis distinctly distal to cotyla medialis (1); cotyla lateralis subequal to cotyla medialis (2); cotyla medialis distinctly distal to cotyla lateralis (3). Musser and Cracraft [44], character 346; Livezey and Zusi [95], character 2250.

649. Fossa parahypotarsalis lateralis, depth: shallow or barely visible (1); deep (2). Musser and Cracraft [44], character 347; Livezey and Zusi [95], character 2258.

650. Fossa parahypotarsalis medialis, depth: shallow or barely visible (1); deep (2). Musser and Cracraft [44], character 348; Livezey and Zusi [95], character 2257.

651. Hypotarsus, crista(e) medialis hypotarsi and crista(e) lateralis hypotarsi, relative plantar prominence: crista(e) medialis hypotarsi more plantarly prominent than crista(e) lateralis hypotarsi (1); crista(e) subequal in plantar prominence (2); crista(e) medialis hypotarsi less plantarly prominent than crista(e) lateralis hypotarsi (3). Musser and Cracraft [44], character 349; Livezey [113], character 336.

652. Crista medialis hypotarsi and crista lateralis hypotarsi, relative distal extents and magnitude: distal-most terminus(i) of medial crest(s) much more distally extensive, lateral crista(e) proximodistally truncate, about 1/2–2/3 proximodistal length of the medial crista (1); subequal (2); distal-most terminus(i) of lateral crest(s) more distally extensive, medial crista(e) proximodistally truncate, approximately 1/2 proximodistal length of the lateral crista, and reduced to a mediolaterally thin, osseous sheet (3); distal-most terminus(i)

of lateral crest(s) more distally extensive, medial crista(e) proximodistally truncate, reduced to at least approximately 1/3 or less proximodistal length, thin lamina (4).

653. Hypotarsus, tendon of m. flexor digitorum longus (FDL) enclosed by a canal: no, sulcus present for m. (1); m. enclosed in a canal (2). Noncomparable where muscle absent or sulcus/canal for muscle absent/indiscernible, or where homology is uncertain. Musser and Cracraft [44], character 350; Livezey and Zusi [95], characters 2284–2286; Mayr and Clarke [63], character 103.

654. Hypotarsus, canal for tendon of m. flexor digitorum longus (FDL, if present), centrally located: no (1); yes (2). Noncomparable where not enclosed in canal.

655. Hypotarsus, tendon of m. flexor perforatus digiti II (FPII) enclosed in a canal: no, sulcus present for m. (1); m. enclosed in a canal (2). Noncomparable where muscle absent or sulcus/canal for muscle absent/indiscernible, or where homology is uncertain. Musser and Cracraft [44], character 352; Bertelli et al. [14], character 75.

656. Hypotarsus, tendon of m. flexor perforans et perforatus digiti II (FPPII) enclosed in a canal: no, sulcus present for m. (1); m. enclosed in a canal (2). Noncomparable where muscle absent or sulcus/canal for muscle absent/indiscernible, or where homology is uncertain.

657. Hypotarsus, tendon of m. flexor hallicis longus (FHL) enclosed in a canal: no, sulcus present for m. (1); m. enclosed in a canal (2). Noncomparable where muscle absent or sulcus/canal for muscle absent/indiscernible, or where homology is uncertain.

658. Fossa infracotylaris dorsalis, depth: shallow (1); deep (2). Musser and Cracraft [44], character 354; Livezey and Zusi [95], character 2259; Mayr and Clarke [63], character 104.

659. Proximal tarsometatarsus, dorsal margin of lateral cotyla, additional, subtriangular, distal-reaching tuberositas present that may be concave laterally: absent (0); present (1).

660. Proximal tarsometatarsus, dorsal aspect (medial portion), arcus extensiorius (ossified retinaculum): not present, remains of retinaculum are only impressio retinaculi extensorii (1); present (2). Mayr [120], character 35.

661. Proximal portion of tarsometatarsus, dorsal aspect, foramina vascularia proximalia: essentially equal in height (1); lateral foramina distinctly distal to medial foramina (2); medial foramina distal to lateral foramina (3). Musser and Cracraft [44], character 353; Livezey and Zusi [95], character 2264.

662. Sulcus extensorius, proximal portion medial to dorsal foramina vascularia proximalia, depth: essentially absent (1); present, shallow (2); present, deep (3).

663. Sulcus extensorius, proximal portion lateral to dorsal foramina vascularia proximalia, depth: essentially absent (1); present, shallow (2); present, deep (3).

664. Dorsal foramina vascularia proximalia, enclosed in ovoid depression that is plantar to plane of sulcus extensorius: yes (1); no, foramina coplanar with sulcus extensorius (2); yes, but only lateral foramen is plantar to sulcus extensorius (3). Musser and Cracraft [44], character 354; Livezey and Zusi [95], character 2259.

665. Proximal portion of tarsometatarsus, dorsal aspect, foramina vascularia proximalia, proximodistally elongate: no (1); yes (2).

666. Proximal portion of tarsometatarsus, plantar aspect, foramina vascularia proximalia, proximodistally elongate: no (1); yes (2); only one is proximodistally elongate, typically medial foramen (3).

667. Proximal portion of tarsometatarsus, dorsal aspect, additional foramen or pair of circular foramina proximal to foramina vascularia proximalia that do not open to plantar face: absent (0); present, one foramen (1); present, pair of foramina (2).

668. Proximal portion of tarsometatarsus, plantar aspect, foramina vascularia proximalia, number: a pair is present (1); a pair is present with an additional foramen located on the lateral side (2); only one foramen is present (3); a pair is present with at least an additional foramen located on the medial side (4). Musser and Cracraft [44], character 355.

669. Proximal portion of tarsometatarsus, plantar aspect, foramina vascularia proximalia: lateral foramina distinctly distal to medial foramina (1); lateral and medial foramina

about equal in height (2). Noncomparable where only one foramen is present. Musser and Cracraft [44], character 356. Ksepka and Clarke [121], character 78.

670. Sulcus extensorius, depth along shaft of tarsometatarsus: shallow or absent (1); deep sulcus depth is roughly half that of tarsometatarsus (2); cavernous, only planter lamina of tarsometatarsus present (3). Musser and Cracraft [44], character 357; Livezey and Zusi [95], character 2305.

671. Sulcus extensorius, medial and lateral cristae, relative dorsal prominence: medial crista more dorsally prominent than lateral crista (1); subequal (2); lateral crista more dorsally prominent than medial crista (3).

672. Tuberositas m. tibialis cranialis, dorsal prominence: prominent and raised dorsal to sulcus extensorius (1); diminutive, barely raised dorsally (2).

673. Tuberositas m. tibialis cranialis, number: 1 (1); 2 (2). Noncomparable for taxa with scoring (1) for character (canal) due to fusion of tuberosities.

674. Tuberosities m. tibialis cranialis, relative proximodistal lengths: subequal in proximodistal length (1); medial tuberosity much more proximodistally elongate (2). Noncomparable where one tuberositas present or where tuberosities fused.

675. Tuberosities m. tibialis cranialis, subtending ossified bridge and canal opening distally along midline of tarsometatarsus: absent (0); present (1).

676. Processus calcaris: absent or diminutive (1); present (2).

677. Fossa supratrochlearis plantaris: absent or indistinct (0); present, distinctly concave (1). Musser and Cracraft [44], character 363; Livezey and Zusi [95], character 2329.

678. Trochlea metatarsi II, plantarly prominent eminentia (medio) plantaris (ala): absent (0); present (1).

679. Trochlea metatarsi II, eminentia (medio) plantaris (ala): rounded (1); subrectangular (2). Noncomparable in absence of ala. Musser and Cracraft [44], character 364; Livezey and Zusi [95], character 2352.

680. Trochlea metatarsi II, eminentia (medio) plantaris (ala, if present): projects plantarly (1); projects plantomedially (2); projects laterally (3). Noncomparable in absence of ala.

681. Trochlea metatarsi III, proximodistal length as compared to that of body of tarsometatarsus: length from proximal to distal end is less than 1/5 length of entire tarsometatarsus from top of eminentia intercotylaris to distal end of trochlea metatarsi III (1); length from proximal to distal end is 1/5 or greater length of entire tarsometatarsus (2). Musser and Cracraft [44], character 365.

682. Trochlea metatarsi IV, foveae ligamentorum collateralium, status and form sensu depth and width relative to that of associated trochlea: present, moderate depression (1); small, shallow, almost absent (2). Musser and Cracraft [44], character 366; Livezey and Zusi [95], character 2349.

683. Trochleae metatarsalia II-IV, distal view, relative dorsal elevations: II < III $\geq$ IV and II < IV (1); II < III > IV and II = IV (2); II = III = IV (3). Musser and Cracraft [44], character 367; Livezey and Zusi [95], character 2363.

684. Trochleae metatarsalia II-IV, relative distal extents: II < III $\geq$ IV and II < IV (1); II < III > IV and II subequal to IV (2); II > IV, III $\geq$ IV (3).

685. Distal portion of tarsometatarsus, trochlea metatarsi II, facies articularis phalangealis, sulcus trochlearis (narrow groove between lateral and medial rims of trochlea): remains distinct on facies dorsalis (1); obsolete on facies dorsalis, terminating medially toward fovea ligamentorum collaterallium at distal apex of trochlea (2). Musser and Cracraft [44], character 368; Livezey [113], character 354.

686. Trochlea metatarsi II, plantarly deflected: no (1); yes (2). Mayr and Clarke [63], character 108.

687. Trochlea metatarsi IV, plantar ala: not prominent, plantar extent significantly less than that of trochlea metatarsi II (1); conspicuously prominent, plantar extent subequal to that of trochlea metatarsi II (2). Note: not to be confused with flange on plantar surface of trochlea metatarsi II, common in many genera.

688. Pedal digit 4: phalanges 3 and 4, relative lengths: phalanx 4 longer (1); both phalanges of equal length (2); phalanx 4 shorter (3). Bertelli et al. [14], character 76.

689. Three anterior toes notably elongated: absent (0); present (1). Bertelli et al. [14], character 77.

690. Pedal I:1, pronounced bowing: absent (0); present (1). Musser et al. [44], character 97.

691. Pedal I:1, length as compared to III:1: not as follows (1); I:1 about half the length of III:1 (2). Mayr and Clarke [63], character 110; Mayr [47], character 32.

692. Pedal IV:1, medially protruding phalange at proximal margin: absent (0); present (1). Mayr [47], character 35.

693. Proximal end of pedal II:2 extending little beyond proximal end of III:2: no (1); yes (2). Mayr [47], character 33.

694. Olfactory bulb, size: absent (0); present (1).

695. Olfactory bulb, position: ventral (0); dorsal (1).

696. Olfactory bulb, conformation: clearly bifurcated (0); single tract (1).

697. Telencephalon, shape in dorsal view: rounded (0); anteriorly tapered (1).

698. Wulst, dorsal expansion in anterior view: not expanded (0); distinctly enlarged (1).

699. Wulst, anteroposterior development in dorsal view: extends from olfactory bulb to cerebellum (0); positioned posteriorly (1); anteriorly positioned (2).

700. Wulst, lateral expansion of anterior wulst in dorsal view: not expanded laterally (0); expanded or swollen laterally beyond lateral margin of the telencephalon (1).

701. Junction between telencephalon and cerebellum in dorsal view: v-shaped (0); curved (1).

702. Posterior telencephalon, shape in dorsal view: rounded (0); pointed (1).

703. Contact between telencephalon and dorsal optic lobe in lateral view: straight (0); curved (1).

704. Relative size of optic lobe in lateral view (relative to rhombencephalon): small (0); large (1).

705. Optic lobe in dorsal view: visible (0); not visible (1).

706. Occipital sinus: not visible (0); clearly demarcated on dorsal cerebellum (1).

707. Dorsal projection of cerebellum in lateral view: <50% of posterior margin of telencephalon in contact with cerebellum (0); >50% of posterior margin of telencephalon in contact with cerebellum (1).

708. Relative length of telencephalon and cerebellum in lateral view: short cerebellum (0); long cerebellum (1).

709. Cerebellum, width in dorsal view: narrow (0); moderately wide (1); wide (2).

710. Cerebellum, fissures: distinct (0); indistinct (1).

711. Cerebellum, floccular lobes, mediolateral length: reduced (0); short (1); elongate (2).

712. Cerebellum, floccular lobes, fenestration: absent (0); present (1).

713. Cerebellum, floccular lobes, shape in posterior view: triangular with tapered lateral margin (0); rectangular with thickened dorsal and ventral margins (1).

714. Rhombencephalon, sulcus: conspicuous (0); inconspicuous (1).

715. Pituitary gland, contact with optic tract: separated (0); contacting (1).

716. Carotid artery, entrance into pituitary gland fossa: single (0); paired (1).

717. Brain and endosseous labyrinth, relative position and orientation: vertical long-axis (0); horizontal long-axis (1).

718. Semicircular canal, cross-sectional shape in lateral view: rounded (0); compressed (1).

719. Semicircular canals, number of common crus: two (asc-psc and hsc-psc) (0); three (asc-psc, hsc-psc, and asc-hsc) (1).

720. Endosseous labyrinth, cochlear curvature in lateral view: curved (0); straight (1).

721. Endosseous labyrinth, shape of distal cochlea in lateral view: tapered (0); swollen (1).

**Table A1.** Optimized synapomorphies of selected groups recovered through constrained analyses employing the molecular backbone constraints based on the phylogenies of Kimball et al [66], Reddy et al. [65] and Prum et al. [64]. Synapomorphies with a CI of 1.0 are noted with a subscript 1. Asterisks indicate unambiguous synapomorphies. Dashes indicate groups that were unresolved in strict consensus trees.

| Group Name | Kimball et al. [66] Constraint | Reddy et al. [65] Constraint | Prum et al. [64] Constraint |
|---|---|---|---|
| Gruiformes + Charadriiformes | 1(1)*, 2(1), 27(1)*, 62(0), 74(1), 75(1)*, 92(2)*, 98(1)*, 125(1)*, 142(1), 154(1)*, 173(2)*$^1$, 181(2), 188(1), 223(2)*, 229(2), 236(2), 292(1)*, 311(2), 326(1)*, 333(1), 334(2)*, 341(2)*, 345(2), 361(1), 369(2)*, 401(2), 406(2)*, 407(1), 436(2), 439(2), 443(2)*, 452(2), 453(1), 493(2), 494(2), 506(2), 518(1), 527(1), 546(1), 552(1)*, 565(2), 586(2)*, 606(1), 611(2), 623(3), 641(2)*, 673(2), 677(1), 679(2)*, 686(2), 693(1)*, 698(0), 699(2) | - | - |
| Charadriiformes (including *S.mikkelseni* and *N. grandei*) | 4(1), 16(2)*, 21(2)*, 46(1), 55(1), 68(1), 69(1), 71(2), 86(1)*, 91(2), 99(2)*, 108(0), 111(1), 129(1)*,163(0), 167(2), 180(2)*, 183(2)*, 206(2)*, 237(2), 253(2), 261(2)*, 270(0)*, 279(2)*, 299(1)*, 315(0), 317(0)*, 342(2)*, 346(2)*, 356(3), 359(1), 362(2), 366(2)*, 377(2)*, 393(0), 404(1)*, 449(3), 450(2)*, 451(1)*, 469(1), 470(2)*, 473(1), 489(2), 495(3), 498(2)*, 519(1), 527(3), 532(2)*, 535(1), 554(2), 555(2)*, 579(1), 595(2)*, 598(2)*, 610(2)*, 615(1), 643(2)*, 652(1)*, 671(2)*, 672(2), 691(2)*, 692(0), 719(1)*, 720(0)* | 11(3), 62(0), 71(2), 84(0), 90(2)*, 91(2), 92(2), 108(0), 120(1), 125(1), 129(1), 163(0), 173(2)*, 181(2), 183(2)*, 206(2)*, 223(2), 237(2), 238(1)*, 279(2), 299(1), 311(2), 317(0), 324(2), 326(1), 333(1), 334(2), 341(1), 345(2), 356(3), 359(1), 361(1), 366(2), 369(2)*, 377(2), 391(2), 393(0), 406(2), 407(1), 449(3), 450(2), 451(1), 453(1), 470(2), 479(2), 489(2), 493(2), 495(3), 498(2)*, 506(2), 521(1)*, 535(1), 552(1), 554(2), 584(1), 586(1)*, 595(2), 610(2), 641(2)*, 652(1), 668(3), 670(2)*, 672(2), 677(1)*, 679(2), 686(2), 692(0), 698(0), 707(0), 717(0), 719(1), 720(0), 721(1) | 1(1), 10(1)*, 15(2)*, 18(3)*, 20(1), 32(1), 44(1), 58(1)*, 61(2), 75(1), 86(1), 98(1)*, 125(1)*, 129(1), 144(1), 151(1), 155(1), 165(1), 173(2)*, 175(1), 176(1), 223(2), 248(2)*, 258(1)*, 262(1), 273(2)*, 279(2)*, 292(1)*, 324(2), 338(2), 340(1)*, 355(2)*, 356(3)*, 359(1)*, 377(2)*, 385(1), 400(1), 407(1), 437(1), 443(2), 448(1), 450(2)*, 451(1)*, 453(1)*, 454(2), 455(1), 468(1), 470(2)*, 477(2)*, 478(1)*, 487(2)*, 489(2)*, 497(2), 502(1), 517(1), 527(2)*, 540(1), 546(1)*, 547(1), 579(1), 583(1), 588(2), 592(2), 597(2), 598(2)*, 623(3), 632(2), 639(1), 651(1), 652(1), 662(3), 671(2), 679(2)*, 687(2), 719(1)*, 720(0) |
| *S. mikkelseni* + (*Turnix* + *N. grandei*) | 47(2), 87(1), 90(2)*, 103(1), 104(2), 120(2), 142(2), 188(2), 190(1), 229(1), 232(2), 292(2)*, 296(2), 297(2), 306(2), 319(1), 320(1), 341(1), 348(0), 375(1), 385(2), 395(1), 408(0), 411(1), 412(1), 416(2), 418(2), 419(2), 422(1), 425(2), 430(1), 436(1), 439(1), 442(2), 444(2), 445(1), 458(1), 464(1), 466(1), 472(2), 492(2), 521(1)*, 526(1), 539(2), 542(3), 549(1), 553(1)*, 558(2), 568(1), 579(3), 585(1), 589(2), 596(1), 600(2), 605(1), 621(2), 622(1), 626(0), 685(1), 717(0) | - | - |
| *Turnix* + *N. grandei* | 34(1)*, 91(1), 150(2), 155(2), 160(2), 192(2), 225(1), 245(2), 511(2), 513(1), 582(2), 586(1)* | - | - |

**Table A1.** *Cont.*

| Group Name | Kimball et al. [66] Constraint | Reddy et al. [65] Constraint | Prum et al. [64] Constraint |
|---|---|---|---|
| Charadriiformes (excluding *Scandiavis mikkelseni* and *Nahmavis grandei*, excludes *Turnix* under Kimball et al. 2019 constraint) | 18(1)*, 78(2)*, 204(2)*, 216(2), 258(1)*, 272(0)*, 273(2)*, 280(2), 308(2)*, 356(4), 392(1), 396(1), 401(1), 413(2)*, 429(2)*, 454(2)*, 457(2), 467(1)*, 477(2)*, 478(1)*, 481(1), 487(2), 502(1)*, 510(2)*, 512(2), 518(2), 531(1), 543(2), 552(2), 570(1)*, 577(1), 584(2), 590(1)*, 591(2)*, 608(2)*, 632(2), 638(1), 688(3)* | 15(2), 18(1)*, 36(0), 40(1), 49(0), 78(2), 150(2), 151(1)*, 154(1)*, 225(1), 227(1)*, 242(2), 292(0), 565(2)*, 582(2), 611(2)* | - |
| Crown Charadriiformes (Most exclusive clade containing fossil humeri) | 21(1), 51(1), 52(1)*,87(3)*, 98(2)*, 108(1)*, 216(1), 222(0), 237(1), 248(2)*,280(1), 299(2), 323(2)*, 336(2)*, 341(3)*, 342(1), 345(1)*, 346(1), 355(2)*, 356(3), 392(0), 401(2), 418(3), 436(1), 449(1), 481(2), 494(1), 540(2)*, 555(1)*, 558(2)*, 568(1), 592(2), 609(2)*, 610(1)*, 611(1), 641(1), 643(1)*, 650(1), 653(1)*, 663(2), 667(0), 672(1), 680(2)* | 42(1)*, 43(1)*, 47(1), 50(2), 71(1), 90(1)*, 136(2), 146(1), 148(2)*, 223(2), 227(2)*, 280(1), 313(1)*, 339(2)*, 347(0), 369(1)*, 370(1), 416(1), 483(2)*, 507(2), 519(2)*, 565(2), 636(1)*, 693(2), 699(0), 716(1), 717(1)*, 721(0) - | - |
| SMF av 619 + (*Chionis alba* + IGM 100/1435) | 20(2), 21(2), 24(4), 31(2), 32(1), 34(1), 43(1), 44(3), 50(3), 53(1), 56(2), 66(1), 69(3), 75(2), 77(1), 85(2), 86(2), 87(4), 99(1), 112(2), 116(1), 122(2), 124(2), 158(1), 163(1), 165(2), 177(1), 179(2), 183(1), 204(1), 215(2), 216(2), 223(1), 239(1), 242(2), 246(1), 249(3), 273(1), 282(3), 285(1), 299(1), 312(2), 313(1), 317(1), 329(2), 333(2), 334(3), 342(2), 371(1), 385(3), 406(1), 418(1), 427(2), 443(1), 455(2), 461(1), 469(2), 474(2)*, 479(1), 487(1), 494(3), 501(1), 504(2), 506(1), 507(2), 514(1), 530(2), 531(2), 572(2), 573(1), 575(2), 577(2), 581(1), 583(1), 584(1), 588(1), 589(2), 590(3), 592(1), 599(1), 601(2), 608(1), 615(1), 638(2), 652(2), 664(2), 672(2), 677(0), 683(2), 691(1) | - | 20(2), 24(4), 31(2), 32(1), 34(1), 50(3), 53(1), 56(2), 58(1), 66(1), 69(3), 75(2), 77(1), 78(2), 85(2), 86(2), 87(4), 99(1), 116(1), 140(1), 148(1), 158(1), 177(1), 179(2), 183(1), 204(1), 223(1), 242(2), 260(1), 273(1), 285(1), 299(1), 312(2), 317(1), 329(2), 333(2), 334(3), 342(2), 369(2), 370(2), 371(1), 379(1), 406(1), 418(1), 425(1), 427(2), 443(1), 455(2), 461(1), 469(2), 474(2)*, 479(1), 483(1), 493(2), 501(1), 504(2), 506(1), 514(1), 530(2), 531(2), 572(2), 577(2), 581(1), 583(1), 584(1), 589(2), 590(3), 592(1), 599(1), 601(2), 608(1), 615(1), 632(2), 638(2), 652(2), 671(2), 672(2), 677(0) |

**Table A1.** *Cont.*

| Group Name | Kimball et al. [66] Constraint | Reddy et al. [65] Constraint | Prum et al. [64] Constraint |
|---|---|---|---|
| *Chionis alba* + IGM 100/1435 | 475(1)* | 20(2), 24(4), 31(2), 32(1), 34(1), 44(3), 50(3), 53(1), 56(2), 58(1), 66(1), 69(3), 75(2), 77(1), 78(2), 85(2), 86(2), 87(4), 99(1), 116(1), 140(1), 148(1), 158(1), 177(1), 179(2), 183(1), 204(1), 216(2), 223(1), 242(2), 260(1), 273(1), 285(1), 299(1), 312(2), 317(1), 329(2), 333(2), 334(3), 342(2), 369(2), 370(2), 371(1), 379(1), 406(1), 418(1), 425(1), 427(2), 443(1), 455(2), 461(1), 469(2)*, 474(2)*, 475(1)*, 479(1), 483(1), 493(2), 494(3), 501(1), 504(2), 506(1), 514(1), 530(2), 531(2), 577(2), 581(1), 583(1), 584(1), 589(2), 590(3), 592(1), 599(1), 601(2), 608(1), 615(1), 632(2), 638(2), 652(2), 671(2), 672(2), 677(0), 683(2), 691(1), | 475(1)* |
| Gruiformes (including extinct taxa) | 34(1), 42(2), 49(0), 67(2)*, 77(1), 124(2), 131(2), 138(1), 197(3), 228(2), 289(1), 345(3), 350(2), 378(2)*, 379(2), 388(2)*, 423(0)*, 426(1), 433(1), 468(1), 475(1), 504(2)*, 506(3), 511(2)*, 530(2)*, 534(2)*, 537(2)*, 542(2)*, 561(1), 599(1), 624(1)*, 625(2), 636(2)*, 644(2), 661(2)*, 674(2)* | 2(1), 4(2)*, 16(1)*, 27(1), 32(1), 46(2), 49(0), 67(2)*, 74(1)*, 77(1), 84(0), 85(1), 87(3), 92(2), 112(1), 125(1)*, 126(1), 142(1)*, 174(1)*, 188(1)*, 197(3)*, 206(1), 210(1), 236(2)*, 246(2)*, 268(2), 289(1), 291(2), 292(1), 311(2)*, 325(1), 326(1), 328(2), 334(2), 341(2), 345(3), 378(2)*, 388(2)*, 392(0), 396(2)*, 401(2), 406(2), 408(1), 423(0)*, 426(1), 430(1), 433(1), 436(2), 437(1)*, 439(2), 442(3), 443(2), 456(2)*, 472(1), 504(2)*, 506(3)*, 516(2), 518(1), 520(2), 527(1)*, 530(2)*, 531(2)*, 534(2)*, 537(2)*, 546(1)*, 551(1), 553(2)*, 558(1), 561(1), 603(1), 606(1)*, 636(2)*, 641(2)*, 644(2), 646(1), 670(2)*, 679(2)*, 686(2) | 27(1), 36(1), 49(0), 67(2)*, 74(1)*, 124(2), 125(1)*, 131(2), 154(1), 173(2)*, 185(2), 188(1)*, 223(2)*, 227(1), 228(2)*, 229(2), 238(1), 269(2), 292(1), 311(2)*, 324(2), 326(1), 369(2)*, 388(2)*, 406(2), 426(1), 433(1), 436(2), 439(2), 504(2)*, 506(3), 514(1), 518(1)*, 527(1)*, 530(2)*, 534(2)*, 537(2)*, 542(2), 550(1)*, 551(1), 552(1)*, 558(1), 561(1)*, 584(1), 599(1)*, 624(1)*, 633(1), 641(2)*, 644(2), 674(2)*, 679(2)*, 686(2)*, 693(1)*, 721(1) |
| Kimball et al. (2019): Ralloidea + (Messelornithidae + *Songzia*) Reddy et al. (2017): Messelornithidae + (*Songzia* + Ralloidea) Prum et al. (2015) Messelornithidae + Ralloidea (including *Songzia*) | 65(1), 90(2)*, 123(2), 158(1), 257(2)*, 272(0)*, 297(2)*, 318(2), 338(2), 349(2), 351(1), 353(2), 355(2)*, 375(1), 385(2), 400(0), 413(2), 421(1), 430(1), 446(3), 454(2), 467(1)*, 474(2), 485(2), 490(2), 494(3), 515(2), 521(4), 523(2), 569(1), 590(1), 596(1), 622(1)*, 623(2), 645(2), 652(3), 655(2), 660(2)*, 662(3), 664(2) | 1(1), 13(2), 65(1), 90(2), 123(2), 181(2), 237(2), 250(2), 254(1), 257(2)*, 266(0), 272(0)*, 297(2)*, 318(2), 323(1), 338(2), 339(1)*, 344(1), 351(1), 353(2), 356(2)*, 375(1), 413(2), 421(1), 446(3)*, 461(2), 467(1)*, 490(2), 493(2)*, 513(2), 515(2), 521(4), 523(2), 560(1), 570(1), 572(2), 575(1), 590(1), 611(2), 622(1)*, 623(2), 645(2), 652(3), 662(3), 664(2) | 1(1), 2(1)*, 30(1)*, 46(1), 55(1), 65(1), 90(2)*, 91(2), 123(2), 181(2), 195(2), 208(1)*, 215(2), 218(1)*, 237(2), 253(2), 257(2)*, 258(1), 272(0)*, 297(2)*, 318(2), 338(2), 349(2), 353(2), 355(2), 356(2)*, 371(1), 430(1), 446(3)*, 453(1)*, 454(2), 467(1)*, 490(2), 493(2)*, 505(2), 510(2), 521(3), 523(2), 540(2), 545(1), 560(1), 565(2), 570(1), 572(2), 579(3), 611(2), 622(1), 623(2), 645(2), 652(3), 655(2)*, 656(2), 660(2), 662(3), 685(1) |

**Table A1.** *Cont.*

| Group Name | Kimball et al. [66] Constraint | Reddy et al. [65] Constraint | Prum et al. [64] Constraint |
|---|---|---|---|
| Messelornithidae + *S. acutunguis* | 10(1), 13(2)*, 55(1), 62(1), 180(2), 225(1), 261(2), 279(2), 313(1)*, 334(1), 340(1)*, 344(1)*, 366(2), 407(0), 436(1), 441(2), 498(2), 500(3), 507(2), 512(2), 524(1), 540(2), 554(2), 555(2), 608(2), 609(2), 650(1), 671(1) | - | - |
| Crown Ralloidea (including *S. acutunguis* under latter two constraints) | 64(2)*, 66(1)*, 91(2)*, 138(2), 293(2), 308(2), 347(1), 360(2)*, 416(2), 426(2), 450(2)*, 561(2), 613(2)*, 651(2), 677(0) | 34(1)*, 62(0), 64(2), 138(2), 293(2), 308(2), 329(2), 360(2), 361(1), 362(3), 407(1), 422(1)*, 426(2), 450(2), 541(2), 550(0)*, 551(0), 561(2), 613(2), 651(2), 671(3) | 34(1), 62(0)*, 64(2)*, 66(1)*, 138(2), 159(2), 308(2)*, 334(3)*, 347(1)*, 360(2)*, 362(3), 396(1), 407(1)*, 425(2), 426(2), 450(2)*, 551(0), 613(2)*, 620(1)*, 649(1), 651(2)*, 654(1) |
| Messelornithidae | 34(2), 361(3), 371(1), 542(3), 543(2), 550(1)*, 551(1)*, 691(2) | 10(1)*, 55(1)*, 66(2), 180(2)*, 225(1)*, 261(2)*, 262(1)*, 279(2)*, 334(1), 366(2), 371(1), 436(1), 441(2), 498(2)*, 500(3), 507(2)*, 512(2), 524(1), 540(2), 542(3), 543(2), 554(2), 555(2)*, 608(2), 609(2), 650(1)*, 677(1), 691(2) | 10(1)*, 13(2), 68(2), 127(1), 180(2)*, 225(1)*, 261(2)*, 279(2)*, 313(1), 340(1)*, 344(1)*, 361(3), 366(2), 436(1), 441(2), 498(2)*, 500(3), 507(2)*, 512(2), 524(1), 542(3), 554(2), 555(2)*, 608(2), 609(2), 650(1)*, 671(1)*, 677(1), 691(2) |

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
