# Peer review of "New Remains of Scandiavis mikkelseni Inform Avian Phylogenetic Relationships and Brain Evolution"

_diversity, doi:10.3390/d13120651_

Round 1

Reviewer 1 Report

The mansucript is certainly interesting and the specimen presented is quite intriguing.

Although quite well illustrated, the description of both skull and endocast feels incomplete. This section needs to be expanded with comparisons with different extant Charadriiformes.

Discussion is also incomplete and it is more a list and/or repetition of characters listed in the description and in the phylogenetic analysis section. With the exception of the phylogenetic analyses, paleobiological and paleobiogeographical conclusions are not well supported.

Paleobiological inferences are poorly sustained, mainly based on the lack of an extensive description, the absence of proper references for senses/function associated with brain region, and comparisons with extant charadriiforms. Moreover, paleobiogeographic discussion and conclusions needs to be extended. Neither the information nor the references are presented to justify the conclusion reached.

All my comments and suggestions are included in the attached pdf.

Reviewer 2 Report

This is a very thorough piece of work extending knowledge on an important period in avian evolution through the study of an exquisite fossil attributed to Scandiavis mikkelseni. I have no major concerns with the work, but note that I suspect the journal has removed all your formatted italicisation of genus/species names throughout the text.  There are a couple oversights, eg Eocene penguin endocasts were not mentioned, and re the claim on line 35 - this is incorrect - there are more than one Paleogene marine fauna, eg all those dominated by penguins, such as the Paleocene Waipara River fauna in NZ which has multiple higher level taxa; Seymour Island faunas are not lacustrine. Yes some major Northern Hemisphere paleogene sites are lacustrine  - but Fur is not the only marine one globally.

Several instances of reference not converted to numerals were noted.

Figure 6 incorrectly delimits Charadriiformes from Pan-Charadriiformes as Turnix IS a charadriiform.

Overall  a great piece of work robustly presenting a new hypothesis for these fossil taxa.

some minor things indicate don the pdf.

Reviewer 3 Report

Title: a bit presumptuous. I suggest instead "Palaeoneurological description of Scandiavis mikkelseni and its bearing…"

Li 52: I suggest citing Knoll et al. (2018) as an example of the use of Synchrotron in the analysis of a fossil bird.

Knoll et al. 2018. A diminutive perinate European Enantiornithes reveals an asynchronous ossification pattern in early birds. Nature Communications, 9 (937): 1–9.

Li 61-62, 422-423: Name them please. Don't forget undescribed (partial or not) endocasts (e.g., Ludiortyx, Presbyornis). See also the specimen described by Mayr et al. (in press), even though it is more likely to be Oligocene in age than Eocene.

Mayr et al. 2021. Cranium of an Eocene/Oligocene pheasant-sized galliform bird from western North America, with the description of a vascular autapomorphy of the Galliformes. Journal of Ornithology (advance online publication)

Li 143-147: I think it's wrong to use "part and counterpart" for this specimen, which was just split into halves.

Li 178 and below: please always italicize binomina.

Table 3: I suggest using "neurocranium" instead of "cranium" here and, in fact, some explanation/diagram as what you actually measured would be welcome. Furthermore, how can the total skull length be less than the sum of the rostrum and neurocranium?  

Li 168: What do you mean "inferiorly"? Ventrally?

Li 201: The rostral extremity of the rostrum seems to bend dorsally (artefact?) before curving ventrally at the very tip (not visible in the holotype). I think you should describe this aspect in order to make it clear what curvature you are referring to.  

Li 215-216: I don't understand what you mean.

Li 246: This depends on how you define the horizontal. If you had taken the LSC as a proxy for the horizontal (instead of the long axis of the skull as I suppose you did), the skull would have been more ventrally tilted in Fig. 3.

Li 286: What do you mean "interpenetration"?

Li 305: "Surface" is better than "wall".

Li 311: Again, it depends on your point of reference.

Li 312: What do you mean "caudal"?

Discussion: Hoch (1975: pl. 2 figs 3, 4) figured two natural endocasts from the Eocene of Denmark. Although they are likely to belong to Columbiformes instead of Charadriformes (although I'm not sure Scandiavis is a charadriiform, it is certainly not a columbiform), I would appreciate some comparisons with those specimens.

Hoch, E. (1975) Amniote remnants from the eastern part of the Lower
Eocene North Sea basin. In: Lehman, J.P. (Ed.) Problèmes actuels de
Paléontologie (Évolution des Vertébrés). Paris: Centre National de la
Recherche Scientifique, pp. 543–562.

Li 503-516: You should cite and discuss Benson et al. (2017) here.

Benson, R.B.J., Starmer-Jones, E., Close, R.A. and Walsh, S.A. (2017) Comparative analysis of vestibular ecomorphology in birds. Journal of Anatomy, 231, 990–1018.

Round 2

Reviewer 2 Report

The authors have revised the manuscript and adequately addressed all previous concerns. So I recommend it be accepted pending some minor English edits I have made on the pdf.  Many edits relate to the fact that characters are seen in members of a group ie in gruiforms or charadriiforms, not in Gruiformes or in Charadriiformes, which entities house species.  Similarly, it is strictly more proper to have characters in species than in a genus (as a genus has species and is defined by a species; species may be defined by characters). Various other minor edits are indicated in the comments panel and offered to make this paper the best it can be.

Note I suggest an edit to the title and that the species name be added to figure captions so they make sense alone.

Reviewer 3 Report

I still think the title is misleading. The works does not "elucidate*" *avian phylogenetics and brain evolution. Also, the authors shouldn't use both caudal and posterior, rostral and anterior etc. It is also expected from them to acknowledge the reviewers and the editor, who have worked benevolently.
